# Forecasting the dynamics of a complex microbial community using integrated meta-omics

**Francesco Delogu** [1] ✉**, Benoit J. Kunath**[1]**, Pedro M. Queirós**[1]**, Rashi Halder**[1]**, Laura A. Lebrun**[1]**, Phillip B. Pope** [2,3]**, Patrick May** [1]**, Stefanie Widder**[4]**, Emilie E. L. Muller** [5] **& Paul Wilmes** [1,6] ✉

Predicting the behaviour of complex microbial communities is challenging. However, this is essential for complex biotechnological processes such as those in biological wastewater treatment plants (BWWTPs), which require sustainable operation. Here we summarize 14 months of longitudinal meta-omics data from a BWWTP anaerobic tank into 17 temporal signals, explaining 91.1% of the temporal variance, and link those signals to ecological events within the community. We forecast the signals over the subsequent five years and use 21 extra samples collected at defined time intervals for testing and validation. Our forecasts are correct for six signals and hint on phenomena such as predation cycles. Using all the 17 forecasts and the environmental variables, we predict gene abundance and expression, with a coefficient of determination ≥0.87 for the subsequent three years. Our study demonstrates the ability to forecast the dynamics of open microbial ecosystems using interactions between community cycles and environmental parameters.

Microorganisms are ubiquitous on planet Earth[1] and constitute up to 17% of its carbon biomass[2]. Microbial lineages are continuously evolving to fill a diverse set of ecological niches, balancing their complementary metabolic capabilities to form communities[1] which, in turn, affect biogeochemical cycles[3]. Understanding the temporal dynamics of microbial ecosystems and their links to the environment has become a common problem for many research fields spanning biomedicine, agriculture, biotechnology and climate change. While forecasting community composition dynamics has been successfully achieved for some environments (for example, refs. 4,5) and explored theoretically[6], the forecasting of gene expression dynamics over time in relation to environmental conditions remains an open challenge[7]. Although previous work[8,9] have approached the problem in marine systems with relatively

stable environmental conditions, such as those associated with marine oxygen minimum zones, using meta-omic (DNA, RNA, protein) and environmental parameter information to model biogeochemical cycles, a generalized framework for time-variable integration of meta-omic datasets into models of community ecology remains to be established.

The surface community of biological wastewater treatment plants (BWWTPs) represents an excellent candidate to become a model system to establish such a modelling framework for the following three reasons[10]. Firstly, BWWTPs share the challenges linked to most environments, as it is an open system with a constant influx of new populations[11] and exchange of matter and energy with the environment (that is, access to open air and sun irradiation). However, these challenges can be mitigated by keeping operational parameters (for example,

[1]Luxembourg Centre for Systems Biomedicine, University of Luxembourg, Esch-sur-Alzette, Luxembourg. [2]Faculty of Biosciences, Norwegian University of Life Sciences, Ås, Norway. [3]Faculty of Chemistry, Biotechnology and Food Science, Norwegian University of Life Sciences, Ås, Norway. [4]Department of Medicine 1, Research Division Infection Biology, Medical University of Vienna, Vienna, Austria. [5]Génétique Moléculaire, Génomique, Microbiologie, UMR 7156 CNRS, Université de Strasbourg, Strasbourg, France. [6]Department of Life Sciences and Medicine, Faculty of Science, Technology and Medicine, University of Luxembourg, Esch-sur-Alzette, Luxembourg. ✉e-mail: fra.delogu92@gmail.com; paul.wilmes@uni.lu

pH, phosphate and nitrate) within a controllable range. In addition, the microbial community biodiversity is of intermediate range, especially for the floating biomass, allowing fairly comprehensive data acquisition. Secondly, BWWTP communities share common metabolic pathways, albeit every local community has its own equilibrium, and its detailed makeup depends on the operational parameters, geographical location and inflow composition[12–14]. Microbial communities in BWWTPs possess dynamics at different temporal scales that are rather well described: the microbial and chemical composition of the inflow is known to change according to the time of day, the day of the week and the inflow volume[15]. In addition, temperature-driven seasonality has been found to influence the community[12,16], notably the surface community[17], as well as multi-annual trends. While one-time destructive perturbations show an impact on the community (for example, human interventions (such as bleaching, shutdowns[18,19]) and weather (that is, rain)), they are all monitored or encoded in the standard operational parameters of the plants. Finally, forecasting the behaviour of microbial communities in BWWTPs is highly desirable as stable operation allows reclamation of clean water as well as the harnessing of chemical energy[20]. Moreover, its functioning has to minimize undesired production (and uncontrolled release) of greenhouse gases such as $N_2O$[21]. In particular, the surface community is recognized to be a potential source of neutral lipids, a family of molecules of high added value usable for third-generation biodiesel production[20].

When dealing with complex microbial communities, far from lab-scale experiments, empirical modelling can enable efficient representation and forecasting (see Supplementary Information for a short summary of techniques). To achieve this, we explore a combination of strategies to first extract all the temporal information in an agnostic manner, such as through singular value decomposition (SVD)[22], and then perform forecasting by explicitly computing temporal cycles and link those patterns directly to the explanatory variables. SVD can decompose a matrix into two separate matrices of eigenvectors and a vector of eigenvalues (the technique is further explored in refs. 23,24). When applied to gene abundance (or expression) data over time, the first matrix is associated with the set of temporal patterns underlying the data and the second with the 'loadings' (that is, how much each individual gene is contributing to each pattern). The seasonal version of the forecasting method, autoregressive integrated moving average (ARIMA), computes cyclical (seasonality), autoregressive (temporal self-dependence), differencing (difference between consecutive timepoints) and moving-average (averaging of consecutive timepoints) components of a time series[25]. It thereby offers a very flexible framework for time-series modelling[25].

Here we combine SVD and several time-series algorithms into a generalizable framework for modelling the temporal dynamics of multilayered meta-omics data (Extended Data Fig. 1). We demonstrate the power of this framework through analysis of integrated meta-omic and environmental parameter datasets from a microbial community enriched in lipid-accumulating organisms (LAOs) on the surface of the anaerobic tank of the BWWTP in Schifflange (Luxembourg). The sample set comprises 51 time-resolved samples collected between March 2011 and May 2012 for training, with 21 additional samples collected between 2012 and 2016 for testing and validation. For both sets, the biomolecules were co-extracted[26] and the data for the training set were presented in a previous study[27]. We reconstructed the metagenomic (MG) structure of the community, alongside its taxonomy, genetic potential, transcript and protein levels. We employed SVD to extract relevant temporal patterns, which were then clustered into 17 fundamental signals. These were integrated with collected environmental parameters to build an ARIMA model, augmented with seasonal components that could explain the observed signals. Multiple models (ARIMA, Prophet[28] and NNETAR neural network model[29]) were trained to forecast the signals for the subsequent 5 yr. These models are flexible and customizable, being able to explain complex time series, breaking them down to the individual components (ARIMA and Prophet) or

being all-purpose powerful (neural network). However, ARIMA assumes that the parameters behind the process are constant, while Prophet can model a time-dependent evolution of the ARIMA parameters. On the other hand, ARIMA is the only model (in the current R package fable[29] implementation) where it is possible to obtain information about the contribution of the individual variables to the forecasting, making it suitable as an explicative model. This allowed us to correctly predict the gene abundance and expression of the populations in the community.

## Results and Discussion

### Characterization of the microbial community

From the experimental period between 21 March 2011 and 3 May 2012 (ref. 27), we previously obtained and analysed 51 weekly samples, to which we added 21 samples collected in the month of June during the years 2012–2016. The 72 samples were submitted to the same meta-omic analyses—MG, metatranscriptomics (MT) and metaproteomics (MP)—and processed individually to obtain 72 metagenomic assemblies, collections of metagenome-assembled genomes (MAGs), plasmids, viruses, unbinned prokaryotic chromosomal contigs and the corresponding gene expression at the transcriptional and proteomic levels. The combined datasets of the previous time series alongside the new samples were analysed together with updated bioinformatic workflows to allow a coherent comparison between samples along the time series while addressing the batch effect arising from combining the two sets (see Methods and Extended Data Fig. 2a). To form coherent sets spanning the whole time series, we individually clustered the bins (prokaryotic and eukaryotic) and the contigs (viral, plasmid and unbinned) according to their sequence (see Methods), which led to a total of 144 representative MAGs (rMAGs) and 1,681,736 representative contigs (rContigs), yielding 4,711,952 open reading frames (ORFs) (Supplementary Tables 1 and 2). A KEGG Orthology group (KO term) was assigned to 55% of the total retrieved ORFs, while taxonomic affiliations were assigned to 38.5%. The number of ORF copies as well as their detected gene expression and protein abundances were determined over the extended dataset (see Methods). We found on average $2.2 \times 10^6 \pm 4.8 \times 10^5$ (s.d.) ORFs, $9.1 \times 10^5 \pm 1.7 \times 10^5$ transcripts and $2.4 \times 10^5 \pm 2.5 \times 10^4$ protein groups per sample. However, most of the genes were not found to be expressed over the entire dataset or were only detected in a few samples. This suggests that an important fraction of the gene pool in the LAOs is not specifically required for community function or their expression levels are below the detection limit, hinting that their cumulative functional effort may be compartmentalized. This finding supports previous results[30] showing how a large portion of the community is redundant and only few functions are keystone. Read recruitment (on the ORF level) per sample was on average $59 \pm 9\%$ for the MG and $82 \pm 3\%$ for the MT, and peptide matching was $27 \pm 4\%$. The recruitment improved from the previous work on the same datasets, which reported that $26 \pm 3\%$ and $27 \pm 3\%$ of the MG and MT reads mapped against the MAGs, respectively[27]. This is due to an update of the bioinformatic tools used and the inclusion of all the unbinned contigs longer than 1,000 nt in the analysis (see Methods).

The rMAGs spanned the expected phyla of the BWWTP community and included members of the Actinobacteria, Bacteroidetes, Chlorobi, Fusobacteria, Nitrospirae, Proteobacteria and Spirochaetes, in addition to *Candidatus* Gracilibacteria (Fig. 1a), thereby reproducing previously described results[27]. On a more detailed taxonomic level, we were able to identify three strains of *Microthrix parvicella* and 17 strains of *Moraxella* spp. At no point over the course of the time series did a single rMAG largely dominate the community, but the combined populations of the genera *Microthrix* and *Moraxella* exhibited a percentage abundance with medians of 15.9% and 3.6%, respectively[31,32]. The majority of the contigs were not affiliated with defined MAGs (Fig. 1b) and are probably coming from incomplete genomes and alternative regions of the rMAGs, thus encapsulating the within-population diversity of the LAO community.

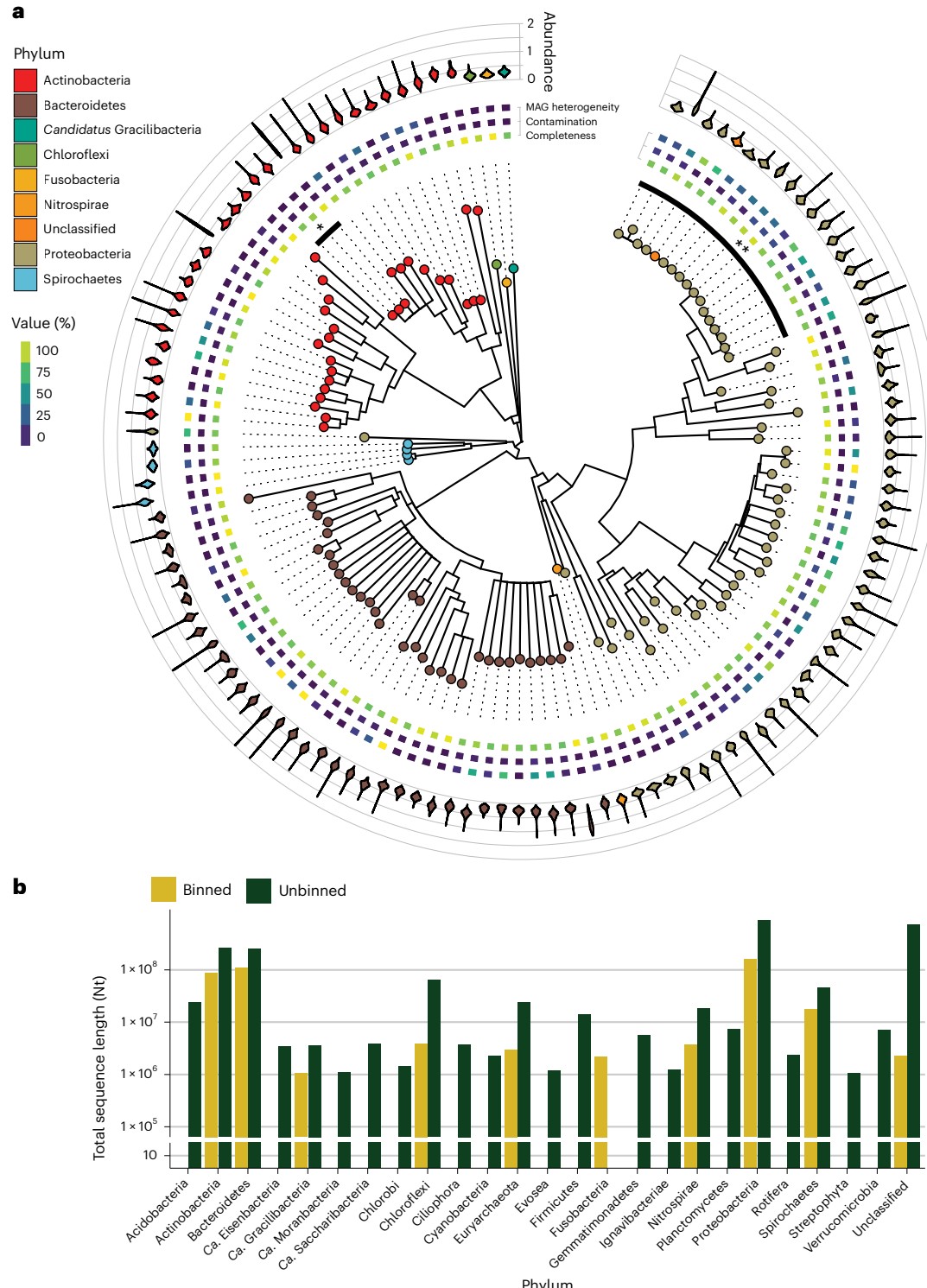

**Fig. 1 | Diversity and quality of the rMAGs and their representativeness in the meta-omic dataset. a**, Phylogenetic tree of the rMAGs in the LAO community (generated using GTDB-Tk[79]) contains the 126 bacterial rMAGs in the system (the 18 archaeal MAGs were not included). The heat map ring contains the CheckM quality measures per rMAG (completeness, contamination and MAG-originally strain heterogeneity), which were filtered to be at least 75% complete and at a maximum 25% contaminated (median: 2%). The violin plots contain the time-averaged (train time series) depth profiles over the contigs forming the rMAG. The two sections of the tree noted as * and ** highlight the strains of *M. parvicella* and *Moraxella* sp., respectively. **b**. The cumulative length of the contigs (longer than 1,000 nt; see Methods) for the 25 most abundant phyla displayed for the rMAGs and unbinned contigs.

## The temporal signals underlying the microbial community

Considering that the information necessary to forecast the community dynamics and linked gene expression may be most represented in any biological (for example, taxonomical or functional representation) or environmental data layer, we decided to include multiple layers in our analysis (the whole workflow is depicted in Extended Data Fig. 1). Regarding the microbial community, we explored multiple taxonomic and functional levels at once and summarized their temporal characteristics. Thus, the three quantification matrices (MG, MT and MP) were used to compute 'summary' matrices according to the ORF descriptors.

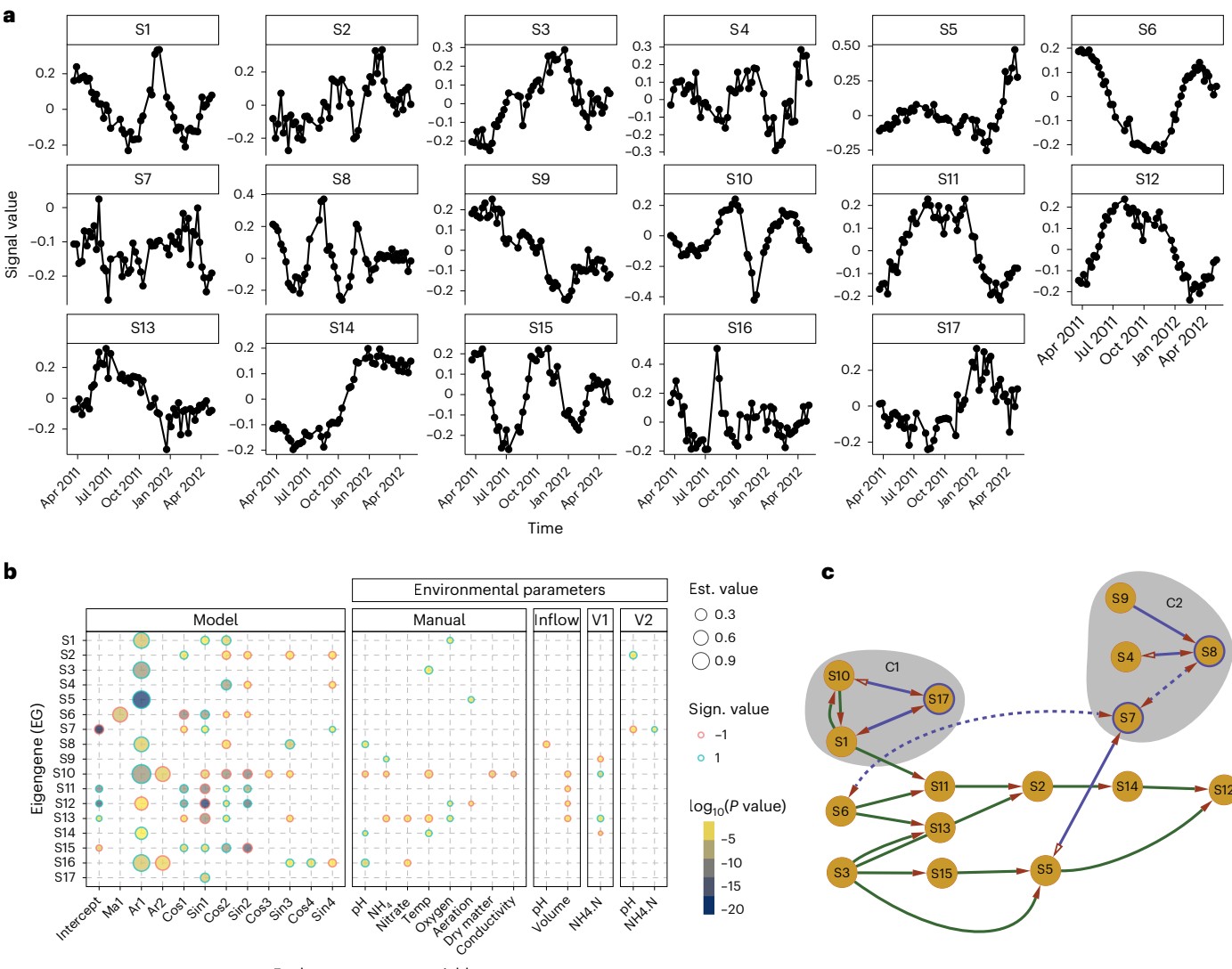

**Fig. 2 | Eigengene modelling using ARIMA augmented with environmental parameters and Fourier terms. a**, The signals S1–17 encapsulate the time-dependent dynamics underlying the microbial community. The scale of the y axis is dimensionless as the eigenvectors. **b**, The signals S1–17 are explained as ARIMA processes under the influence of the environmental variables. The five blocks of explanatory variables are: Model (ARIMA components), Manual (manually collected environmental variables, directly on the sampling location), Inflow (inflow stream of wastewater in the plant), V1 (first anaerobic tank in the plant) and V2 (second anaerobic tank in the plant). Every circle represents a significant

variable according to analysis of variance (Benjamini–Hochberg adjusted $P < 0.05$) for the corresponding signal among S1–17; the size represents the value of the coefficient, the ring colour its sign and the fill colour the $\log_{10}(P$ value). **c**, The signals are connected by a temporal transfer of information, suggesting a succession of ecological events. The signals with a purple edge are putatively nonlinear, and their relationships have been confirmed with convergent cross-mapping analysis. The dashed lines indicate weak transfer of information, while a full arrow and a hollow one represent an imbalance in information transfer (in favour of the solid arrow).

Hence, we computed one matrix per omic layer for the six formed taxonomic descriptors (phylum, class, order, family, genus and species) and two functional ones (KO terms and pathways) (Extended Data Fig. 2b). The resulting 27 matrices (3 original and 24 summary) were used to compute the system's eigengenes (EGs)[22]. In previous work, the first EG in a time series was shown to represent 'steady state' gene expression, encapsulating the largest explained variance (EV). Therefore, the first EG (average EV 50 ± 22% in all the datasets) was removed. We screened the subsequent EGs for time dependency (see Methods), selecting a set of 210 EGs, and assessed how much of the data variation they explained (Extended Data Fig. 2c).

To reduce potential redundancy associated with the time-resolved EGs identified across multiple data layers and bring together the same temporal behaviours, we clustered the set of 210 EGs into 17 representative EGs (see Methods). These are hereafter referred to as signals

(S1–17) and shown in Fig. 2a. We assumed that the 17 signals were not redundant because they were different enough to not cluster together. Each cluster contained multiple EGs with their associated EV (Extended Data Fig. 2d), and we associated the maximum EV of each cluster to its respective signal. In total, signals S1–17 accounted for 91.1% of the 'temporal' EV in the system (while the leftover 8.9% represented noise) and covered all temporal information in the training set.

The 17 representative signals (S1–17) were modelled using the environmental parameters as exogenous variables (after collinearity screening, see Methods and Extended Data Fig. 3) as shown in Fig. 2b. Moreover, the model includes predictors derived from the ARIMA, such as the intercept (the basal abundance/expression), autoregression (the time-lagged self-dependence) and sine/cosine (the cyclical behaviours, including seasonality), which explain the microbial process through its ARIMA components. In summary, we include self-dependent, cyclical

and environmental interactions to explain community dynamics. As seen in Fig. 2b, all the signals are generally explained more via the ARIMA components rather than the environmental ones. This is partially because some of the environmental variables also have a seasonal trend (for example, temperature) and their impact will be significant in the model if their values explain more than the seasonality (that is, having a fine-tuning effect). Therefore, the cyclical environmental patterns, such as temperature and water inflow, end up being factored into the cyclical part of the model, while only the residual effect is assessed by the properly named variable (for example, temperature). Moreover, it is interesting to note how only few of the environmental variables automatically collected by the BWWTP (variable blocks 'Inflow', 'V1' and 'V2') are significant to the model compared with the ones collected manually (Fig. 2b). This may be explained by heterogeneous spatial effects where the surface of the tank is a patchwork of neighbouring habitats with discrepancies in parameter values due to the viscosity of the foam. A similar microenvironment has been observed for flocks in BWWTP where nitrification was shown to happen in the outer 125 μm of the aggregates[33].

The large importance of a 'ground state' in BWWTP is linked to the need for robustness of a system that is operated primarily for public health purposes and that should be hardly perturbed during parameter-controlled operations. Furthermore, it has been shown in an activated sludge population, sampled monthly over 9 yr, that only one out of five microbiome clusters clearly oscillated with the seasons and reached a peak abundance of 22.3% in the community[18]. More possible temporal patterns are depicted in Extended Data Fig. 4.

## The ecological events in the microbial community

Even if the signals S1–17 are linearly independent from one another, we hypothesized that there might be some links through time among them. These links might coalesce the system into cliques of temporally concatenated ecological events that follow each other in an ordered sequence (similar to a domino effect). We therefore used the Granger causality test, which assesses the transfer of information across time between two series of observations, to generate a causal network for S1–17 ($P < 0.05$) with a maximum lag of 16 weeks. We also screened the signals for nonlinearity, and in case one of the nonlinear signals had a link, we verified it with a convergent cross-mapping analysis (see Methods and Extended Data Fig. 5). Incidentally, all signals except for S16 demonstrated a temporal relationship with at least one other signal, resulting in a single network of causality. We decided to focus on two particular cliques of nodes in the network (Fig. 2c) to explore the ecological domino effect: C1 (including S1, S10 and S17) and C2 (S9, S4, S7 and S8). To explore the ecological and environmental aspects of the system, we recalled the two-way relationship between the signals and the other eigengenes they clustered with. In doing so, we considered the generative processes and causal links of the signals which were applied to the 17 clusters. In this way, it was possible to use the top/bottom loadings of the EGs to link the high-level depiction of the system (the signals) to microbial community structure and function. The power of this representation is the amalgamation of the temporal signals, the loadings contributing to them (Extended Data Figs. 6 and 7) and the generative model provided by ARIMA (Fig. 2b) to generate ecological hypotheses that can be further tested. The analysis of the causal network should be considered as a tool to generate hypotheses on how the ecological events in the community have unfolded, utilizing a data-driven approach facilitated by the multilayered meta-omic angle of the study.

The first clique, C1, is composed of the two 'crash' signals, S1 and S10, which predict each other. Indeed, the peak/valley part of the signals, spanning autumn, has a similar shape but opposite sign, while the first part of the signals diverges with S10, showing a sinusoidal shoulder at the beginning. Both signals are strongly dependent on their previous state in time and have clear seasonal components (Fig. 2b). While S1 is

positively influenced by four variables including oxygen concentration as the sole environmental parameter, S10 is negatively impacted by a range of variables at the sampling site (pH, $NH_4$, temperature, dry matter and conductivity). *Podoviridae* and *Mimiviridae*, the two virus families identified in the system, are contributing positively and negatively, respectively, to S1 in the MG (Extended Data Fig. 6). Therefore, we infer two opposite viral mechanics involved in the fast valley-to-peak switch in autumn, which also corresponds to a major transient shift in community structure and substrate availability[27]. *Mimiviridae* target amoebas, which are known to prey on bacteria, indicating a possible multistep, interkingdom curbing process. In the case of the *Podoviridae*, it targets Proteobacteria and Firmicutes, which are highly abundant in the LAO (Fig. 1b). The other crash signal, S10, is characterized by the inverted reaction of the two most abundant bacterial families in the system: Microthrixaceae and Moraxellaceae (belonging to Phylum Proteobacteria). The family Moraxellaceae contributes positively to S1 in the MG, suggesting a takeover of the community, while the gene expression in members of the Microthrixaceae family is repressed (negative impact on S1, positive on S10) as shown in Extended Data Fig. 6. It seems plausible that the rise in Podoviridae would be linked to the rise of its putative host (Moraxellaceae), at the expense of Family Microthrixaceae. However, the decrease in Mimiviridae could have triggered an increase in amoebas, resulting in greater predation on the most abundant bacterial family. These events may subsequently drive S17, a signal solely explained by a cyclic ARIMA component (Fig. 2b), suggesting that the temporal behaviours in the systems cannot always be explained by long-term seasonal and environmental factors, but probably by the ecological interactions of the microbes involved. More specifically, S17 sees the rise in abundance or gene expression of three bacterial families: the fermenting Propionibacteriaceae, the polyphosphate-accumulating Intrasporangiaceae and the autotroph Gallionellaceae. These families point to the reaction of the foam community to the observed shift in autumn. Correspondingly, S17 represents the emergence of lipid-independent metabolic strategies. We also generated an ecological hypothesis for clique C2 and specifically addressed the temporal independence regarding presence and expression of pathways for fatty acid and triacylglycerol in the community (Extended Data Fig. 8). Both topics are discussed in Supplementary Information.

## Forecasting of future timepoints

From the analysis of the signals identified in the training datasets, it is already possible to identify five signal groups: (1) alternative basal states, for example, two alternative stable states of abundance/expression (S5, S14); (2) perturbation, that is, standing wave with varying amplitude and frequency (S4, S8, S15); (3) cyclical, that is, standing wave with constant amplitude and frequency (S6, S11, S12); (4) 'crashes', that is, quick shifts in the state and reversion to basal states (S1, S10, S16); and (5) mixed, that is, the other factors (Extended Data Fig. 4c–f). Alternative stable states, perturbations and crashes (groups 1, 2 and 4) are hard to model without observing multiple times the shift and the perturbation events, respectively. In addition, these scenarios may include permanent shifts into a new community equilibrium or transitory signals in the community that will be eventually resolved (for example, a viral infection). To forecast such events, experimental information (such as one derived from co-culturing) on microbial interactions would be required, which is beyond the scope of this study.

The 17 signals were used to train three models (with various parameters) from the package fable[29], and the best-performing model on the training set was selected for each of them (see Methods and Extended Data Fig. 9). In detail, ARIMA, Prophet and neural network models (with up to four Fourier terms for ARIMA and Prophet) were trained for S1–17 using the environmental variables as external regressors. The 51 weeks spanning the 2011–2012 data were used as a training set as well as to select the three best-scoring models to build a combined

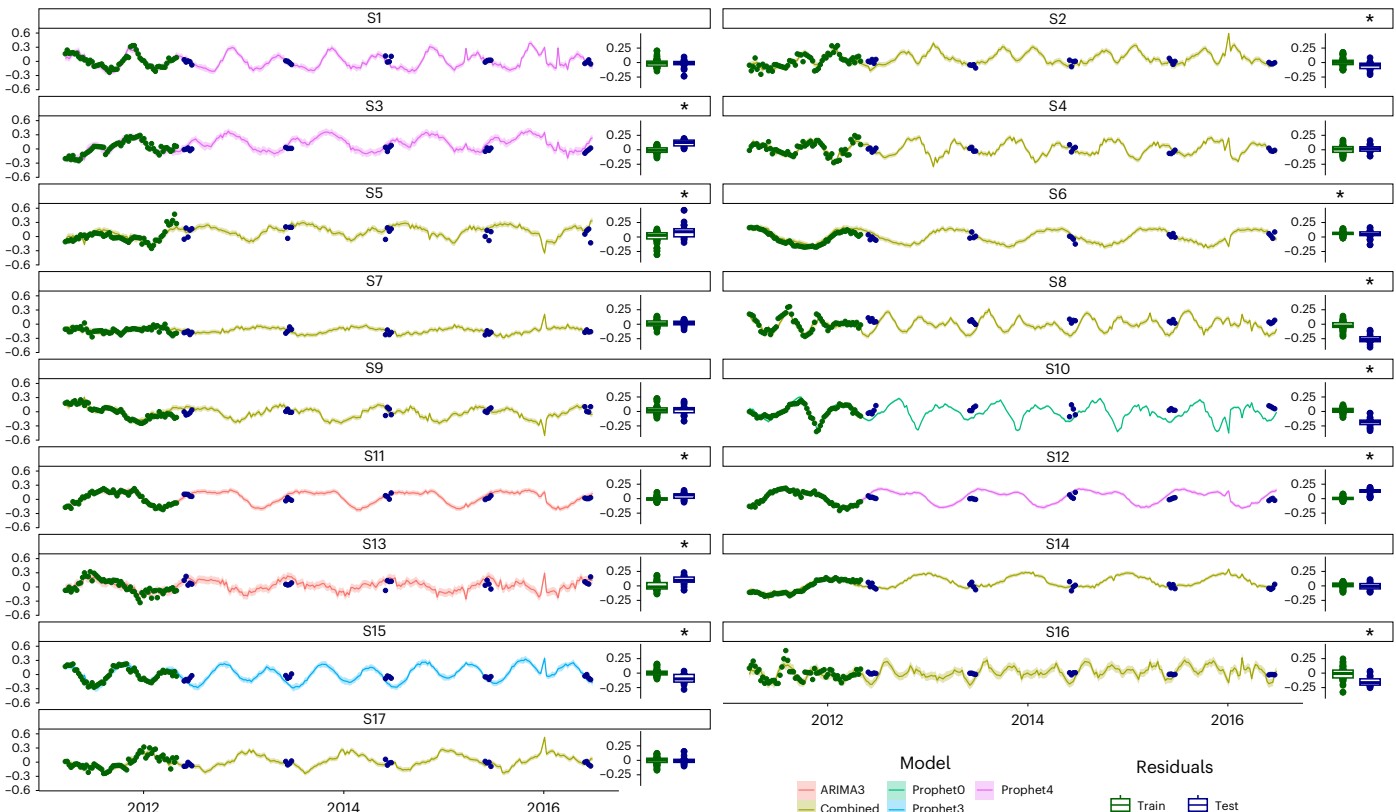

**Fig. 3 | Forecasting of the signals.** The 17 signals are predicted for the years 2011–2016 and compared with the data from June for those years. The green and blue dots represent the training and test data, respectively; the solid line depicts the median of the prediction, while the shaded area represents the 95% confidence interval. The green and blue boxplots on the right of every box depict the distribution of the model residuals from the training and the test sets, respectively. Corresponding scales are provided on the right y axes. The residue displacement from the null distribution was assessed using a Wilcoxon two-sided test (n = 21). The * on top of the boxplot indicates a statistical difference (Benjamini-Yekutieli corrected P < 0.01) between the mean of the residual distribution and 0, indicating incorrect/incomplete modelling (exact P values in Supplementary Table 7). In the boxplots, the central line indicates the second quartile, the lower and upper hinges correspond to the first and third quartiles and the whiskers extend from the hinge to the smallest/largest value no further than ±1.5 × the distance between the first and third quartiles. The samples beyond the range are plotted as individual outlier dots.

one (see Methods). In the end, the model with the smallest root mean square error (RMSE) was selected for forecasting. A total of 21 new samples were collected in the month of June of the subsequent 5 yr to validate the model for the MG and MT data. The month of June was chosen because it is far from disruptive events (such as rain, snow and very cold temperatures) that occur in autumn and winter. To predict the behaviours of the community in these cases, we would have needed a longer training set spanning multiple yearly cycles. To assess the accuracy of the forecasting, we computed the residues of the model and checked whether they were consistent with a white noise distribution. Therefore, we showed in 16 out of 17 cases that the modelling was sufficient to reproduce the training data (Fig. 3). There were six cases in which the modelling was fully successful: S1, S2, S4, S5, S10 and S16. The six correctly forecast signals account for 34.4% of the EV and 37.7% of the EV using the complete S1–17 model. However, the most common outcome of the validation was a good fit to the training set and an insufficient one in the testing (10 out of 17 cases), including signals from all the groups. This could be caused by two phenomena: overfitting of the model to the training set or its insufficient size. Of particular interest is S8, whose signal in the training set remains stable for several months including the end of the training set, probably indicating that the perturbation is over. S4 is strictly tied with S8 (Fig. 2b); however, S4 was modelled and predicted correctly, suggesting a new cycle being established rather than a perturbation setting in. It is difficult to put these results in perspective due to the lack of similar studies covering a similar period and sampling frequency. However, a previous study[18]

that sampled the same BWWTP monthly for 9 yr showed that while five microbial clusters formed the main community, only one of them presented a clear yearly oscillating pattern. The same cluster was present in the BWWTP even after a bleaching event; therefore, it is reasonable to assume that a fraction of the LAO community had a similar cluster and that the signal(s) underlying it continued in the subsequent years.

Unexpectedly, the correct forecasting of S1, which looked like a crash (Extended Data Fig. 4f) and was linked (among other things) to viral increase/decrease, suggests that it is indeed a cycle. We speculate that a recurrent triangular interaction between viruses, amoebas and bacteria might be repeated over time and lead to S1. The integrated meta-omics data should be supported in the future by complementary techniques such as microscopy and co-culturing to confirm this hypothesis. Unfortunately, an analogous trend seen for signal S10 was not equally well represented. Similar to S1, S16 also exhibited a behaviour expected from a system crash. However, the forecasting hinted at a cyclical occurrence; hence, what appeared like a crash is predicted to be a constitutive and repeated behaviour. Another similarity with S1 is that viral families impacted S16, that is, *Mimiviridae* (positively and negatively in the MG) and *Podoviridae* (positively in the MP). Signal S5 showed a sharp upward movement in relation to the general trend before starting to dip towards the end of the time series. Well-known bacteria involved in bulking, such as Moraxellaceae and Gordoniaceae, have loadings contributing towards S5, hinting to a quick jolt in thickening of the foam in summer and an overall cyclical effect that can be forecast over time.

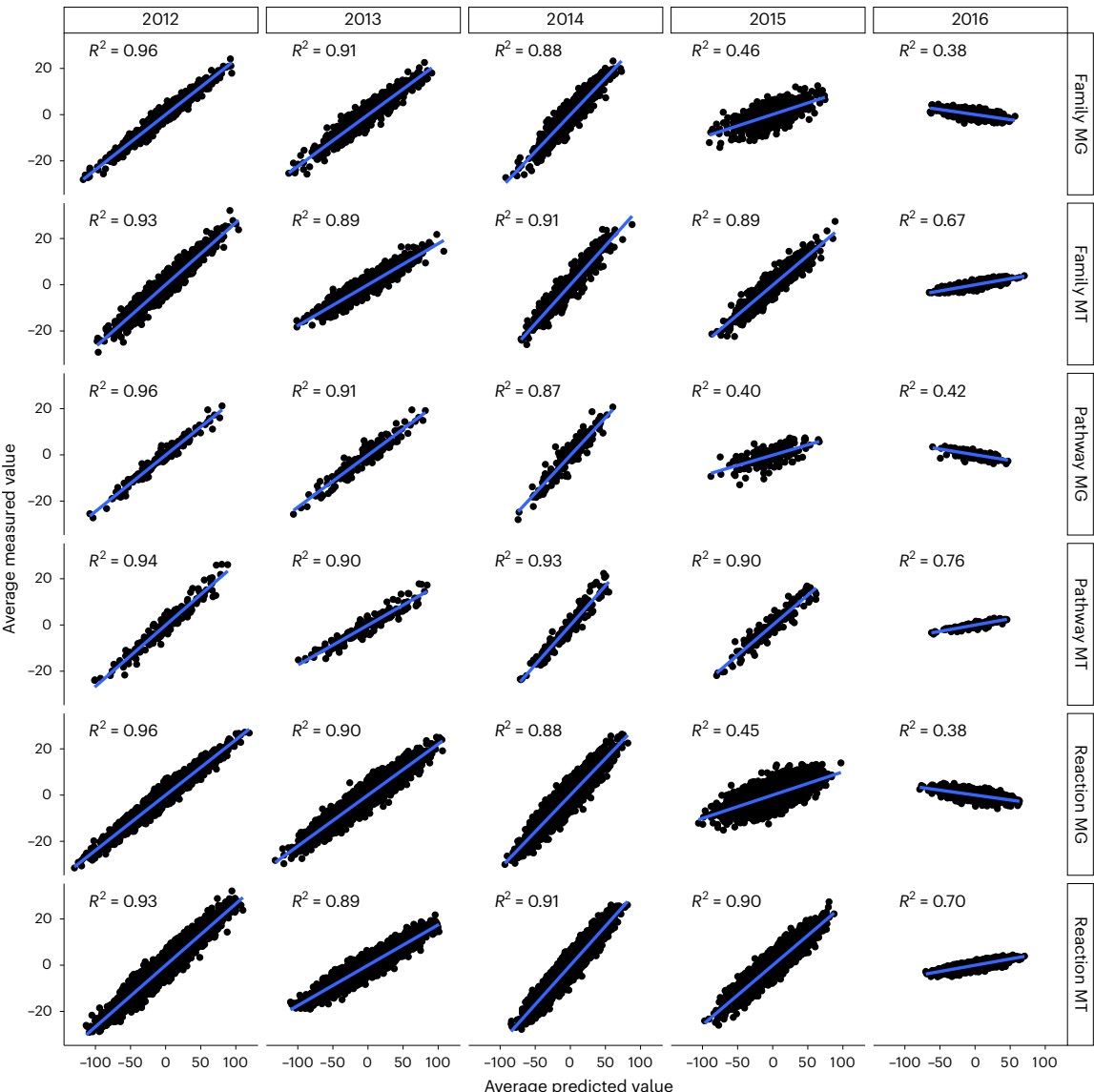

**Fig. 4 | Reconstruction of the June months 2012–2016.** The test samples were reconstructed using the 17 signals and their weights estimated through linear regressions on the training set. The reconstructed matrices are based on MG and MT data summarizing taxonomic families, reactions and pathways. The coefficient of determination $R^2$ (computed as the squared Pearson correlation coefficient) is reported for each panel, with a higher coefficient demonstrating a more accurate prediction.

## Forecasting gene abundance and expression

Following the forecasting of the signals, we decided to try to reconstruct the samples taken from the subsequent years. The samples' information content can be expressed as a linear combination of the signals by creating a linear model using the training set and where the signals are the predictors. Using this approach, we computed how much each of the signals contributed to the samples (that is, finding the betas of the model) and the basal abundance/expression (the intercept) of the samples. We decided to validate the approach using the gene abundance and expression values of the microbial families and reactions (KO term groups). Therefore, we fitted the linear models to those matrices from the training set and combined the results with the previously forecast signals to reconstruct the test matrices. We then compared the reconstructed values with the original ones for each individual sample (Extended Data Fig. 10). The comparisons showed a range of results, including samples that were predicted correctly (data points arranged in a narrow diagonal line), samples with poor predictions (unordered distribution of the data points) and samples with

an unexpected inverse relationship with the prediction (descending diagonal line). When taking into account the explanatory variables in the ARIMA modelling, we already hypothesized a micro-environmental effect at play in the foam, making it a composition of areas with (slightly) different environmental values. We now extend that hypothesis to the sampling unit itself (the foam 'islet', see Methods), which might have individual genetic potential and gene expression characteristics imputable to the process of foam formation, permanence and stability. We therefore assume that islet variability, compounded by the temporal evolution of the system, ultimately has an impact on the sample. Intuitively, if the foam islets were composed of the same genetic makeup but subject to (even small) different environmental conditions, one would expect gene abundances to be relatively stable, yet gene expression might change. Instead, observing the coherent response between MG and MT to the reconstructed samples from Extended Data Fig. 10, it is apparent that the genetic makeup of the islets changes from week to week and gene expression changes accordingly with this alteration. We assume that our modelling creates a 'smoother' representation of the

data, necessarily averaging the observed sample to sample variability. This can be imputable to the SVD step of the modelling, which isolates 'high-level' patterns that harbour lower noise than any individual ORF- or descriptor-based summarization of the data. Moreover, the scale of the values is often larger in the reconstructed samples than in the test ones (Extended Data Fig. 10).

To counter the islet variability, we considered the average of the measured and predicted values over the month of June for each year and computed the coefficient of variation, $R^2$, for each of them (Fig. 2). The $R^2$ is strikingly high ($\geq$0.87) in all the six matrices for the subsequent 3 yr after the training set, but the predictability starts decreasing from the fourth year after the training samples. This implies that in our system (LAO), the observation through meta-omics data and the environmental parameters for 14 months is sufficient to build a reliable predictive model. Moreover, with this model and the monitoring of the environmental parameters, it is possible to correctly chart the community structure and function at any given point within the subsequent 3 yr after the training set.

## Conclusions

We present the temporal reconstruction of the surface microbial community of a BWWTP over 1.5 yr of weekly sampling. The gene abundance and expression show 17 distinct and linearly independent signals (S1–17) across time (Fig. 2a), many of which were explained by the physicochemical parameters and the mathematical components describing self-dependence and seasonality (Fig. 2c). The signals were tied in a 'temporal domino' (Fig. 2b), from which we selected two cliques to successfully describe the 'autumn crash' (C1) and an oscillatory perturbation (C2, see Supplementary Information). The models built on the S1–17 signals and paired with the environmental parameters were subsequently used to forecast the next 5 yr of the LAO community. We demonstrate that six of the forecast signals (S1, S4, S7, S9, S14 and S17) are indeed validated by the future samples (Fig. 3) and cover some interesting aspects of the BWWTP surface community, such as nitrogen metabolism (S4 and S9) and viral interplay (S1 and possibly S7), as well as changes in foam-related metabolism (S17). Importantly, when rebuilding the gene abundance and expression data at the levels of taxonomic families, reactions and pathways and extrapolating to the future samples (June 2012–2016), the results over the averaged month of June showed a very high degree of predictability for the subsequent 3 yr after the training set ($R^2 \geq 0.87$). However, a clear fading was apparent starting from the third year (Fig. 4).

Overall, the present approach covers most of the time-dependent information in the system. It furthermore enables us to describe a complex community with its behaviour in a number of temporal patterns, which is easy for a human to interpret (in our case, 17 signals), and link these to their underlying generative processes, as well as the environmental parameters, taxa and functions supported by them. Furthermore, the method allows reliable forecasting of these fundamental signals that represent a seasonality and temporal span (>1 yr, hence more than one expected full cycle of the system), indicating that the time- and environment-dependent components can explain the community during regular BWWTP operations. We hope that further work, especially sampling the BWWTP at higher time frequencies (for example, hours) and/or for longer periods (multi-annual training sets), could be integrated for a more detailed systemic description and increased forecasting ability to cover those phenomena poorly constrained by the current model. Finally, we infer that there are environmental drivers in the macroscopic composition of the LAO community behaviour and that we can correctly reconstruct the samples from 3 yr into the future when averaged over a 1-month period. However, we also infer that the community exhibits a high degree of variation, making the prediction of a specific sample inaccurate with this method. The current work forecasts a BWWTP during its normal operations, and it could be exploited to predict population and gene expression levels in

the temporal medium range when knowing the environmental parameters. However, a potentially interesting development would be to test what happens when introducing 'critical' values of the environmental parameters in the model to simulate an environmental disturbance. To use this approach, important details about the experimental design should be considered. The chosen (micro)biological system should be sampled at time intervals that are relevant for the research question (for example, cell doubling times if one wants to study microbial community composition dynamics) and spanning multiple time cycles.

## Methods

### Sampling and preprocessing

Floating LAO biomass was sampled from the air–water interface of the anoxic activated sludge tank at the Schifflange wastewater treatment plant (Esch-sur-Alzette, Luxembourg; 49° 30′ 48.29″ N; 6° 1′ 4.53″ E) in the form of a single islet (examples illustrated in Fig. 2 of ref. 30). The sampling frequency—weekly—was chosen as it is the generation time of the activated sludge in the BWWTP (the average time it remains in the system) and the average doubling time of the dominant *Microthrix* population[32]. For each sampling date, indicated as dates in the format YYYY-MM-DD, one entire 'islet' was sampled using a levy cane of 500 ml. Samples were quickly homogenized and collected in 50 ml sterile Falcon tubes and then immediately flash-frozen by immersion in liquid nitrogen and stored at −80 °C to guarantee optimal sample integrity and quality.

For the 51 timepoints of the training set (21 March 2011 to 3 May 2012), samples were treated in 2012 as previously described[27]: 200 mg was subsampled from the collected islet using a sterile metal spatula, at all times guaranteeing that the samples remained in the frozen state, and used for subsequent biomolecular extraction according to a previously published procedure (using the Qiagen AllPrep DNA/RNA/protein mini kit-based method on 'LAO-enriched mixed microbial community'[26]).

Additional concomitant biomolecular extractions were applied to a total of 21 samples collected during the month of June from 2012 to 2016 and extracted in a separate experiment in 2018. The sample preprocessing protocol was carried out on a customized robotic system owned by the lab (Beckman-Coulter_Platform Biomek 4000 NXP Span8 Gripper) following the same protocol as for the training set sample extraction described above with few differences. The biomolecular extraction was then performed using the commercial AllPrep DNA/RNA/protein mini kit (Qiagen, 80004), conducted on a customized robotic system owned by the lab (Tecan-LU_UNILU_EWS_EXTRACTION_EU-0908-Freedom EVO 200). An RNase treatment followed by DNA precipitation was carried out on the DNA, and the RNA was purified using the commercial kit Zymo RNA Clean & Concentrator-5 (R1013). RNA quality was assessed as in the previous study for the same environment[30].

### High-throughput meta-omics

DNA (400 ng) was sheared using NGS Bioruptor (Diogenode, UCD300) with 30 s ON and 30 s OFF for 10 cycles. DNA libraries were prepared using TruSeq Nano DNA kit (Illumina, FC-121-4002) employing standard protocol with 8 PCR cycles. The libraries were prepared for a 350 bp average insert size. RNA (1 μg) was depleted of ribosomal RNA using the RiboZero kit (Illumina, MRZB12424). Ribosomal RNA-depleted samples were further processed and prepared using the TruSeq Stranded mRNA library preparation kit (Illumina, RS-122-2101). The fragmentation time was reduced to 3 min. The samples were amplified for 8 PCR cycles. The prepared libraries were quantified using Qubit 4 (Thermo Fisher) and quality checked using Bioanalyzer 2100 (Agilent). Sequencing was performed on a NexSeq 500 instrument using 2 × 150 bp read length at the LCSB sequencing platform (RRID SCR_021931).

### Collection of environmental variables

The environmental variables were collected on site by the researcher(s) while they were performing the sampling. These include dry matter,

phosphate, nitrate, ammonium, oxygen, conductivity, pH, temperature and oxygen (Supplementary Table 5), following previously established protocol[27]. The other variables were retrieved from the automated data collection routine of the Schifflange BWWTP, which measures these values online and aggregates them as 2 h averages starting at 1:00. These recordings include the same variables for different parts of the plant (inflow, both vats, outflow) with the addition of other measurements such as the in/outflow volume. For simplicity, we used exclusively the variable pertaining to the inflow, both vats and outflow in this study (Supplementary Table 6). The Schifflange plant is depicted in https://sivec.lu/installation/station-depuration/, with the various components named in German. The variables were screened for collinearity (Extended Data Fig. 3) using the Pearson correlation coefficient to allow a rational selection, resulting in 15 variables used from the 59 initial ones. The variables Oxygen_manual, Dry_matter, NH4.N, Vat1_NH4.N and Vat2_NH4.N were transformed using the square root function.

### Co-assembly of metagenomics and metatranscriptomics reads

All the samples from the training and the test datasets followed the same bioinformatic pipeline. Sample-wise preprocessing of the MG and MT data was performed using IMP (v.3.0)[34] (https://git-r3lab.uni.lu/IMP/imp3) with custom parameters, that is, (1) Illumina Truseq2 adapters were trimmed, and (2) the step involving the filtering of reads of human origin step was omitted for the preprocessing. The reads were corrected using BayesHammer[35] per sample, per omic. The resulting MG and MT reads were assembled with metaSPAdes (v.3.13.1)[36] and rnaSPAdes (v.3.13.1)[37], respectively. The MG and MT reads of each sample were re-assembled together using the contigs and 'highly filtered' transcripts from the first assemblies as trusted contigs.

### Contig sorting into biological subsets

Contigs longer than 1,000 nt from each sample were retained and sorted into four subsets: eukaryotes, plasmids, viruses and chromosomal prokaryotes. First, the contigs were screened for eukaryotes using EukRep (v.0.6.7)[38]; the resulting non-eukaryotic contigs were searched for plasmidial sequences with Plasflow (v.1.1.0)[39] and cbar (v.1.2)[40] as well as for viral sequences using virsorter (v.1.0.6, categories 1 and 2)[41] and deepvirfinder (v.1.0)[42]. A contig was considered viral or plasmidial if both tools agreed in the prediction; all leftover sequences were considered chromosomal prokaryotic. Later, some contigs of the latter group were moved to the eukaryotic (see 'Taxonomic and functional annotation' section).

### Binning and clustering

The chromosomal prokaryotic subsets of each sample were binned using IMP (v.3.0)[34] with MaxBin[43], MetaBAT[44] and binny[45] plus a refinement step with DAS Tool[46]. The resulting bins were dereplicated along the entire time series with dRep (v.0.5.4)[47] to create rMAGs on the basis of the results of CheckM (v.1.0.7)[48], such as contamination and completeness (results for the rMAGs are shown in Fig. 1a). Similarly, the eukaryotic subsets were binned with MetaBat[44] and dereplicated using dRep (v.0.5.4)[47] without genome quality assessment resulting in rMAGs. All the plasmidial, viral and the unbinned contigs from the eukaryotic and chromosomal prokaryotic subsets were clustered using CD-HIT (v.4.6.8)[49] on each of those subsets. We refer to the subset of the clustered unbinned contigs as rContigs. The collection of the rMAGs and the rContigs constitutes the representative database (rDB) of the system.

### Taxonomic and functional annotation

The rMAGs and the rContigs were annotated taxonomically using the Contig Annotation Tool and Bin Annotation Tool (v.5.1.2)[50], respectively. The ORFs were predicted from the rDB using AUGUSTUS (c3.3.3)[51] for the eukaryotic set and IMP (v.3.0)[34] for all the other sets. The ORFs were annotated using Mantis (v.1.02)[52] with the heuristic approach

and using kofam[53], tigrfam[54], EGGNOG[55], Pfam-A[56] and NCBIG[57]. Subsequently, only the entries with KO terms assigned by kofam were retained for analysis.

### MG and MT quantification and filtering

The filtered MG and MT reads were aligned to the ORF reference set using bwa[58] and sorted using samtools (v.1.11)[59]. The resulting sorted bam files were processed using bam2hits (v.1.0.9)[60] and the output split with a maximum number of 100,000 ORFs per subset while respecting the bam2hits read groups. Each subset was quantified with mmseq (v.1.0.9)[60] and mmcollapse[61], then the quantifications per sample were the normalized form of fragments per kilobase million, merged and re-normalized to fragments per kilobase million. Values of gene abundance and expression inferior to $10^{-7}$ were considered equal to 0, and ORFs and transcripts that were not present in at least 20% of the training set were discarded from further analysis.

### MP quantification and filtering

Raw MP data were retrieved from the PRIDE repository with accession number PXD013655 (ref. 27); samples were processed as previously described[32] and re-analysed. The complete set of predicted ORFs was subsetted to obtain smaller sample-specific databases. The MG alignment files generated in the previous step were processed with featurecounts[62], and all the ORFs with a count greater than 0 for the given sample were included in the appropriate sample. Each sample-specific database was concatenated with a cRAP database of contaminants (https://thegpm.org/cRAP; downloaded in July 2019) and the human UniProtKB Reference Proteome (UniProt Consortium, 2021), and decoys were generated by adding the reversed sequences of all protein entries to the databases for the estimation of false discovery rates. The search was performed using SearchGUI (v.3.3.20)[63] with X!Tandem[64], MS-GF+[65] and Comet[66] as search engines and the following parameters: trypsin was used as the digestion enzyme, and a maximum of two missed cleavages was allowed. The tolerance levels for matching to the database were 10 ppm for MS1 and 15 ppm for MS2. Carbamidomethylation of cysteine residues and oxidation of methionines were set as fixed and variable modifications, respectively. Peptides with length between 7 and 60 amino acids and with a charge state between +2 and +4 were considered for identification. The results from SearchGUI were merged using PeptideShaker-1.16.45 (ref. 67), and all identifications were filtered to achieve a protein false discovery rate of 1%. The sample-specific peptide-spectrum matches obtained for each analysis were then used to calculate dataset-wide protein groups using the Occam subgroup method from the Pout2Prot algorithm[68]. The dataset-wide protein group output was then submitted to Prophane[69] with default parameters to retrieve the quantitative values using normalized spectral abundance factor. Values of protein abundance inferior to $10^{-3}$ were considered equal to 0, and only proteins present in at least 20% of the training samples were retained for further analysis.

### Batch effect correction

The whole data analysis was conducted in R 3.4.4. First, we transformed the MG, MT and MP data using the central log ratio with the function 'clr'[70] to overcome the inherent problems of compositional data[71,72]. To estimate the batch effect between the train and test samples introduced by the different experimental procedures (mainly the robotic biomolecular extraction in the test samples and the read length), we regressed every entry in the MG and MT matrices with a linear model (with the function 'lm') as:

$$Y = \alpha + \beta_E X_E + \beta_T X_T + \varepsilon \qquad (1)$$

where $Y$ is the central log ratio (clr) transformed quantification matrix; $\alpha$ is the intercept of the model; $X_E$ and $X_T$ are the environmental and

technical variables (number of reads, average length of reads), respectively; $\beta_E$ and $\beta_T$ are the vectors of the environmental and technical coefficients, respectively; and $\varepsilon$ is the randomly distributed Gaussian error $N(0, \sigma^2)$. The non-normality of $\beta_T$ was assessed with the Shapiro test[73] (function 'shapiro.test'), sampling 10 times 5,000 ORFs at random per technical variable for the MG and MT matrices and computing the scores in Supplementary Tables 3 and 4, respectively. Therefore, we corrected the quantification matrices as:

$$Y' = Y - \beta_T X_T, \qquad (2)$$

subtracting the estimated batch effect from the quantification matrices. The distributions of $\beta_T$ are shown in Extended Data Fig. 2a.

### Eigengenes and their analysis

The EGs for the training set (samples from 21 March 2011 to 3 May 2012) were computed as singular right eigenvectors obtained with the function 'svd'. The data were normalized according to the basal expression[22] computing the quantification matrices as:

$$Y = U\Sigma V^T \qquad (3)$$

where the first element of the eigenvalues vector $\Sigma$ has been replaced by 0. The EGs were recomputed from the normalized matrices and subsequently tested using the Ljung–Box test ('Box.test'), the augmented Dickey–Fuller test ('adf.test') and two Kwiatkowski–Phillips–Schmidt–Shin ('kpss.tests') tests with null hypotheses 'trend' and 'level', respectively. If at least two of the four tests were passed ($P < 0.05$ for Ljung–Box and Kwiatkowski–Phillips–Schmidt–Shin tests; $P > 0.05$ for Dickey–Fuller test) the EG was considered time-dependent. The $i^{th}$ EG was modelled using seasonal ARIMA modelling (where the subtraction of the seasonal effects on the data was not required beforehand). The ARIMA model is described by three non-seasonal parameters: $P$ (autoregressive terms), $d$ (number of integrations for differencing) and $q$ (moving average terms). Considering that the training set did not span two cycles (the hypothetical period of seasonal patterns), we added up to four Fourier transform terms to the model as a proxy for the seasonal component. The Fourier transform can identify in a series of data the sum of sine and cosine waves underlying the data. In this way, if the period of time is correct (in this case, 1 yr), the Fourier terms can explicitly provide the seasonal part of the temporal behaviour. Using multiple terms allows for complex seasonal effects, while limiting the maximum number to 4 prevents overfitting of the data. For this, we used the 'arima' function of the package fable (v.0.3.1)[29] as:

$$\mathbf{EG}_i = \text{ARIMA}(X + F(K = \{0 - 4\})) \qquad (4)$$

where $X$ is the matrix of the environmental variables, and the Fourier term includes a number of sine and cosine components $K$, ranging from 0 to 4. The value of $K$ therefore spans from no seasonal effect ($K = 0$) to increasingly complex ones. The best model of the five was selected according to their $R^2$ values. The best model thus provided the weights for the environmental variables ($X$) for the parameters $P$, $d$ and $q$ and for as many sine and cosine terms as the selected $K$ parameter. We called the ensemble of those variables the 'explanatory variables', and we assessed their significance using analysis of variance ('anova' function).

### Eigengenes clustering and Granger causality network

Considering that we required a clustering approach that is independent of scale, we computed the Pearson correlations between pairs of EGs, the output was made absolute and the Minkowski distance was computed. The clusters were retrieved using the 'cutreeDynamic' function (deepSplit=0, pamRespectsDendro=FALSE, minClusterSize=3) from the dynamicTreeCut package[74] (because it can accommodate a complex structuring of the data), resulting in 17 groups

(Extended Data Fig. 2d). From each of the 17 groups, a representative EG was selected according to the following criteria: (1) MG or MT (because MP data do not exist beyond the training set) and (2) smoothest profile (minimal median of the absolute de-trended time series). The resulting EGs are S1–17 in Fig. 2a.

The signals were tested two at a time with the Granger causality test (grangertest) from the lmtest package (v.0.9-38)[75], and if $P < 0.05$, the two signals were considered connected. The signals were screened for nonlinearity via empirical dynamic modelling as implemented in the R package rEDM (v.1.14.0)[76]. We first identified the best number of lags (embedding value) to analyse the signals using the simplex function and default parameters. The signals were screened with the S-map method[77], and only three signals appeared to be putatively nonlinear: S7, S8 and S17. All the causal links identified with the Granger causality test were also tested with the convergent cross-mapping method (Extended Data Fig. 5) using the function ccm with library size=c(20,50,1) and default parameters. If one of the signals connected with the Granger test was one of the putatively nonlinear ones, we verified the link using convergent cross-mapping, again from the rEDM package[76]. Visualization of the network was performed with Cytoscape[78] while manually adjusting the edges and directionality arrows to add the empirical dynamic modelling to the Granger causality results.

### Modelling the signals and model selection

For each signal, we trained multiple models using three techniques (ARIMA, Prophet and neural network using the functions ARIMA, Prophet and NNETAR, respectively, all implemented in the R package fable[29]), alongside a range of values for the parameters accounting for seasonal components. Each signal was modelled as a separate process whereby the signal itself was the target of the model and the environmental parameters were the only exogenous variables. Therefore, we did not use any information transfer among the signals in the modelling.

We fitted ARIMA with up to four Fourier components (see 'Eigengenes and their analysis' section), while the parameters $P$, $d$ and $q$ were automatically optimized by the function, leading to five ARIMA models (one for each increment of Fourier transform terms, starting with 0). For the Prophet modelling, we specified seasonality (period = 'year', type = 'additive' and order = from 0 to 4, analogously to the Fourier transform terms of the ARIMA) and growth (type = 'logistic'), resulting in five Prophet models. The neural network function was used whereby the number of nodes in the hidden layer was set to 10, 20 and 30. The 13 models for each signal were scored according to their RMSE, and the 3 models with the lowest RMSE were combined (weighted by 1 − RMSE), as a 14th ensemble model. The RMSE was calculated for the 14 models as well (Extended Data Fig. 9). For each signal, the model with the lowest RMSE was selected for the putative generative process and used to forecast the test set with the function 'forecast' of the fable package and supplying environmental parameter readings.

### Forecasting the signals and reconstruction of future samples

The 17 signals were forecast ('forecast' function of the fable package[29]) for the 5 yr following the training set, using the fitted models and the environmental variables (recorded in the forecasting period) as exogenous variables. The forecast signals were therefore used to 'reconstruct' the information in the future samples, that is, predict the actual gene abundance (MG) and expression value (MT) matrices associated with the test samples. This was possible because the matrices used to summarize the LAO community can be expressed using a linear combination of the 17 signals plus a basal gene abundance/expression (that we previously removed in the analysis). We therefore decided to 'reconstruct' June 2012–2016 matrices for the reaction, pathway and family summarization of gene abundances (MG) and expression values (MT). We ran linear regression ('lm' function) using the six training set

matrices for the categories above as target variables and the 17 signals as explanatory variables. We then 'reconstructed' the test matrices using a linear combination of the forecast signals over the test set, weighted by the betas and offset by the intercept (basal level) derived from the linear model of the training set while also adding the intercept (basal level). The reconstructed and the real samples were compared on an individual basis (Extended Data Fig. 10) and on a month-averaged basis (Fig. 4).

### Reporting summary

Further information on research design is available in the Nature Portfolio Reporting Summary linked to this article.

### Data availability

The generated MG and MT reads (FASTQ) files, as well as the previously produced data, are available as NCBI BioProject PRJNA230567. The MP data are available from the PRIDE repository, with accession number PXD013655 (ref. 27). Source data are provided with this paper.

### Code availability

The meta-omics pipeline IMP (v.3.0)[34] is maintained and developed at the GitLab page: https://git-r3lab.uni.lu/IMP/imp3. The code used in the analysis is available at https://git-r3lab.uni.lu/ESB/lao/lao_ts and https://github.com/fdelogu/microforecast, while the data required to start the analysis are available on Zenodo at https://doi.org/10.5281/zenodo.7225349. The full list of software and R package versions are listed in the Git pages.

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

## Acknowledgements

We thank the Luxembourg National Research Fund (FNR) for supporting this work through various funding instruments. Specifically, a PRIDE doctoral training unit grant (PRIDE/15/10907093), CORE grants (CORE/17/SM/11689322), a European Union ERASysAPP grant (INTER/SYSAPP/14/05) and an ATTRACT grant (A09/03) all awarded to P.W., as well as a CORE Junior (C15/SR/10404839) grant to E.E.L.M. The project received financial support from the Integrated Biobank of Luxembourg with funds from the Luxembourg Ministry of Higher Education and Research. This work was also supported by the European Research Council (ERC) under the European Union's Horizon 2020 research and innovation programme (Grant Agreement No. 863664). The work of P.M. was funded by the 'Plan Technologies de la Santé du Gouvernement du Grand-Duché de Luxembourg' through the Luxembourg Centre for Systems Biomedicine (LCSB), University of Luxembourg. S.W. was supported by the Austrian Science Fund (FWF) Elise Richter V585-B31. P.B.P. is grateful for the support from the Research Council of Norway (FRIPRO programme: 250479) and the Novo Nordisk Foundation (Project No. 0054575). The authors acknowledge the ULHPC for providing and maintaining the computing resources. We duly thank Mr G. Bissen and Mr G. Di Pentima of the Syndicat Intercommunal a Vocation Ecologique (SIVEC) for access to the Schifflange wastewater treatment plant.

## Author contributions

F.D. and P.W. contributed to the planning and designing of the overall study and analyses. F.D. performed the data analyses. P.M.Q. contributed the protein annotation software. B.J.K. performed the MP measurement. E.E.L.M. and L.A.L. collected and performed the biomolecular extractions on the samples. R.H. performed the DNA and RNA sequencing. F.D., P.B.P., P.M., S.W., E.E.L.M. and P.W. participated in discussions related to this work. F.D., P.M., S.W., E.E.L.M. and P.W. wrote and revised the manuscript. All authors read and approved the final manuscript.

## Competing interests

The authors declare no competing interests.

## Additional information

**Extended data** is available for this paper at https://doi.org/10.1038/s41559-023-02241-3.

**Correspondence and requests for materials** should be addressed to Francesco Delogu or Paul Wilmes.

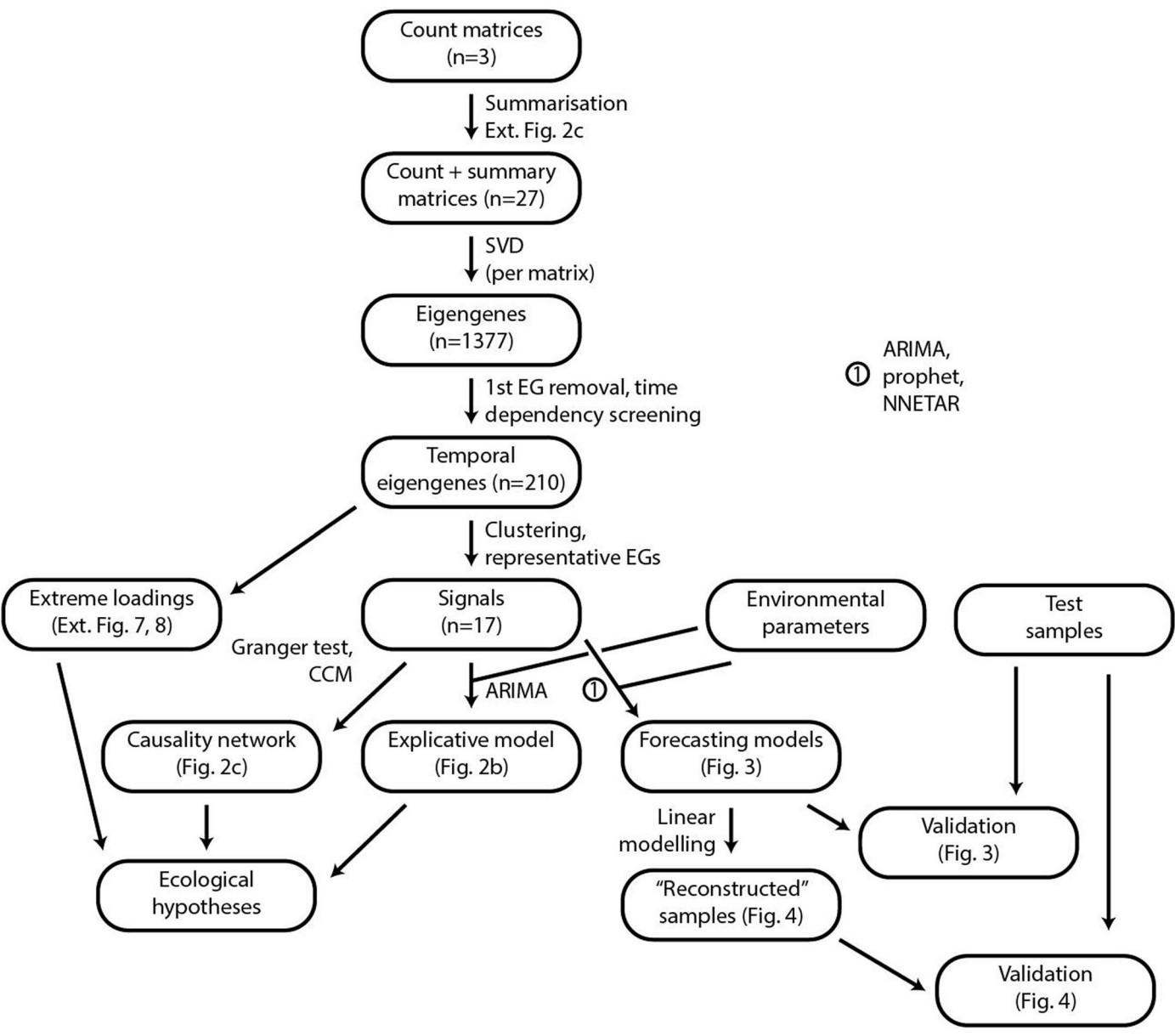

**Extended Data Fig. 1 | Workflow of the analysis.** Each box represents a piece of data in the analysis while the arrows show their relationships. When necessary the type of action to move from a box to the next is reported on the side of the arrow. When available the reference figures are indicated in the workflow. The analysis starts with the count matrices for MG, MT and MP and ends with the ecological hypotheses and the validation of the forecasting and the future sample reconstruction.

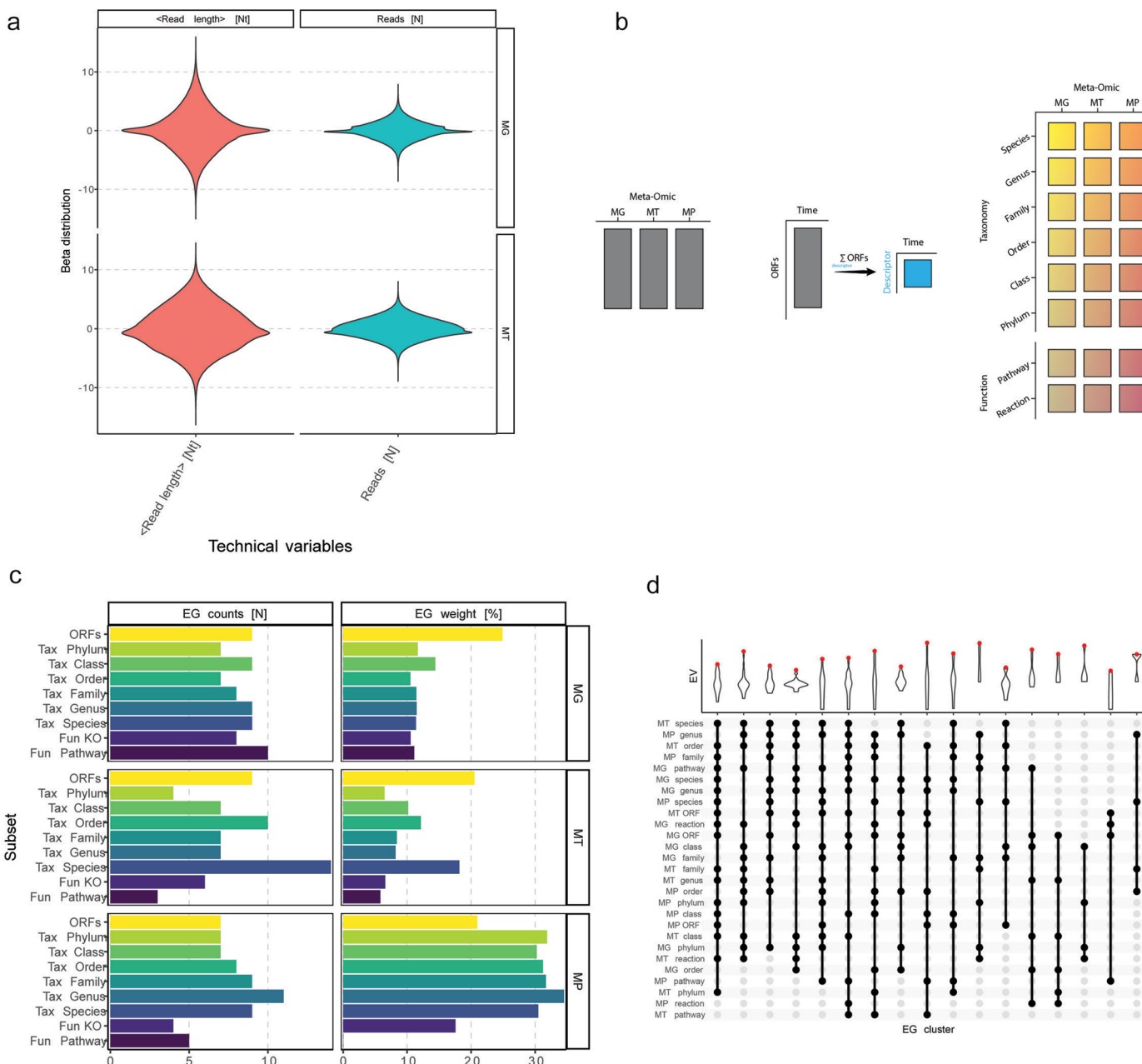

**Extended Data Fig. 2 | Technical overview.** a. Technical effect estimation. The data were regressed with the experimental variables (that is environmental parameters) and the technical ones (that is read length and number of reads). The plot shows the distribution of the betas resulting from the regression for the MG and MT ORF-based matrices. b. The three ORF-based omic quantification matrices are summarised by summing up the lines with the same ORF descriptor. The final result is a collection of 24 matrices + the original three. c. The six panels show the number of time-dependent EGs and the EG weights (equivalent to the Explained Variance) per omic in the nine summarisation matrices. The first EG (that is the basal state of the system) was removed and all the EG weights re-scaled per matrix. In the y axis 'Fun' stands for 'Function' and 'Tax' for 'Taxonomy'.

The number of selected EGs changes depending on the omic and the descriptor, however some trends can be seen in the EG weight. For MG and MT the EG weight is the largest, signifying that it is, if taken alone, the most informative layer of information. Interestingly in MT the second largest, with a decent margin, is the Species level, which can be explained as a level in which most of the individual genes information is conserved (that is genes of the same species will be expressed together over time). d. EG clustering. The columns represent the 17 EG clusters while rows indicate the different types of summarisation matrices. In the top panel the violin plots depict the distribution of the explained variance (EV) from the EGs in the cluster. The red dot indicates the maximal EV in the distribution and the EV of the cluster. On the y-axis there are the 27 matrices.

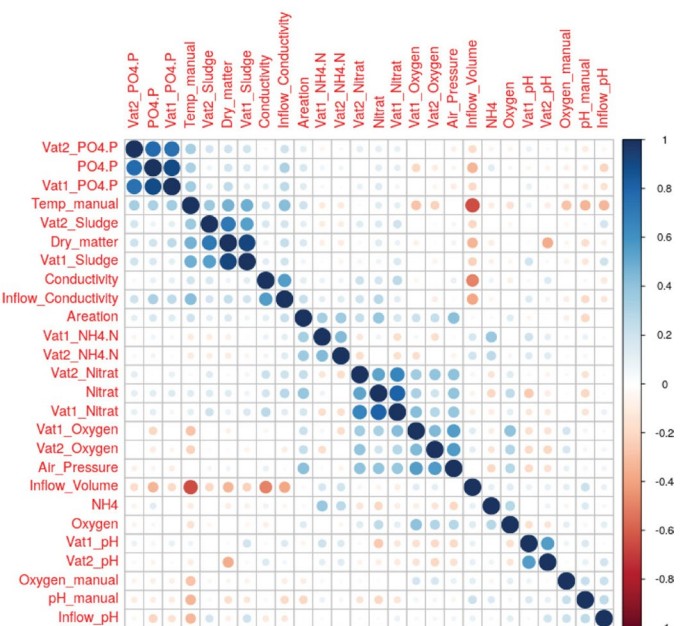

**Extended Data Fig. 3 | Correlation of the environmental variables.** Corr-corr plot of the correlations between the selected starting environmental variables to explain the signals. From here the final variables were selected.

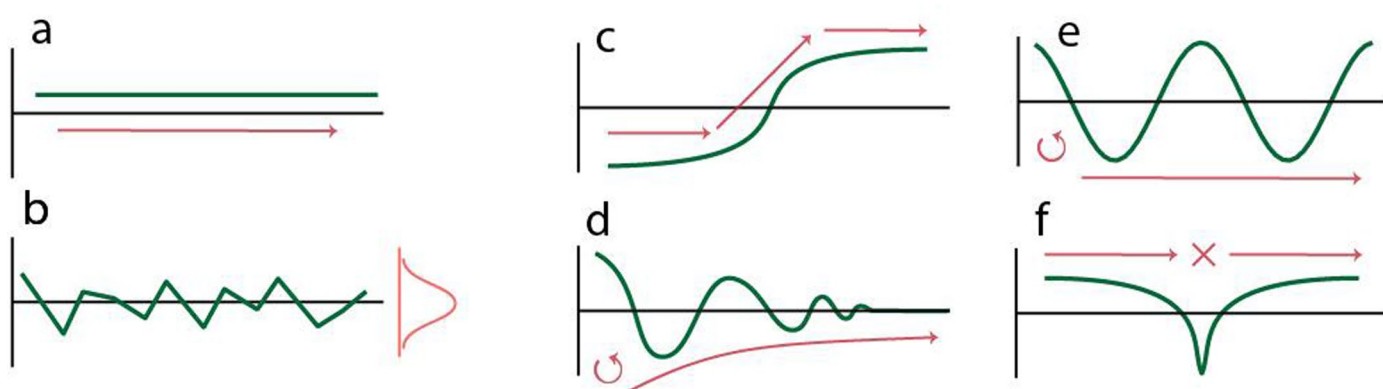

**Extended Data Fig. 4 | Example of 6 patterns detectable in time series.** a. Basal level, like the one excluded by removing the first EG in the analysis; b. Random noise; c. level change; d. perturbation; e. cycle; f. Crash. In real time series more patterns are usually combined (at least with noise) to create the main data behaviour over time.

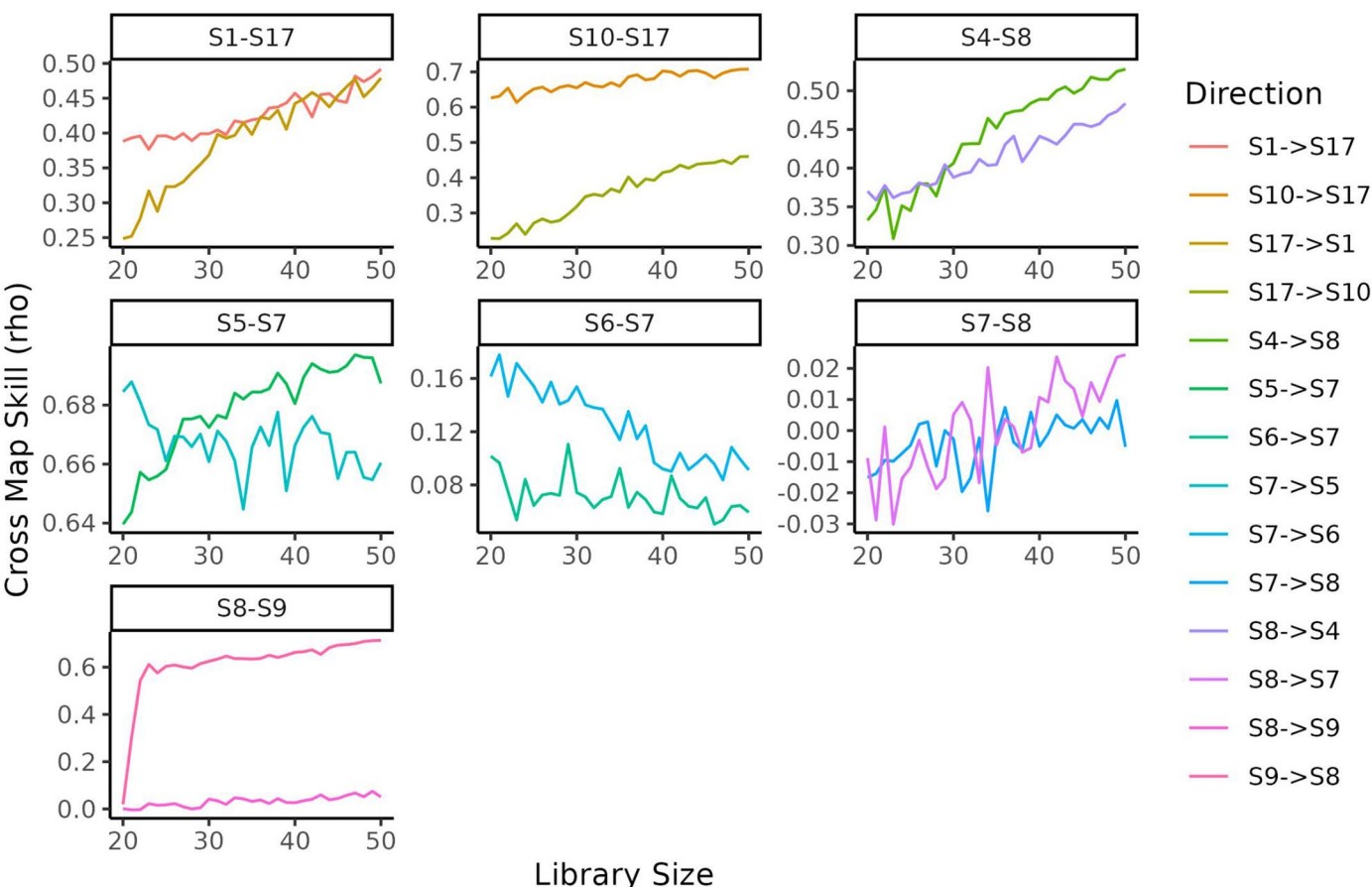

**Extended Data Fig. 5 | Convergent cross mapping plots for the causality links with putatively nonlinear signals.** The Cross Map Skill (rho) indicates the goodness of the forecasting across increasing sizes of the number of samples (Library Size). The link S9->S8 is the only fully confirmed one with a unidirectional information transfer. The edges S10-S17 and S4-S8 have a bi-directional influence which is stronger in the direction already predicted by the Granger causality test. For the edges S7-S8 and S6-S7 the Cross Map Skill shows a faint bi-directional influence; whilst for S1-S17 and S5-S7 a strong bi-directional influence.

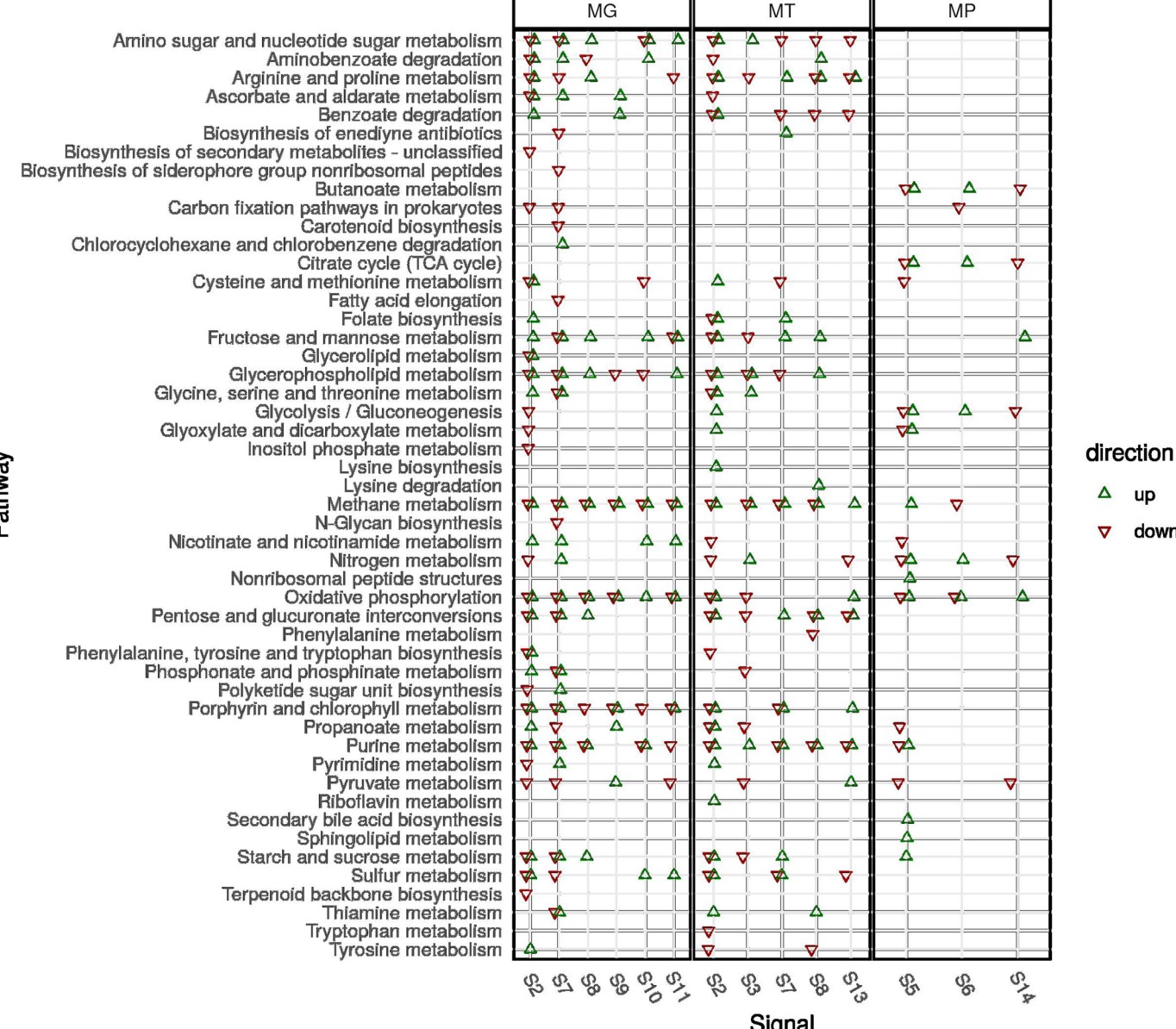

**Extended Data Fig. 6 | Loadings at the family level.** On the y-axis the taxonomic families intersect the signals they contribute to from the x-axis. If the loading is in the top 5% a green arrow pointing up marks the intersection. Similarly if the loading is in the bottom 5% (strongly negative) a red arrow pointing down marks the intersection. The vertical blocks separate the three omics, whilst the horizontal blocks separate the archaea (A.), bacteria and viruses (V.). No eukaryotic families were found to be in the top/bottom 5% of the loadings. The plot also integrates lower taxonomic labels (that is species and genus) and some of them might have opposite orientations, leading to families with both types of arrows.

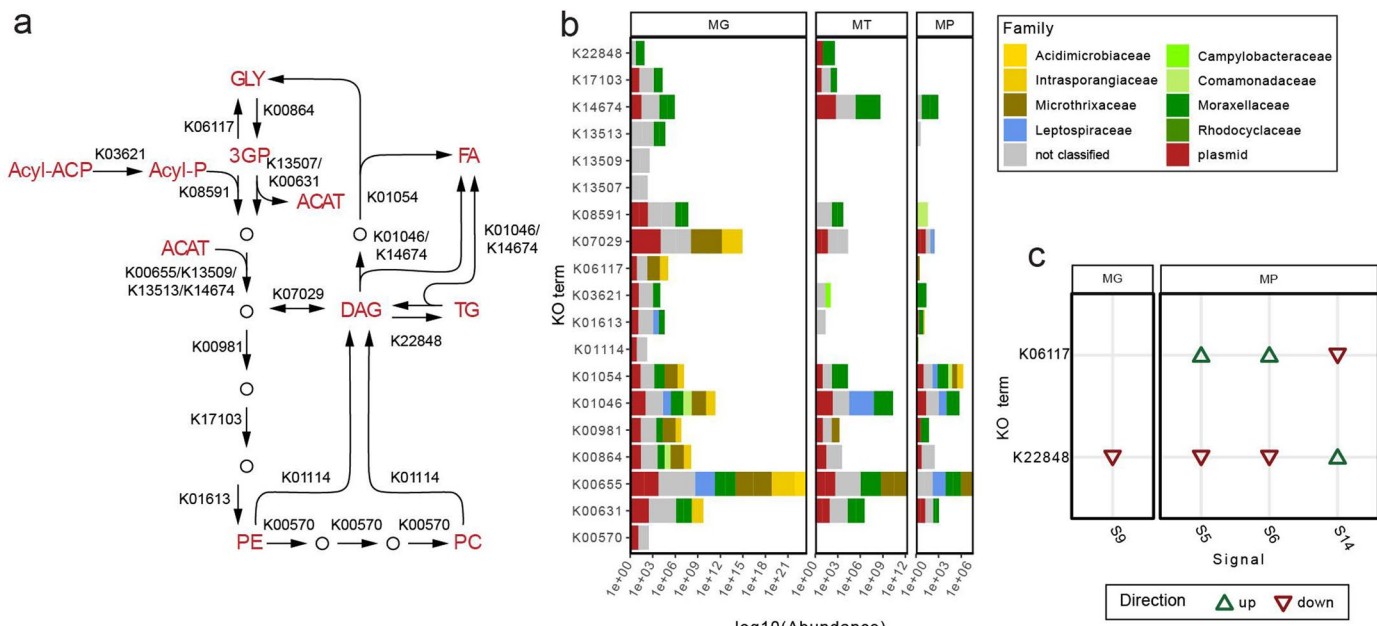

**Extended Data Fig. 7 | Loadings at the pathway level.** On the y-axis the metabolic pathways intersect the signals they contribute to from the x-axis. If the loading is in the top 5% a green arrow pointing up marks the intersection. Similarly if the loading is in the bottom 5% (strongly negative) a red pointing down marks the intersection. The vertical blocks separate the three omics. The plot integrates also lower metabolic labels (that is KO) and some might disagree in orientation, leading to pathways with both types of arrows.

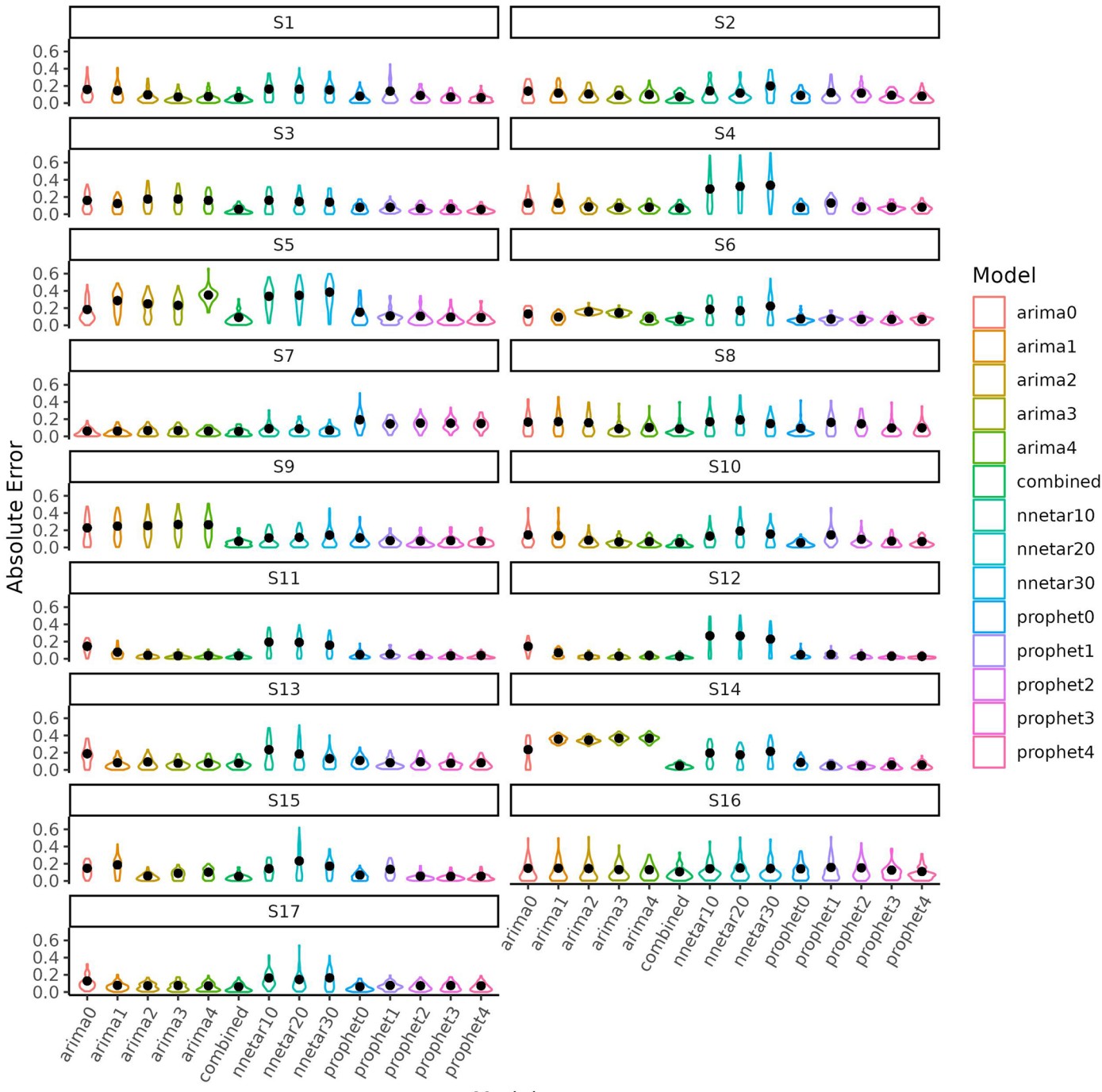

**Extended Data Fig. 8 | Triacylglycerol accumulation as a key metabolic community-wide trait.** a. Enzymatic reactions (with high abundance in at least one of the omics from LAO) leading to triacylglycerol accumulation in the community. GLY: Glycerol, Acyl-ACP: Acyl Carrier Protein, Acyl-P: Acyl phosphate, 3GP: 3-glycerol phosphate, ACAT: Acetyl-CoA, FA: Fatty Acid, DAG: Diacylglycerol, TG: Triacylglycerol, PE: phosphatidylethanolamine, PC: phosphatidylcholine. The enzyme class with KO number K22848 is responsible for the conversion of DAG in TG and, ultimately, the accumulation of TG. b. Gene and gene product abundances for the various enzymatic groups involved in the accumulation of TG varies in amount and taxonomic origin. The families belonging to the same phylum have similar colours to matching phyla in Fig. 1a. Therefore, Actinobacteria are in shades of yellow, Proteobacteria in shades of green while Leptospiraceae inherited the bure from the Spirochaetes. c. The gene abundance of K22484 is influenced by S9, indicating a, perhaps indirect effect on NH4 levels.

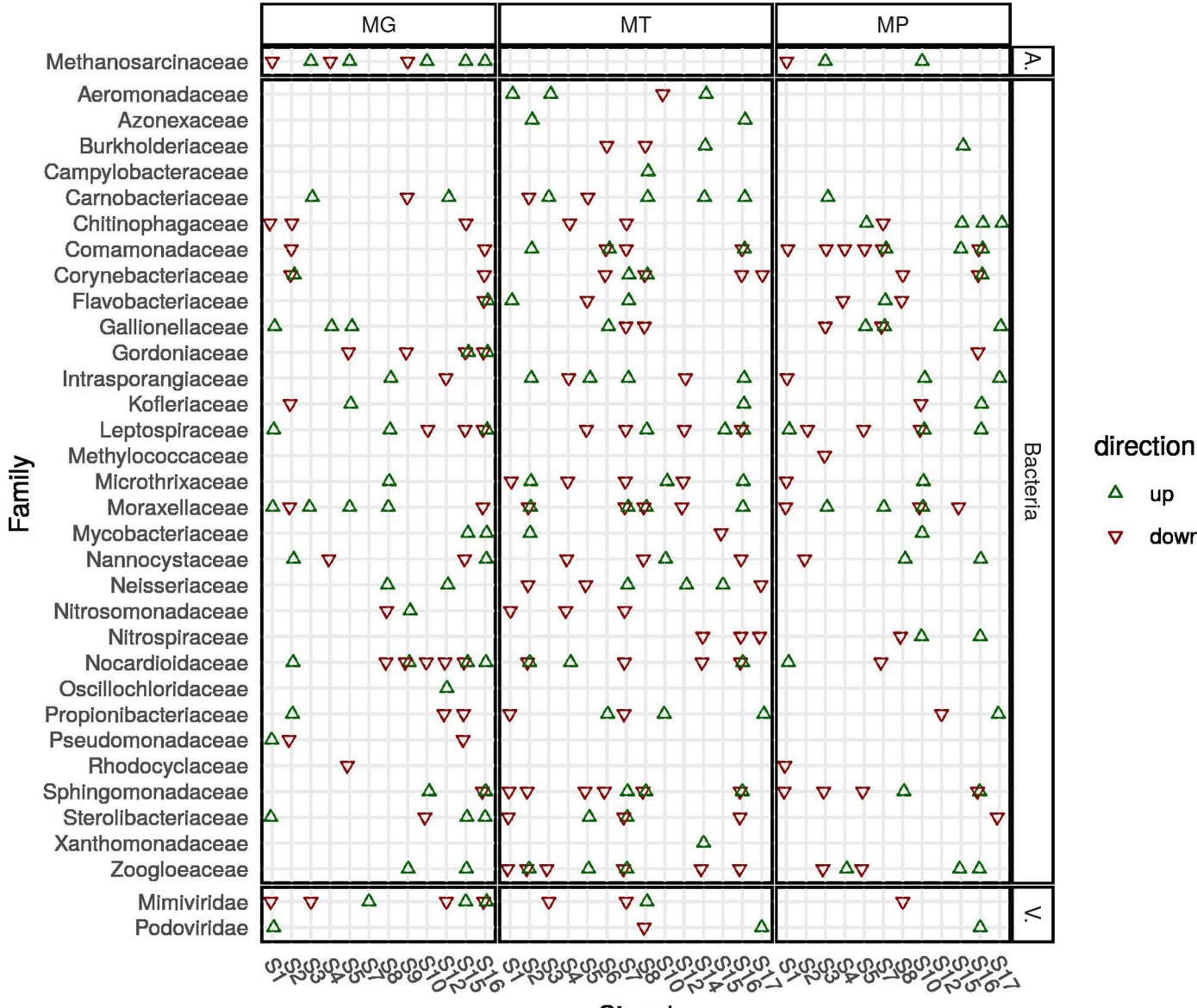

**Extended Data Fig. 9 | Selection of the models.** Absolute error profiles over the training data of the tested models for each of the seventeen signals S1-17; the black dot indicates the RSME. A low RMSE indicates that the predictions and the real data are close; vice-versa a high value shows distant data points. Therefore RMSE is useful when comparing multiple models. Elongated violin plots indicated a spread of values (that is both correctly and incorrectly predicted weeks), a 'short' and 'wide' distribution with an upper tail indicated a 'focused' prediction overall with some outliers, whilst a simple 'short' and 'wide' distribution is obtained for very coherent predictions (that is constantly correct or incorrect).

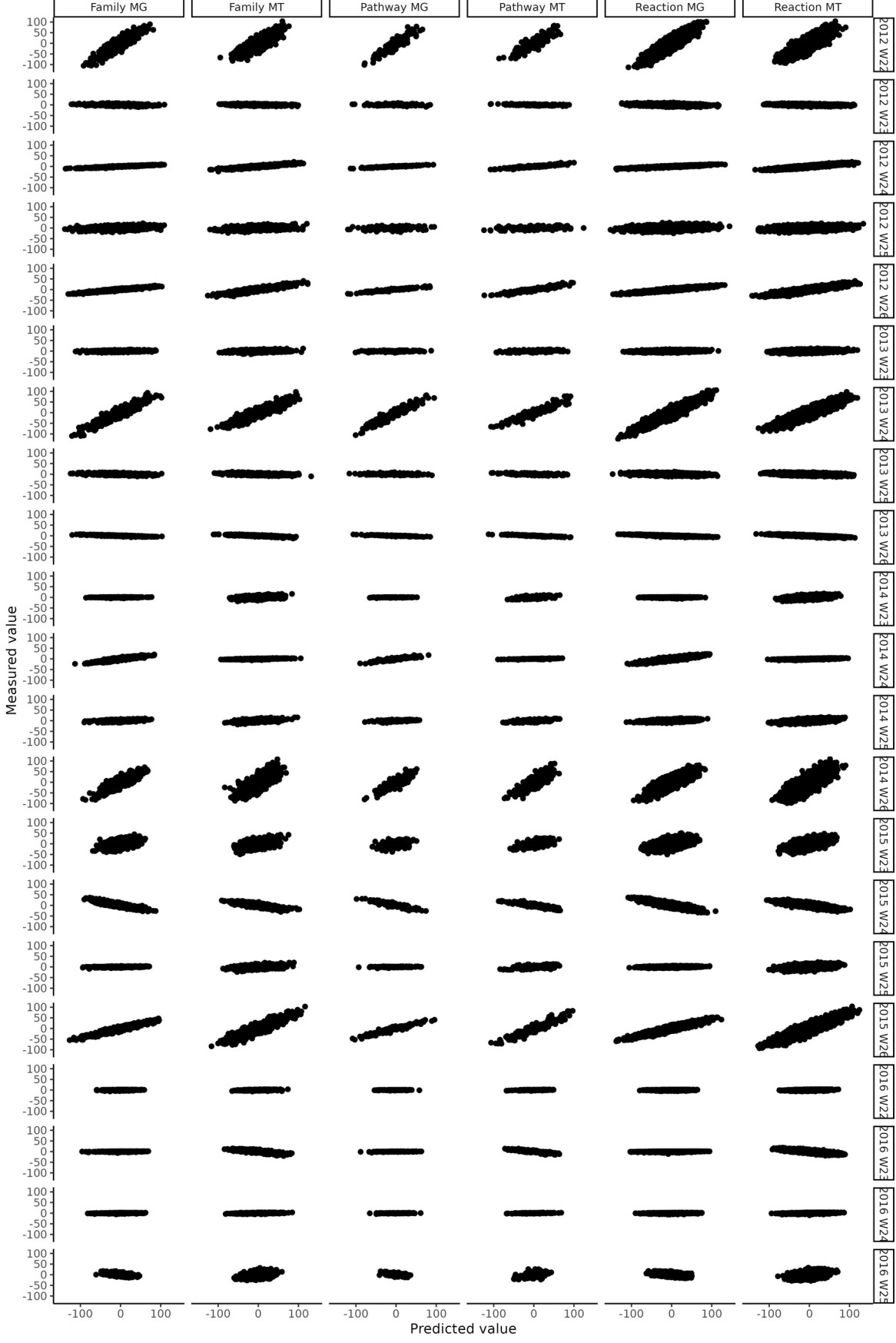

**Extended Data Fig. 10 | Predictions per-week.** Reconstructed abundance and gene expression of all the microbial families, reactions and pathways in the community versus the real one for each sample in the test set.

Francesco Delogu

# Reporting Summary

## Statistics

For all statistical analyses, confirm that the following items are present in the figure legend, table legend, main text, or Methods section.

| n/a | Confirmed | |
|---|---|---|
| ☐ | ☒ | The exact sample size ($n$) for each experimental group/condition, given as a discrete number and unit of measurement |
| ☐ | ☒ | A statement on whether measurements were taken from distinct samples or whether the same sample was measured repeatedly |
| ☐ | ☒ | The statistical test(s) used AND whether they are one- or two-sided<br>*Only common tests should be described solely by name; describe more complex techniques in the Methods section.* |
| ☐ | ☒ | A description of all covariates tested |
| ☐ | ☒ | A description of any assumptions or corrections, such as tests of normality and adjustment for multiple comparisons |
| ☐ | ☒ | A full description of the statistical parameters including central tendency (e.g. means) or other basic estimates (e.g. regression coefficient) AND variation (e.g. standard deviation) or associated estimates of uncertainty (e.g. confidence intervals) |
| ☐ | ☒ | For null hypothesis testing, the test statistic (e.g. $F$, $t$, $r$) with confidence intervals, effect sizes, degrees of freedom and $P$ value noted<br>*Give P values as exact values whenever suitable.* |
| ☒ | ☐ | For Bayesian analysis, information on the choice of priors and Markov chain Monte Carlo settings |
| ☒ | ☐ | For hierarchical and complex designs, identification of the appropriate level for tests and full reporting of outcomes |
| ☐ | ☒ | Estimates of effect sizes (e.g. Cohen's $d$, Pearson's $r$), indicating how they were calculated |

*Our web collection on statistics for biologists contains articles on many of the points above.*

## Software and code

Policy information about availability of computer code

| Data collection | Sequencing data were generated with an illumina machine and the software provided by the manufacturer for the base calling. |
|---|---|
| Data analysis | The software (and versions when available) used within this work include:<br>SPAdes (ver. 3.13.1)<br>cbar (ver. 1.2)<br>plasflow (ver. 1.1.0)<br>VirSorter (ver. 1.0.6)<br>deepVirFinder (ver 1.0)<br>euktep (ver. 0.6.7)<br>metaeuk (ver. 3.8dc7e0b)<br>IMP (ver. 3)<br>dRep (ver. 0.5.4)<br>CheckM (ver. 1.0.7)<br>CD-HIT (ver. 4.6.8)<br>mmseq (ver. 1.0.9)<br>CAT/BAT (ver. 5.1.2)<br>mantis (ver. 1.02)<br>AUGUSTUS (ver. 3.3.3)<br>R statistical package (ver. 3.4.4)<br>Cytoscape (ver 3.5.1)<br>snakemake (ver from 7.7.0) |

```
bwa (ver 0.7.17)
bam2hits (ver 1.0.9)
mmseq (ver 1.0.9)
mmcollapse
SearchGUI(ver 3.3.20)
PeptideShaker (ver1.16.45)
samtools (ver 1.11)
R (ver 3.3.4)
fable (ver 0.3.1)
lmtest (ver 0.9-38)
rEDM (ver 1.14.0)
```

For manuscripts utilizing custom algorithms or software that are central to the research but not yet described in published literature, software must be made available to editors and reviewers. We strongly encourage code deposition in a community repository (e.g. GitHub). See the Nature Portfolio guidelines for submitting code & software for further information.

## Data

Policy information about availability of data

All manuscripts must include a data availability statement. This statement should provide the following information, where applicable:

- Accession codes, unique identifiers, or web links for publicly available datasets
- A description of any restrictions on data availability
- For clinical datasets or third party data, please ensure that the statement adheres to our policy

The generated metagenomics and metatranscriptomics reads (FASTQ) files, as well as the previously produced data, are available as NCBI BioProject PRJNA230567. The MP data from the PRIDE repository with accession number PXD013655. The environmental parameter measurements are available at https://github.com/fdelogu/microforecast.

## Research involving human participants, their data, or biological material

Policy information about studies with human participants or human data. See also policy information about sex, gender (identity/presentation), and sexual orientation and race, ethnicity and racism.

| | |
|---|---|
| Reporting on sex and gender | N/A |
| Reporting on race, ethnicity, or other socially relevant groupings | N/A |
| Population characteristics | N/A |
| Recruitment | N/A |
| Ethics oversight | N/A |

Note that full information on the approval of the study protocol must also be provided in the manuscript.

# Field-specific reporting

Please select the one below that is the best fit for your research. If you are not sure, read the appropriate sections before making your selection.

☐ Life sciences     ☐ Behavioural & social sciences     ☒ Ecological, evolutionary & environmental sciences

For a reference copy of the document with all sections, see nature.com/documents/nr-reporting-summary-flat.pdf

# Ecological, evolutionary & environmental sciences study design

All studies must disclose on these points even when the disclosure is negative.

| | |
|---|---|
| Study description | Analysis and forecasting of a multi meta-omics time series from the foam community resident in the anoxic tank from a wastewater treatment plant. |
| Research sample | The samples are individual floating foam islets (one per sample) from the anoxic tank of the biological wastewater treatment plant in Schifflange (Luxembourg). |
| Sampling strategy | Individual floating sludge islets were collected from the same spot of the anoxic tank, along with physico-chemical parameters of the water, i.e. pH, temperature, conductivity, oxygen.<br>This work is part of an ongoing multi-annual project. Thus, all the samples were subjected to the same experimental protocols. Please refer to detailed methods on sampling procedures in previous publications: |

| Data collection | https://doi.org/10.1038/ncomms6603<br>https://doi.org/10.1038/npjbiofilms.2015.7 |
| --- | --- |
| Data collection | Laura A. Lebrun and Emilie E.L. Muller performed the concomitant biomolecular extractions resulting in fractions of DNA, RNA, proteins and metabolites for each in situ sample. R. Halder performed the DNA and RNA library preparation and next-generation sequencing (NGS) to obtain MG and MT data. |
| Timing and spatial scale | All the samples were taken always from the same tank. The 52 training set covers the time period March 2011 to May 2012 with roughly weekly sample. The test set includes 21 samples from the month of June in the years 2012-2016. |
| Data exclusions | No sample was excluded. ORFs, transcripts or proteins detected in less than 20% of the training set were not analysed. |
| Reproducibility | Experimental procedures adhered to previously published protocols. Open source software was used in all the computational analyses. The code for the statistical analysis is available on Github. |
| Randomization | The training and test set samples were extracted and sequenced in different time and using different methods as specified in the manuscript making a full randomization impossible. However, the batch effect has been formally addressed. The training set series was randomized for extraction and sequencing. |
| Blinding | Blinding does not apply. |

Did the study involve field work? ☒ Yes ☐ No

## Field work, collection and transport

| Field conditions | The anoxic tank of the biological wastewater plant was sampled under any condition (as long as there was foam present). |
| --- | --- |
| Location | SIVEC wastewater treatment plant in Schifflange (Luxembourg). |
| Access & import/export | Access was granted to the research personnel based on agreement between the principal investigator, Prof. Paul Wilmes (on behalf of the research institution), and the wastewater treatment facility management (Mr. Bissen and Mr. Di Pentima) from the Syndicat Intercommunal a Vocation Ecologique (SIVEC), Schifflange, Luxembourg. All research personnel are informally introduced to the management and personnel of the facility prior to conducting any work. Research personnel were not provided with keys or electronic access cards, and thus could only enter the premises upon the permission of personnel at the entrance of the facility. |
| Disturbance | Sampling had a minimum-to-no impact on the operations of the wastewater treatment facility. The work of the researchers did not require (complete or partial) shutdown or any operational disruption of the facility. Sampling was performed by the research personnel (Emilie E.L. Muller and Laura A. Lebrun) without any involvement of the staff of the facility. Research personnel either brought their own equipment or used equipment from the site, which was dedicated to them, thus not hindering any operations or personnel within facility. Researchers could access operational readings (e.g. temperature, inflow, outflow, etc.) of the facility directly via a dedicated web portal of the facility using login credentials provided by the facility management. Two formal meetings weres organized between researchers and management of the facility over the past five years. |

# Reporting for specific materials, systems and methods

We require information from authors about some types of materials, experimental systems and methods used in many studies. Here, indicate whether each material, system or method listed is relevant to your study. If you are not sure if a list item applies to your research, read the appropriate section before selecting a response.

## Materials & experimental systems

| n/a | Involved in the study |
| --- | --- |
| ☒ ☐ | Antibodies |
| ☒ ☐ | Eukaryotic cell lines |
| ☒ ☐ | Palaeontology and archaeology |
| ☒ ☐ | Animals and other organisms |
| ☒ ☐ | Clinical data |
| ☒ ☐ | Dual use research of concern |
| ☒ ☐ | Plants |

## Methods

| n/a | Involved in the study |
| --- | --- |
| ☒ ☐ | ChIP-seq |
| ☒ ☐ | Flow cytometry |
| ☒ ☐ | MRI-based neuroimaging |

