## [Peer Review File · Nature Ecology & Evolution]

Peer Review Information

Journal: Nature Ecology & Evolution

Manuscript Title: Forecasting the dynamics of a complex microbial community using integrated meta-omics

Corresponding author name(s): F. Delogu, P. Wilmes

Editorial Notes:

Reviewer Comments & Decisions:

Decision Letter, initial version:

11th January 2023

Dear Professor Wilmes,

Thank you again for your patience while it took us some extra time to recruit a full panel of reviewers. Your manuscript entitled "Forecasting of a complex microbial community using meta-omics" has now been seen by three reviewers, whose comments are copied below. The reviewers have raised a number of concerns which we would like to see addressed in a heavily revised manuscript before we can reach a final decision regarding publication in Nature Ecology & Evolution.

We therefore invite you to revise your manuscript taking into account all reviewer comments. Please highlight all changes in the manuscript text file.

* If you have not done so already please begin to revise your manuscript so that it conforms to our Article format instructions at <http://www.nature.com/natecolevol/info/final-submission>. Refer also to any guidelines provided in this letter.

[REDACTED]

2Note: This URL links to your confidential home page and associated information about manuscripts you may have submitted, or that you are reviewing for us. If you wish to forward this email to co-authors, please delete the link to your homepage.

Nature Ecology & Evolution is committed to improving transparency in authorship. As part of our efforts in this direction, we are now requesting that all authors identified as 'corresponding author' on published papers create and link their Open Researcher and Contributor Identifier (ORCID) with their account on the Manuscript Tracking System (MTS), prior to acceptance. ORCID helps the scientific community achieve unambiguous attribution of all scholarly contributions. You can create and link your ORCID from the home page of the MTS by clicking on 'Modify my Springer Nature account'. For more information please visit www.springernature.com/orcid.

[REDACTED]

Reviewer expertise:

Reviewer #1: Predictive ecology, time series analysis

Reviewer #2: Wastewater microbial ecology

Reviewer #3: Multi 'omics, microbial ecology

Reviewers' comments:

Reviewer #1 (Remarks to the Author):

Review for "Forecasting a complex microbial community using meta-omics" by Delogu et al.

The paper presents a case study for forecasting microbial communities' dynamics using biological wastewater treatment plants. I do not have the expertise to comment on the multi-omics analysis the

2authors did and the correctness of the various decisions they made. Thus, my comments focus on the derived temporal data and its analysis.

I find this work has several merits. In particular, I like a lot how the authors in Figure 2a identify 17 "signals" that represent the essential temporal behavior of the community and how they interpret these signals in Supplementary Figure 6. However, I also have strong concerns about the methods used by the authors to arrive at these signals (see below). These concerns make me doubt the validity of the results, and thus I cannot recommend the paper for publication in its present form.

Major comments

1. Line 172. The authors write, "the three quantification matrices (MG, MT and MP) were used to compute summary matrices according to the ORFs' descriptors. Hence we computed one matrix per omic layer for the six formed taxonomic descriptors (Phylum, Class, Order, Family, Genus and Species) and two functional ones (KO terms and pathways)". I could not find the exact definition of how these matrices are constructed anywhere in the text or Methods section. Furthermore, it is also unclear how and if these 27 matrices can be "aggregated" to obtain a single list of "eigengenes" using the SVD method. I consider this an important issue that needs to be clarified in the revised manuscript.

2. As a follow-up to the above comment, I have another concern about the application by the authors of the SVD method described by Orly et al. to obtain the "eigengenes". The dimensional reduction is obtained by choosing only a few singular values (i.e., "eigengenes"). This dimensional reduction is meaningful only if there is a significant separation between the magnitude of these chosen few singular values and the rest. In the "Eigengenes and their analysis" methods section, the authors indicate they choose the subset of eigengenes to study by testing if it is time-dependent or not according to some criterion. Therefore, to check if these eigengenes are meaningful, it is necessary to test and discuss if there is a separation in the size of the singular values or not. Supplementary Figure 4 goes in this direction. Although in line 184 of the main text, the author claim this figure considers 210 eigengenes, in reality, it seems limited to consider only the summary matrices. And even in that case, it seems unclear that there is a separation for all multi-omics datasets (e.g., for the MG dataset considering the EG counts).

3. Line 194. Assuming that the 210 eigengenes identified by the authors are meaningful (see points 1 and 2 for my comments), I also have some concerns about the methods and interpretation used to cluster them into 17 groups ("signals" in the authors' language). First, how robust are these clustering into 17 groups to different clustering methods? And why choose the "dynamicTreeCut" method described in the methods section and not others? Furthermore, each cluster contains several eigengenes with their particular temporal behavior. Then, the authors the "signal" at time t associated with a cluster as the maximum at time t of all eigengenes it contains. What is the justification for this choice? Why not the average or the median?

4. Line 195. The authors claim that the constructed signals to each cluster account for "91.1% of the explained variance in the system". But in Line 181, they claim that the first eigengene accounted for "around 15%" of the explained variance, thus leaving only 85% to be explained. The authors should

3clarify this point.

The methods section does not contain enough information to understand the authors' model for the eigengenes. I presume the model Eq. 4 describes the model, but I was unable to understand the meaning of "fourier" or to parse the notation "fourier($K=\{0-4\}$)". I was utterly unintelligible to me. Thus, I could not decide on the modeling approach's validity and methods.

5. Line 237. The authors use Granger causality to assess the interaction between the 17 identified clusters. However, this notion of causality is known to be limited to linear systems only. Did the authors check for the validity of applying this method? Also, there are other methods to test for causal relations that do not require the assumption of linearity, such as the "convergent cross-mapping method" by Sugihara et al. Is the network between clusters reported in Figure 2c robust to other methods for testing causality? From this perspective, I find not convincing the interpretations derived from this network that the authors present in the section "The temporal domino of ecological events."

6. Line 335. On forecasting. The methods section lacks enough details to understand how the authors constructed the different forecasting models. For example, it is unclear if a model uses the value of all other signals to forecast a given signal. Furthermore, it is not completely clear how the authors selected the models. For example, the authors chose the best model according to the RMSE. But it is unclear whether they calculated the RMSE for all training data.

Minor comments

1. Line 198. It is unclear the meaning of "non-collinear environmental parameters".

2. Line 207. It is unclear the meaning of "mathematical variables".

There are several typos and opportunities to improve the redaction and presentation of the results.

Reviewer #2 (Remarks to the Author):

I was very pleased to have the opportunity review this manuscript. I apologise for the delay in returning the review.

However, I had to spend a little time acquainting my self with the rudiments of Singular Value Decomposition (SVD). There are however, still some parts I don't understand and I will mention them.

The manuscript describes a useful way to analyse large time series of genomic data from any environment and to make forecasts from that data.

The SVD is, I now realise, a foundational concept in the data driven models and a very logical way in which to analyse a genomic time series where there will be many more genes than sampling events.

If I understand correctly, the authors use SVD to create number of reduced dimensionality matrices, where each column represents a sampling event and each row is an "eigengene". The time eigengenes that appear to be time dependent are then analysed as a time series. The time series were then used

4to train a variety of models that were then used for prediction. The prediction was successful when the tested against aggregated samples taken subsequently in the time series.

The method and the jargon (eigengenes) is lifted from a classic paper for analysing single genomes and it took me quite a while to understand what had been done. However, I can foresee many time series being analysed in something like this manner.

I had two areas of difficulty. Firstly, understanding what had been done, the method and the terminology could perhaps be explained more clearly. I appreciate that in many circles SVD is very well understood, but that might not be the case for many readers of this journal.

Secondly a possible weak area in the paper, or my understanding, or both is the section from lines 253 to 309 where the authors attempt to relate the environment. I feel this is a little speculative. Particularly because the variation is related to components that don't seem to be in the ARIMA model. For example the references to viruses in line 260. The reader is referred to supplementary figure 6 which is just an illustration of the various possible responses in a time series. in the signals to the virus sequences. The same mysterious reference to supplementary figure 6 is made on line 291, but I cannot see how the eigengene is related to a particular microbial group. I am clearly missing something here.

If the authors are proposing a generalised framework for data driven predictions then it is more likely to be adopted if it is described and explained with greater clarity. Some reference to general text books on SVD and data driven models might be helpful.

I have the following detailed comments to make.

Line 37 "maturity" I think the work maturity might be premature.

Line 127 "A week is the In" Is something missing here?

Line 178 The reliance on Alter et al 2000 to introduce SVD is a weakness. It might be helpful to suggest one of the many excellent text books on data driven models.

Line 279 (and line 224) "ecological interactions" The authors may wish to mention work by Axel Rossberg (Nature Communications 12 (1), 1-11).

Line 405 "The signals .. according to them" I don't understand this sentence.

Reviewer #3 (Remarks to the Author):

The manuscript submitted by Delogu et al., demonstrates a potentially generalizable framework to forecast microbial community ecology over time using a biological wastewater treatment plant

5(BWWTP) as a model ecosystem. This framework uses time-resolved meta-omics (metagenomic, metatranscriptomic, and metaproteomic) data related to taxonomic and functional information as well as environmental process parameter information to resolve “eigengenes” (EGs) for use in forecasting. The authors building on a pre-existing dataset first described in Herold and colleagues (Herold M, Martínez Arbas S, Narayanasamy S, Sheik AR, Kleine-Borgmann LAK, Lebrun LA, Kunath BJ, Roume H, Bessarab I, Williams RBH, Gillece JD, Schupp JM, Keim PS, Jäger C, Hoopmann MR, Moritz RL, Ye Y, Li S, Tang H, Heintz-Buschart A, May P, Muller EEL, Laczny CC, Wilmes P. Integration of time-series meta-omics data reveals how microbial ecosystems respond to disturbance. *Nat Commun.* 2020 Oct 19;11(1):5281. doi: 10.1038/s41467-020-19006-2. PMID: 33077707; PMCID: PMC7572474) combine the power of singular value decomposition (SVD), described by Alter and colleagues for genome-wide expression data, to reduce data complexity and identify patterns of related function or activity. These patterns can reflect both measured process parameters and omic information, as well as mathematical expressions e.g., factors resolved from the analysis.

After removing the first element or principal component, the authors tested for time-dependency and then applied a seasonal autoregressive integrated moving average (ARIMA) model implemented in fable resolving 210 EGs from 51 training data sets spanning one full cycle. These EGs were correlated with one another and then clustered into 17 signal groups considered to be non-redundant but temporally linked. A time resolved causality network was constructed revealing potential modules or “cliques” of temporally coupled ecological interactions (excluding one signal, S16 that did not appear coupled) that was used to develop specific hypothesis about BWWTP processes and events e.g., biological meaning. The authors explore two of these cliques with respect to viral lysis and predation respectively. Interestingly, the production of triacylglycerol (TG) and fatty acids (FA) by lipid accumulating organisms (LAOs) was not found to be time-dependent due in part to the redundancy of pathway components spread across many different taxonomic groups. The presence of genes involved in conversion of diacylglycerol and other LAO-related pathway components on plasmids suggested the potential for lateral gene transfer to play a role in generating this redundancy.

After exploring the biological meaning associated with a subset of the 17 signal groups identified in the training data, the authors turned their attention to forecasting future BWWTP processes and events. Three models (ARIMA, Prophet and neural networks) were testing with different hyperparameters implemented in fable using environmental variables as external regressors. An additional 21 samples were collected from the month of June from 2012-2016 to validate the model using metagenomic and metatranscriptomic data sets. Five of the signal groups were accurately predicted accounting for ~22% of the variance. The authors also attempted to predict gene abundance and expression patterns based on the 17 signal group matrices. To do this they constructed a linear model for the test data based on the signal group forecasting including weighted values and intercepts and compared the results to the original values for each sample. This approach accurately predicted patterns of gene abundance and expression the following year but became much less accurate over the five-year test sampling interval.

Overall, this is an interesting paper that uses well established statistical methods in a creative way to model microbial ecology at the surface of a BWWTP. The use of SVD followed by correlation analysis, clustering and network mapping provides a useful framework for identifying temporal patterns in time-series data that may not be apparent when working with a single type of omics data. The authors do a

6good job exploring the biological meaning in relevant signal groups and forecasting efforts seem promising albeit with several limitations that remain to be addressed or reconciled.

In the introduction, the authors refer to previous efforts to forecast compositional and gene expression dynamics in microbial communities and refer to the latter as an open challenge. In this light, it would be helpful to comment on prior work by Reed and colleagues (Reed DC, Algar CK, Huber JA, Dick GJ. Gene-centric approach to integrating environmental genomics and biogeochemical models. *Proc Natl Acad Sci U S A*. 2014 Feb 4;111(5):1879-84. doi: 10.1073/pnas.1313713111. Epub 2014 Jan 21. PMID: 24449851; PMCID: PMC3918765) and Louca and colleagues (Louca S, Hawley AK, Katsev S, Torres-Beltran M, Bhatia MP, Kheirandish S, Michiels CC, Capelle D, Lavik G, Doebeli M, Crowe SA, Hallam SJ. Integrating biogeochemistry with multiomic sequence information in a model oxygen minimum zone. *Proc Natl Acad Sci U S A*. 2016 Oct 4;113(40):E5925-E5933. doi: 10.1073/pnas.1602897113. Epub 2016 Sep 21. PMID: 27655888; PMCID: PMC5056048) that use gene-centric approaches to model microbial community gene expression at the level of transcripts and proteins. These methods would seemingly apply in the context of BWWTPs and should be considered when discussing an integrated framework for modeling community structure and function.

With respect to the methods used, the authors are highly encouraged to provide more detail on their statistical and modeling workflows. As presented it would be difficult to reproduce their results even if provided the input matrices. More information is needed for reproducibility including how the hyperparameters were selected for the seasonal ARIMA, etc., and how the linear model was developed to reconstruct the expression patterns for the June test data. Inclusion of an actual workflow diagram for the modeling effort would be helpful and guide a reader through the methods and results with more clarity. The authors are also encouraged to provide more context for the data sets being used in this study. The 51 data sets used in training have been previously described by the authors but seemingly analyzed using a different workflow in the present study. For example, the present manuscript uses a different number of rMAGs from the prior study, but the criteria used for increased inclusivity is not clearly defined. Moreover, the current study uses GTDB taxonomy while the prior study used AMPHORA2 making direct comparison of the taxonomic groups identified extremely difficult. The authors should provide a more coherent and intertextual narrative linking this past manuscript to the present one in review.

With respect to model development, the authors note in passing that their training data set did not span two complete cycles, and that they combined Fourier terms to compensate for this omission with respect to seasonality. This seems like a relatively important aspect of model development and needs to be better explained. For ARIMA models, the rule of thumb is that a minimum of 50 samples is needed and that more observations e.g., >100 are needed when dealing with seasonal components. This has implications for model development and potential overfitting. There is an emphatic statement that the model was robust for two years before significant decay in R2 values from linear regression of the training data. However, the training data ends on May of the same year that the first June data set is resolved. One might surmise that there is significant hysteresis in the system that would carry over for some time e.g., months before other factors could alter the trajectory of the system. Thus, from the standpoint of training and test data it is unclear why the authors did not include the June 2012 samples in their training data set or at least explain this proximity in time with respect to training and test data. Finally, and with respect to results interpretation, six signal groups were considered

7accurately predicted in the forecasting part of the work including S1, S2, S4, S5, S10 and S16. The training set for S10 was not fully captured by the model and therefore was excluded from analysis. It would be of interest to relate the remaining signal groups back to the network rendered in Figure 2C. In this case S16 was excluded because it did not demonstrate a temporal relationship with other groups making the total number of groups identified as significant with respect to the network equal to four (S1, S2, S4 and S5). Why were some groups identified from C1 and C2 but not others? What is the interpretation of S4 and S5? Beyond indicating that a model can be constructed and used to forecast BWWTP processes for a year or more how might this framework actually be used in a monitoring or process engineering context?

Specific comments:

Introduction:

Line 104-105: Why the emphasis on LAO? A clear indication of the specific reason behind choosing to study a specific group of organisms is required in the introduction for readers less familiar with the system under study.

Line 107-108: Based on this sentence it seems that metaproteomic dataset was also generated by the authors in this study but under the methods section, they state that raw metaproteomic data was retrieved from PRIDE and reanalysed. It is unclear which of the two is correct. This relates back to the general comments above that that data sets used in this study were described in previous manuscripts and more intertextual linkages are needed.

Line 114-115: It would be useful to also include a line or two on why the three specific models were selected, what other models exist for the same and what are their shortcomings. The latter aspect is really important when introducing time series analysis to new readers and speaks to points raised above about the number of samples used in training.

Line 116: Why was June chosen for future predictions? A line of two explaining this in the introduction would also be good. Have there been any interesting observations within the treatment plant previously during this month?

Results and discussion:

Line 124-125: 51 metagenomic assemblies instead of genomic assemblies. In the same line they state a "collection of metagenome-assembled genomes" and don't mention exactly how many were obtained. Reword for clarity... "... we obtained 51 weekly samples used in the generation of MG, MT and MP data sets..." Provide a table describing these data with relevant features including sample type, number of reads, number of contigs, number of ORFs, Kos, number of peptides per sample, etc. These data have been previously published so this is a good place to elaborate on the connectivity...

Line 126: Exact numbers of how many contigs were in bins, how many were plasmids, viruses, and were unbinned is required along with read mapping information on a per sample basis.

Line 127 (Sentence formation error): "A week is the In order to..."

Line 128-130: Clustering was performed based on sequence identity but what was the identity percent cut off used isn't mentioned. The line points to the methods section but it isn't mentioned there either.

Line 130-155: Another place to make connections to prior work.

Figure 1 caption: MAG completion and contamination isn't mentioned under results or the methods section. Based on the caption a completion of 75% and a contamination of maximum 25% is used as a cut-off to filter the MAGs. It is unclear why these specific cutoffs were used. The yellow color is missing from the legend.

Figure 2: Would be helpful to use same colour coding in this figure as in Figure 1 for Families within a given Phylum level grouping (can shade the colours if more than one representative within a family).

Figure 2B: A clear description of the model panel including the ARIMA components on the x axis would help understand the figure and the results better.

Line 160-164: Why not adopt established community standards for high and medium quality MAGs described in Bowers and colleagues (Bowers RM, Kyrpides NC, Stepanauskas R, Harmon-Smith M, Doud D, Reddy TBK, Schulz F, Jarett J, Rivers AR, Eloie-Fadrosh EA, Tringe SG, Ivanova NN, Copeland A, Clum A, Becraft ED, Malmstrom RR, Birren B, Podar M, Bork P, Weinstock GM, Garrity GM, Dodsworth JA, Yooseph S, Sutton G, Glöckner FO, Gilbert JA, Nelson WC, Hallam SJ, Jungbluth SP, Ettema TJG, Tighe S, Konstantinidis KT, Liu WT, Baker BJ, Rattei T, Eisen JA, Hedlund B, McMahon KD, Fierer N, Knight R, Finn R, Cochrane G, Karsch-Mizrachi I, Tyson GW, Rinke C; Genome Standards Consortium; Lapidus A, Meyer F, Yilmaz P, Parks DH, Eren AM, Schriml L, Banfield JF, Hugenholtz P, Woyke T. Minimum information about a single amplified genome (MISAG) and a metagenome-assembled genome (MIMAG) of bacteria and archaea. *Nat Biotechnol.* 2017 Aug 8;35(8):725-731. doi: 10.1038/nbt.3893. Erratum in: *Nat Biotechnol.* 2018 Feb 6;36(2):196. Erratum in: *Nat Biotechnol.* 2018 Jul 6;36(7):660. PMID: 28787424; PMCID: PMC6436528)?

Line 187-189: Reword for clarity. I think you mean that given their independence the matrices will contain some degree of redundant information...

Line 301-307: What is the figure that summarizes this information?

Data and code availability:

Line 492: Zenodo is misspelled.

Materials and Methods:

Line 541: Oxygen is written twice.

Line 558: It would be nice to state the parameters rather than say custom parameters especially if someone is trying to exactly replicate the analysis.

Line 578: What was the reason for using these three binners specifically?

Line 590: It would be better to specify the different databases Mantis uses rather than mentioning that all databases were used.

Line 660: Performed three tests according to the text above but mentioned four tests here.

Line 692: (Sentence formation error): "test test with the function forecast form the fable package..."

Line 698: change tense "run" should be "ran"

Line 700: change tense "rebuild" should be "rebuilt or reconstructed"

General comment: Some software used in this study have version numbers but majority of them are missing this key piece of information. Secondly, at places abbreviations are used without explaining the full form first. Eg. CAT and BAT used for taxonomic assignment.

Supplementary figure 9: caption broken up by the figure.

Supplementary figure 2: reduce size of labels in red.

*****END*****

Author Rebuttal to Initial comments

First of all we want to thank the editor and reviewers for their very constructive comments. In the present document we addressed their comments integrating more information, numerical outputs and plots as support. The manuscript was revised accordingly as well as the corresponding code which is available at <https://github.com/fdelogu/microforecast>.

The review has been broken down and the individual comments addressed. The comments have been tagged as **RX.CX**, where R stands for "Reviewer", C for "Comment" and the Xs their respective number. When multiple comments targeted the same aspect we cross-referenced the comments using their **RX.CX** tags. When referring to line numbers they are referred to the "clean" version of the manuscript, however we included also a version of the manuscript where the changes we have made have been tracked.

Reviewer expertise:

10Reviewer #1: Predictive ecology, time series analysis

Reviewer #2: Wastewater microbial ecology

Reviewer #3: Multi 'omics, microbial ecology

Reviewers' comments:

Reviewer #1 (Remarks to the Author):

R1.C0 Review for "Forecasting a complex microbial community using meta-omics" by Delogu et al.

The paper presents a case study for forecasting microbial communities' dynamics using biological wastewater treatment plants. I do not have the expertise to comment on the multi-omics analysis the authors did and the correctness of the various decisions they made. Thus, my comments focus on the derived temporal data and its analysis.

I find this work has several merits. In particular, I like a lot how the authors in Figure 2a identify 17 "signals" that represent the essential temporal behavior of the community and how they interpret these signals in Supplementary Figure 6. However, I also have strong concerns about the methods used by the authors to arrive at these signals (see below). These concerns make me doubt the validity of the results, and thus I cannot recommend the paper for publication in its present form.

Response We thank the reviewer for their recognition of the merits of our study and for their detailed comments on the temporal data and analysis of the original version of the paper. We recognise the reviewer's concerns which have allowed us to strengthen the original version of the study. Please find our detailed responses to the reviewers' comments, in particular on the methodological approach, and corresponding changes to the revised manuscript outlined below.

Major comments

R1.C1 Line 172. The authors write, "the three quantification matrices (MG, MT and MP) were used to compute summary matrices according to the ORFs' descriptors. Hence we computed one matrix per omic layer for the six formed taxonomic descriptors (Phylum, Class, Order, Family, Genus and Species) and two functional ones (KO terms and pathways)". I could not find

11the exact definition of how these matrices are constructed anywhere in the text or Methods section. Furthermore, it is also unclear how and if these 27 matrices can be "aggregated" to obtain a single list of "eigengenes" using the SVD method. I consider this an important issue that needs to be clarified in the revised manuscript.

Response We thank the reviewer for this important comment. The summary matrices were computed based on the three count matrices (containing the gene, transcript and protein abundances) by summing up the values for the same descriptor. For instance, in order to compute the MT genus matrix we summed up all the entries within the same genus, resulting in a matrix with a dimension of #genera x #samples.

Once the summary matrices were computed, SVD was applied to each matrix (including both original and summary matrices) individually. This approach resulted in the large number of eigengenes (210) after time-dependency screening. However, as we aimed to identify a reference system of eigengenes, we further clustered the 210 and identified 17 clusters, from which we picked 17 representative eigengenes each, hereafter referred to as signals.

To clarify this we added **Supplementary Figure 1** to explain this aspect of the whole analysis and referenced it in the main text (**Line 200**).

R1.C2 As a follow-up to the above comment, I have another concern about the application by the authors of the SVD method described by Orly et al. to obtain the "eigengenes". The dimensional reduction is obtained by choosing only a few singular values (i.e., "eigengenes"). This dimensional reduction is meaningful only if there is a significant separation between the magnitude of these chosen few singular values and the rest. In the "Eigengenes and their analysis" methods section, the authors indicate they choose the subset of eigengenes to study by testing if it is time-dependent or not according to some criterion. Therefore, to check if these eigengenes are meaningful, it is necessary to test and discuss if there is a separation in the size

13of the singular values or not. Supplementary Figure 4 goes in this direction. Although in line 184 of the main text, the authors claim this figure considers 210 eigengenes, in reality, it seems limited to consider only the summary matrices. And even in that case, it seems unclear that there is a separation for all multi-omics datasets (e.g., for the MG dataset considering the EG counts).

Response We thank the reviewer for this comment. Based on our analyses, the time-dependent eigengenes are often among the ones with the largest eigenvalues. In order to clarify all aspects relating to eigengenes, eigenvalues and temporal dependencies, here follows a detailed description of the way we used the eigenvalue calculations and how it intersects with the time-dependency.

After the eigengenes of all the matrices (both original and summaries) were computed, we assessed the fractional eigenvalues (i.e. percentage of the eigenvalue weight per matrix) associated with them. Please find below the plot of the fractional eigenvalues per matrix. Every subfigure represents a matrix, identified by the facet column label (listing the omics levels) and the facet row labels (listing the matrix feature types). For every matrix on the x axis are listed the indices of the eigengenes and on the y axis the associated fractional eigenvalues.

We then considered the argument put forward in Alter *et al.*, whereby the first eigengene encodes the basal state of the system and therefore does not carry any temporal information. We followed their reasoning in removing the first eigengene and then recalculated the fractional

eigenvalues. The figure below follows the faceting scheme as the figure above but the eigenvalues are recomputed excluding the first eigengene.

The system studied by Alter *et al.*, was a pure culture of yeast, whilst our system is a complex microbial community. Therefore, we considered that there may be more than one eigengene associated with a large eigenvalue that is not dependent on time. A possible reason for this could be the presence of alternative steady states or technical variation and batch effects. In relation to the latter, we had already formally addressed the batch effects between the two extraction and sequencing campaigns. However, within the same campaign, the sampling was performed by different researchers. Therefore, we screened the eigengenes for temporal dependencies. In many cases, the eigengenes that were selected were also some with the largest eigengenes among their respective matrices. The figure below shows the eigengene indices (time-dependent vs time-independent) associated with their respective fractional eigenvalues. The faceting follows the same scheme as the two figures above.

Our analysis demonstrated that time-dependent eigengenes often exhibit large eigenvalues. We provided a detailed description of our eigenvalue calculations and addressed the issue of the first eigengene encoding the basal state of the system. Considering the complexity of our microbial community, compared to the axenic culture studied by Alter et al., we explored the possibility of multiple eigengenes representing different steady states or technical variation. In the end, we examined the time-dependency of the eigengenes, often selecting those with the largest eigenvalues.

R1.C3 Line 194. Assuming that the 210 eigengenes identified by the authors are meaningful (see points 1 and 2 for my comments), I also have some concerns about the methods and interpretation used to cluster them into 17 groups ("signals" in the authors' language). First, how robust are these clustering into 17 groups to different clustering methods? And why choose the "dynamicTreeCut" method described in the methods section and not others? Furthermore, each cluster contains several eigengenes with their particular temporal behaviour. Then, the authors the "signal" at time t associated with a cluster as the maximum at time t of all eigengenes it contains. What is the justification for this choice? Why not the average or the median?

Response We thank the reviewer for this comment. When clustering the eigengenes, we followed the strategy suggested in the WGCNA workflow [Langfelder 2008a]. This includes 3 steps: correlation, calculating distance and clustering. Considering we required a clustering

approach that is independent of scale (e.g. Pearson correlation) and can accommodate a complex structuring (e.g. DynamicTreeCut), the WGCNA workflow follows these criteria (we have now pointed this out in **Line 745**). In the paragraphs below are shown the steps and checks we took in the clustering process.

Clustering distances. Different distance metrics were computed and the resulting distances were highly correlated using the Mantel test (function `mantel.rtest` from the package `ada4` [Dray 2007] in R). Please find below a correlation plot with the results of the Mantel tests:

Clustering and separation of the clusters. Given the high degree of correlation for the different distance metrics, we proceeded with hierarchical clustering using the Minkowski distance. We selected DynamicTreeCut because it was shown to accommodate the natural grouping occurring in biological clustering whereby the data typically arrange themselves in clusters and subclusters [Langfelder 2008b]. According to this, it is better not to pick a “hard threshold” to separate the clusters but to account for a dynamic range which DynamicTreeCut allows. The clustering of the eigengenes was as follows:

After clustering we picked, for each cluster a representative eigengene to which we referred to as a signal. We indeed visually explored (plot below) the clusters to see if all the eigengenes followed the same trends. In some cases, it was possible to observe an “outlier” eigengene (i.e. in clusters S9 and S11), which was due to setting the minimum module size to 3 (to avoid the generation of too many clusters but leave enough room to small ones). In both cases of S9 and S11 “non-outlier” eigengenes were selected to represent the clusters. The plots are automatically produced by running the code we uploaded at: <https://github.com/fdelogu/microforecast>.

In conclusion, we followed the clustering strategy of WGCNA because it followed the required criteria. We computed various distance metrics and we observed they were highly correlated. Visual inspection of the cluster revealed the general efficiency of the strategy.

R1.C4 Line 195. The authors claim that the constructed signals to each cluster account for "91.1% of the explained variance in the system". But in Line 181, they claim that the first eigengene accounted for "around 15%" of the explained variance, thus leaving only 85% to be explained. The authors should clarify this point.

The methods section does not contain enough information to understand the authors' model for the eigengenes. I presume the model Eq. 4 describes the model, but I was unable to understand the meaning of "fourier" or to parse the notation "fourier(K={0-4})". I was utterly unintelligible to me. Thus, I could not decide on the modeling approach's validity and methods.

Response We thank the reviewer for these important comments. In brief, the 15% value was related to the first eigenvalue, then removed, and the 91.1% value is the aggregate temporal variance (i.e. only with time-dependent eigengenes and removing the first ones). Please find our responses to linked individual points below:

Explained variance. In Alter *et al.* [Alter 2000], which introduced the use of SVD in microbiology, it was postulated that the first eigengene is associated with basal gene expression (in our case we generalise the assumption to the three omic levels, i.e. gene, transcript and

19protein level). Since basal expression is independent of time, Alter *et al.* opted for removal of the first eigengene and re-calculated the fractional eigenvalue weights. To account for this in our own study, we modified the text in **line, 208** to emphasise the order of these operations and the meaning of the provided values:

In previous works, the first EG in a time-series has been demonstrated to represent “steady state” gene expression, encapsulating the largest explained variance (EV). The largest EG was therefore removed from this analysis to capture only the time-dependent information. Indeed, the first EG showed the largest EV (average 50%, s.d. 22%) in all the datasets), therefore we excluded it from the subsequent analysis, recalculated the EV, and all the following values were provided as the “temporal EV”.

EG modelling. The individual eigengenes were modelled using seasonal ARIMA (where the subtraction of the seasonal effects on the data is not required beforehand). However, knowing that the R package we used (fable **[O’Hara]**) was unable to detect seasonality effects that did not span at least two cycles (in our case it means having two years of continuous data), we decided to add an explicit term for seasonality (option provided by the package fable **[O’Hara]**). Hence, the addition of the Fourier transform term, which fits a number of sine and cosine functions to the data equal to the number of the parameter K in Eq. 4. The parameter K is therefore representative of the seasonal effects, i.e., K=0: no seasonal effect, K=1: simple seasonality, ..., K=4: complex seasonality with multiple overlapping cycles.

The ARIMA parameters p (auto-regressive terms), d (number of integration for differencing) and q (moving average terms) were automatically picked by the *arima* function from the fable package to obtain a best fit to the data.

Each eigengene was therefore modelled five times, once for each possible value of K, and the best model (selected using the R^2 value of the models) was considered in the analysis.

In order to account for these individual aspects, we have updated the text to explain better the seasonal ARIMA modelling we used alongside the corresponding input and output (**Line 724**):

The i^{th} EG was modelled using seasonal ARIMA modelling (where the subtraction of the seasonal effects on the data is not required beforehand). The ARIMA model is described by three non-seasonal parameters: p (auto-regressive terms), d (number of integration for differencing) and q (moving average terms). Considering that the training set did not

span two cycles (the hypothetical period of seasonal patterns), we added up to 4 Fourier transform terms to the model as a proxy for the seasonal component. For this, we used the arima function from the package *fable*⁴² as:

$$EG_t = \text{arima}(X + \text{fourier}(K = \{0 - 4\})) \quad (\text{Eq. 4});$$

where X is the matrix of the environmental variables and the Fourier term includes a number of sine and cosine components K , ranging from 0 to 4. The value of K therefore spans from no seasonal effect ($K=0$) to increasingly complex effects. The best model of the five was selected according to their R^2 values. The best model hence provided the weights for the environmental variables (X), for the parameters p , d , q and for as many sine and cosine terms as the selected K parameter. We called the ensemble of those variables the “explanatory variables” and we assessed their significance using ANOVA (anova).

R1.C5 Line 237. The authors use Granger causality to assess the interaction between the 17 identified clusters. However, this notion of causality is known to be limited to linear systems only. Did the authors check for the validity of applying this method? Also, there are other methods to test for causal relations that do not require the assumption of linearity, such as the “convergent cross-mapping method” by Sugihara et al. Is the network between clusters reported in Figure 2c robust to other methods for testing causality? From this perspective, I find not convincing the interpretations derived from this network that the authors present in the section “The temporal domino of ecological events.”

Response We thank the reviewer for pointing this out. We did run nonlinearity tests and nonlinear causality inferences using the package *rEDM* [Ye 2019]. We first screened with the simplex function the optimal number of dimensions for the embedding:

nature portfolio

Following this, we used the optima for each signal to assess the dependency between Nonlinearity (θ) and the Forecasting Skill (ρ):

From the plots, three signals showed a possible nonlinear nature: S7, S8 and S17; whilst all the others showed monotonous decrease of Forecast Skill at increasing values of the Theta parameter. We therefore used the Convergent Cross Mapping (CCM) framework to check the Cross Map Skill (the ability of reconstructing the data using within the CCM across ranges of

22parameters) of the pairs of nodes found in our causality analysis. In the plot below the library size stands for the number of samples used to estimate the Cross Map Skill.

The link S9->S8 is the only fully confirmed one with a unidirectional information transfer. The edges S10-S17 and S4-S8 have a bi-directional influence which is stronger in the direction already predicted by the Granger causality test. For the edges S7-S8 and S6-S7 the Cross Map Skill shows a faint bi-directional influence; whilst for S1-S17 and S5-S7 a strong bi-directional influence. We included the figure above as **Supplementary Figure 8** and added the following text in the **Methods** section (**Line 757**):

The signals were screened for nonlinearity via Empirical Dynamic Modelling as implemented in the R package rEDM⁹¹. We first identified the best number of lags (Embedding value) to analyse the signals using the simplex function and default parameters. The signals were screened with the S-map method⁹³ and only three signals appeared to be putatively nonlinear: S7, S8 and S17. All the causal links identified with the Granger causality test were tested also with the Convergent Cross Mapping method

(Supplementary Figure 8) using the function ccm with library size=c(20,50,1) and default parameters. The visualisation of the network was performed with Cytoscape⁹⁴ while adjusting manually the edges and directionality arrows to add the Empirical Dynamic Modelling to the Granger causality results.

To summarise, we conducted nonlinearity tests and nonlinear causality inference as required using rEDM. Three signals (S7, S8, S17) showed potential nonlinearity, while others displayed a monotonic decrease in forecast skill with increasing theta. Using Convergent Cross Mapping (CCM), we confirmed a unidirectional link (S9->S8) and observed bi-directional influences for other edges. We included Supplementary Figure 8 and updated the Methods section to incorporate rEDM and CCM results in the network visualization.

R1.C6 Line 335. On forecasting. The methods section lacks enough details to understand how the authors constructed the different forecasting models. For example, it is unclear if a model uses the value of all other signals to forecast a given signal. Furthermore, it is not completely clear how the authors selected the models. For example, the authors chose the best model according to the RMSE. But it is unclear whether they calculated the RMSE for all training data.

Response To address this very relevant reviewer's comment, we have added a more thorough description to cover all the aspects of the model fitting and selection, thereby dividing the original paragraph "Signal forecasting and gene abundance/expression reconstruction" into two subsections entitled "Modelling the signals and model selection" and "Forecasting the signals and reconstruction of future samples" (**Line 793**). The first subsection explicitly addresses that each signal was modelled in an independent generative process given the chosen modelling technique (with seasonal components if possible; see our response to **R1.C4** above) and the environmental parameters as the only exogenous variables; therefore the other signals were not used as explanatory variables. Moreover, we have now provided the details for all the parameters used in the models, including model ensembling and model selection (lowest RMSE on all the models for each signal). The first revised subsection is as follows (**Line 770**):

Modelling the signals and model selection

For each signal we trained multiple models using three techniques (ARIMA, Prophet and neural network using the functions ARIMA, prophet and NNETAR, respectively, using the R package fable⁴²), alongside a range of values for the parameters accounting for seasonal components. Each signal was modelled in a separate process, whereby the signal itself was the target of the model and the environmental parameters were the only

exogenous variables. Therefore, we did not use any information transfer among the signals in the modelling.

We fitted the ARIMA with up to four Fourier components (see Methods - Eigengenes and their analysis) whilst the parameters p , d and q were automatically optimised by the function; leading to five ARIMA models (one for each increment of Fourier transform terms, starting with 0). For the Prophet modelling we specified seasonality (period="year", type="additive" and order=from 0 to 4, analogously to the Fourier transform terms of the ARIMA) and growth (type="logistic"), resulting in five Prophet models. The neural network function was used whereby the number of nodes in the hidden layer were set to 10, 20 and 30. The thirteen models for each signal were scored according to their RMSE and the three models with the lowest RMSE were combined (weighted by $1-RMSE$), as a fourteenth ensemble model. The RMSE was calculated for the fourteen models as well. For each signal, the model with the lowest RMSE was selected for the putative generative process and used to forecast the test set with the function forecast from the fable package and supplying environmental parameter readings..

The second subsection addresses the forecasting into the future and the "reconstruction" of the future samples for validation:

Forecasting the signals and reconstruction of future samples

The 17 signals were forecast (forecast function from the fable package⁴²) for the 5 years following the training set, using the fitted models and the environmental variables (recorded in the forecasting period) as exogenous variables. The forecast signals were therefore used to reconstruct the information in the future samples, i.e. the actual gene abundances (MG) and expression values (MT) matrices. This was possible because the matrices used to summarise the LAO community can be expressed using a linear combination of the 17 signals plus a basal gene abundance/expression (that we previously removed in the analysis). We therefore decided to "reconstruct" June 2012-2016 matrices for the reaction, pathway and family summarisation of the gene abundances (MG) and expression values (MT). We ran linear regression (lm function) using the six training set matrices for the categories above as target variables and the 17 signals as explanatory variables. We then "reconstructed" the test matrices using a linear combination of the forecasted signals over the test set, weighted by the betas and offset by the intercept (basal level) derived from the linear model on the training set while

25also adding the intercept (basal level). The reconstructed and the real samples were compared on an individual basis (Supplementary Figure 12) and on a month-average (Figure 4).

Minor comments

1. Line 198. It is unclear the meaning of "non-collinear environmental parameters".

Response The environmental parameters were checked for collinearity (i.e. being linearly dependent). Collinearity leads to worse models, hence we reduced the used variables to a set of non-collinear ones. To reflect this, we have changed the text as follows:

The 17 representative signals (S1-17) were modelled using the environmental parameters as exogenous variables (after collinearity screening, see Methods) as shown in Figure 2b.

2. Line 207. It is unclear the meaning of "mathematical variables".

Response We changed the occurrences of "mathematical variables" and similar terms to "ARIMA components" to be more explicit and coherent with the rest of the manuscript.

There are several typos and opportunities to improve the redaction and presentation of the results.

Response We thank the reviewer for pointing this out. We have carefully revised the manuscript to improve the redaction and presentation of the results.

Reviewer #2 (Remarks to the Author):

R2.C0 I was very pleased to have the opportunity review this manuscript. I apologise for the delay in returning the review.

However, I had to spend a little time acquainting my self with the rudiments pf Singular Value Decomposition (SVD). There are however, still some parts I don't understand and I will mention them.

The manuscript describes a useful way to analyse large time series of genomic data from any environment and to make forecasts from that data.

The SVD is, I now realise, a foundational concept in the data driven models and a very logical way in which to analyse a genomic time series where there will be many more genes than sampling events.

If I understand correctly, the authors use SVD to create number of reduced dimensionality matrices, where each column represents a sampling event and each row is an "eigengene". The time eigengenes that appear to be time dependent are then analysed as a time series. The time series were then used to train a variety of models that were then used for prediction. The prediction was successful when the tested against aggregated samples taken subsequently in the time series.

The method and the jargon (eigengenes) is lifted from a classic paper for analysing single genomes and it took me quite a while to understand what had been done. However, I can foresee many time series being analysed in something like this manner.

Response We thank the reviewer for their recognition of the work and their breakdown of the work. We acknowledge the reviewer's detailed comments regarding the overall aim and scope of the work. The comments and suggestions have allowed us to strengthen the study and manuscript. Wherever possible, we have in particular clarified the language used and have refrained from using too much jargon.

R2.C1. I had two areas of difficulty. Firstly, understanding what had been done, the method and the terminology could perhaps be explained more clearly. I appreciate that in many circles SVD is very well understood, but that might not be the case for many readers of this journal.

27

Response In response to comments by reviewer 1, we have added a new figure (**Supplementary Figure 1**) to depict the general workflow (see **R1.C1**) and expanded widely on the methodology (see **R1.C4** and **R1.C6**).

R2.C2 Secondly a possible weak area in the paper, or my understanding, or both is the section from lines 253 to 309 where the authors attempt to relate the environment. I feel this is a little speculative. Particularly because the variation is related to components that don't seem to be in the ARIMA model. For example the references to viruses in line 260. The reader is referred to supplementary figure 6 which is just an illustration of the various possible responses in a time series. in the signals to the virus sequences. The same mysterious reference to supplementary figure 6 is made on line 291, but I cannot see how the eigengene is related to a particular microbial group. I am clearly missing something here.

Response We thank the reviewer for these points. We have now included more details in the analyses related to the environment, the components and the modelling.

From eigengenes to signals and vice versa. The 17 signals that were used in the ARIMA models and discussed in the paragraph “**The temporal domino of ecological events in LAO**” are the representative eigengenes from the 17 clusters obtained from the clustering of all the time-relevant eigengenes. Therefore, when the ARIMA modelling identified a generative process and a link to the environmental variables for a given signal, we extended that to all the eigengenes in the cluster corresponding to the given signal. We added the following text to highlight the 2-way nature of the relationship and how this was linked to the ecological hypotheses (**Line 287**):

In order to explore the ecological and environmental aspects of the system we recalled the two-way relationship between the signals and the other eigengenes they clustered with. In doing so, we considered the generative processes and causal links of the signals which were applied to the 17 clusters. In this way it was possible to use the top/bottom loadings of the eigengenes to link the high level depiction of the system (the signals) to the taxa and functions of the microbial community.

In the specific case of the viral family Mimiviridae, this viral component is strongly contributing (i.e. top 5% in the MG and top negative 5% in the MT) to S7 (**Supplementary Figure 10**). We

understand that these analyses culminate in data-driven hypotheses which we now refer to in the revised text:

The analysis of the causal network has to be considered as a tool to generate hypotheses on how the ecological events in the community have unfolded, utilising a data-driven approach allowed by the multi meta-omic angle of the experiment.

Loading plots. We thank the reviewer for picking up on our incorrect referencing to Supplementary Figure 6 in the text. The correct figures are now correctly referred to as **Supplementary Figure 10 and 11** in the revised version of the manuscript. We have also changed the references in the text accordingly (**Lines 349, 480**).

R2.C2 If the authors are proposing a generalised framework for data driven predictions then it is more likely to be adopted if it is described and explained with greater clarity. Some reference to general text books on SVD and data driven models might be helpful.

Response We agree with the reviewer’s suggestion to make the descriptions and explanations on the framework clearer. In this context, we have added the references to general books on linear and applied algebra that explain SVD, namely “Linear Algebra and Its Applications” [Strang 2006] and “Projection Matrices, Generalized Inverse Matrices, and Singular Value Decomposition” [Yana 2011] in the **Introduction**. Moreover, we introduced a new overview figure (Supplementary Figure 1) and the code is available at the github page: The code to reproduce the analysis is available at <https://github.com/fdelogu/microforecast>.

I have the following detailed comments to make.

Line 37 “maturity” I think the work maturity might be premature.

Response Word substituted with “chance”.

Line 127 “A week is the In” Is something missing here?

Response The stub of the sentence was removed.

Line 178 The reliance on Alter et al 2000 to introduce SVD is a weakness. It might be helpful to suggest one of the many excellent text books on data driven models.

Response We have now added the references to two useful textbooks in the **Introduction** section (see **R2.C2**).

Line 279 (and line 224) “ecological interactions” The authors may wish to mention work by Axel Rossberg (Nature Communications 12 (1), 1-11).

Response We thank the reviewer for their suggestion. We have added the reference to the introduction.

Line 405 “The signals .. according to them” I don’t understand this sentence.

Response In response to the reviewers’ comments, the entire paragraph has been revised to improve readability:

Following the forecasting of the signals, we decided to try to reconstruct the samples taken from the subsequent years. The samples’ information content can be expressed as a linear combination of the signals by creating a linear model using the training set and where the signals are the predictors. Using this approach, we computed how much each of the signals contributed to the samples (i.e. finding the betas of the model) and the basal abundance/expression (the intercept) of the samples. We decided to validate the approach using the gene abundance and expression values of the microbial families and reactions (KO term groups), therefore we fitted the linear models to those matrices from the training set and combined the results with the previously forecasted signals to reconstruct the test matrices. We then compared the reconstructed values with the original ones for each individual sample (Supplementary Figure 12).

Reviewer #3 (Remarks to the Author):

R3.C0 The manuscript submitted by Delogu et al., demonstrates a potentially generalizable framework to forecast microbial community ecology over time using a biological wastewater treatment plant (BWWTP) as a model ecosystem. This framework uses time-resolved metagenomics (metagenomic, metatranscriptomic, and metaproteomic) data related to taxonomic and functional information as well as environmental process parameter information to resolve “eigengenes” (EGs) for use in forecasting. The authors building on a pre-existing dataset first

30described in Herold and colleagues (Herold M, Martínez Arbas S, Narayanasamy S, Sheik AR, Kleine-Borgmann LAK, Lebrun LA, Kunath BJ, Roume H, Bessarab I, Williams RBH, Gillece JD, Schupp JM, Keim PS, Jäger C, Hoopmann MR, Moritz RL, Ye Y, Li S, Tang H, Heintz-Buschart A, May P, Muller EEL, Laczny CC, Wilmes P. Integration of time-series meta-omics data reveals how microbial ecosystems respond to disturbance. *Nat Commun.* 2020 Oct 19;11(1):5281. doi: 10.1038/s41467-020-19006-2. PMID: 33077707; PMCID: PMC7572474) combine the power of singular value decomposition (SVD), described by Alter and colleagues for genome-wide expression data, to reduce data complexity and identify patterns of related function or activity. These patterns can reflect both measured process parameters and omic information, as well as mathematical expressions e.g., factors resolved from the analysis.

After removing the first element or principal component, the authors tested for time-dependency and then applied a seasonal autoregressive integrated moving average (ARIMA) model implemented in fable resolving 210 EGs from 51 training data sets spanning one full cycle. These EGs were correlated with one another and then clustered into 17 signal groups considered to be non-redundant but temporally linked. A time resolved causality network was constructed revealing potential modules or “cliques” of temporally coupled ecological interactions (excluding one signal, S16 that did not appear coupled) that was used to develop specific hypothesis about BWWTP processes and events e.g., biological meaning. The authors explore two of these cliques with respect to viral lysis and predation respectively. Interestingly, the production of triacylglycerol (TG) and fatty acids (FA) by lipid accumulating organisms (LAOs) was not found to be time-dependent due in part to the redundancy of pathway components spread across many different taxonomic groups. The presence of genes involved in conversion of diacylglycerol and other LAO-related pathway components on plasmids suggested the potential for lateral gene transfer to play a role in generating this redundancy.

After exploring the biological meaning associated with a subset of the 17 signal groups identified in the training data, the authors turned their attention to forecasting future BWWTP processes and events. Three models (ARIMA, Prophet and neural networks) were testing with different hyperparameters implemented in fable using environmental variables as external regressors. An additional 21 samples were collected from the month of June from 2012-2016 to validate the model using metgenomic and metatranscriptomic data sets. Five of the signal groups were accurately predicted accounting for ~22% of the variance. The authors also attempted to predict gene abundance and expression patterns based on the 17 signal group matrices. To do this they constructed a linear model for the test data based on the signal group forecasting including weighted values and intercepts and compared the results to the original values for each sample.

This approach accurately predicted patterns of gene abundance and expression the following year but became much less accurate over the five-year test sampling interval.

Overall, this is an interesting paper that uses well established statistical methods in a creative way to model microbial ecology at the surface of a BWWTP. The use of SVD followed by correlation analysis, clustering and network mapping provides a useful framework for identifying temporal patterns in time-series data that may not be apparent when working with a single type of omics data. The authors do a good job exploring the biological meaning in relevant signal groups and forecasting efforts seem promising albeit with several limitations that remain to be addressed or reconciled.

Answer We thank the reviewer for the detailed summary and their constructive comments. We addressed them and integrated them in the revised version manuscript.

R3.C1 In the introduction, the authors refer to previous efforts to forecast compositional and gene expression dynamics in microbial communities and refer to the latter as an open challenge. In this light, it would be helpful to comment on prior work by Reed and colleagues (Reed DC, Algar CK, Huber JA, Dick GJ. Gene-centric approach to integrating environmental genomics and biogeochemical models. *Proc Natl Acad Sci U S A*. 2014 Feb 4;111(5):1879-84. doi: 10.1073/pnas.1313713111. Epub 2014 Jan 21. PMID: 24449851; PMCID: PMC3918765) and Louca and colleagues (Louca S, Hawley AK, Katsev S, Torres-Beltran M, Bhatia MP, Kheirandish S, Michiels CC, Capelle D, Lavik G, Doebeli M, Crowe SA, Hallam SJ. Integrating biogeochemistry with multiomic sequence information in a model oxygen minimum zone. *Proc Natl Acad Sci U S A*. 2016 Oct 4;113(40):E5925-E5933. doi: 10.1073/pnas.1602897113. Epub 2016 Sep 21. PMID: 27655888; PMCID: PMC5056048) that use gene-centric approaches to model microbial community gene expression at the level of transcripts and proteins. These methods would seemingly apply in the context of BWWTPs and should be considered when discussing an integrated framework for modeling community structure and function.

Answer We thank the reviewer for highlighting this prior work on gene-centric approaches to for modelling microbial community gene expression. To account for the reviewer's comment, we added some sentences and modified the ones already in the **Introduction** to add the aforementioned works. The revised text reads as follows (**Line 48**):

Whilst forecasting community composition dynamics has been successfully achieved for some environments (e.g. Larsen et al.⁴ and García-Jiménez et al.⁵) and explored

theoretically (Sullivan et al.⁶), the forecasting of gene expression dynamics over time in relation to environmental conditions remains an open challenge⁷. Although the work of Reed et al.⁸ and Louca et al.⁹ has approached the problem in marine systems with relatively stable environmental conditions such as in the oxygen minimum zone the challenge remains open due to the lack of a generalised framework that capitalises on the latest advances in meta-omics data and the systematic integration thereof over time.

R3.C2 With respect to the methods used, the authors are highly encouraged to provide more detail on their statistical and modeling workflows. As presented it would be difficult to reproduce their results even if provided the input matrices. More information is needed for reproducibility including how the hyperparameters were selected for the seasonal ARIMA, etc., and how the linear model was developed to reconstruct the expression patterns for the June test data. Inclusion of an actual workflow diagram for the modeling effort would be helpful and guide a reader through the methods and results with more clarity. The authors are also encouraged to provide more context for the data sets being used in this study. The 51 data sets used in training have been previously described by the authors but seemingly analyzed using a different workflow in the present study. For example, the present manuscript uses a different number of rMAGs from the prior study, but the criteria used for increased inclusivity is not clearly defined. Moreover, the current study uses GTDB taxonomy while the prior study used AMPHORA2 making direct comparison of the taxonomic groups identified extremely difficult. The authors should provide a more coherent and intertextual narrative linking this past manuscript to the present one in review.

Answer We acknowledge the reviewer's request for more details concerning the analysis workflows. Please find below a description of how we have addressed the individual aspects in the revised version of the manuscript:

Modelling details. A more detailed description of model selection, modelling and forecasting has been added to the revised version of the manuscript, more specifically in the method section (see **R1.C4** and **R1.C6**). The code for the analyses is available at <https://github.com/fdelogu/microforecast>.

ARIMA hyperparameters. A full description of the parameters for ARIMA and their selection has been added as well (see **R1.C6**). In brief, the parameters were automatically selected by the ARIMA function of the R package `fable` to better fit the data.

Linear modelling. A description of the linear model was provided in the methods section (see **R1.C6**). In brief, we used the `lm` function from the base package of R.

Workflow diagram. The new **Supplementary Figure 1** has been added to explain the analysis workflow (see **R1.C1**).

Re-analysis of previous data and context. The present study involved a reanalysis of the previous 51 samples while adding 21 rationally selected additional samples with corresponding meta-omics data analysed using the latest state-of-the-art tools, databases and strategies. The present study also included all the contigs longer than 1000nt in the analysis, not only the contigs included in the bins. Moreover, all the 72 samples were processed and analysed together to have a coherent dataset. To account for the reviewer's comment, we have now included additional sections in the revised manuscript to relate our present study to the previous work:

Taxonomic composition (**Line 174**):

The rMAGs spanned the expected phyla of the BWWTP community, and included member of the Actinobacteria, Bacteroidetes, Chlorobi, Fusobacteria, Nitrospirae, Proteobacteria and Spirochetes in addition to Candidatus Gracilibacteria (Figure 1a), thereby reproducing the results described in Herold et al. 2020⁴³.

Read recruitment (**Line 166**):

*Read recruitment (on the ORF level) per sample was on average 59% (s.d. 9%) for the MG, 82% (s.d. 3%) for the MT, whilst the peptide matching was 27% (s.d. 4%). The recruitment improved from the previous work, which reported the 26% (s.d. 3%) and 27% (s.d. 3%) of the MG and MT reads mapped against the MAGs, respectively⁴³. This is due to an update of the bioinformatics tools used and the inclusion of all the unbinned contigs longer than 1000nt in the analysis (see **Methods**).*

Genetic potential (**Line 160**):

However, the vast majority of the genes were not found to be expressed over the entire dataset or were only detected in a few samples whereby a maximum of 16.8×10^6 ORFs were detected in one sample. This suggests that a significant fraction of the gene pool in the LAOs is not specifically required for community function but rather their cumulative functional effort may be compartmentalised, fitting the previous results from Roume et

al.⁴⁴ showing how a large portion of the community is redundant, and only few functions are keystone.

Moreover, we clarified the analyses of the present study in the **Introduction (Line 115)**:

We reanalysed the data of a previously analysed time series spanning more than one year with 51 samples collected from March 2011 to May 2012 (which served as training set) together with 21 additional samples rationally selected and collected during the months of June for the years 2012-2016 (which we used for validation). From the samples we co-extracted [...]

and in the **Results (Line 142)**:

From the experimental period between 2011-03-21 and 2012-05-03 we previously obtained and analysed 51 weekly samples, to which we added 21 samples collected in the month of June during the years 2012-2016. The 72 samples were submitted to the same multi-omic analyses (MG, MT and MP) and processed individually to obtain 72 metagenomic assemblies, collections of metagenome-assembled genomes (MAGs), plasmids, viruses, unbinned prokaryotic chromosomal contigs and the corresponding gene expression at the transcriptional and proteomic levels. The combined datasets of the previous time series together with the new samples allowed the use of the enhanced bioinformatic strategies and allowed coherent comparison between the samples along the time series.

We have provided a more comprehensive description of model selection, modelling and forecasting in the methods section. We have added a detailed explanation of the ARIMA parameters as well as a description of the linear modelling. We have added a workflow diagram and added context and comparison with the previous works.

R3.C3 With respect to model development, the authors note in passing that their training data set did not span two complete cycles, and that they combined Fourier terms to compensate from this omission with respect to seasonality. This seems like a relatively important aspect of model development and needs to be better explained. For ARIMA models, the rule of thumb is that a minimum of 50 samples is needed and that more observations e.g., >100 are needed when dealing with seasonal components. This has implications for model development and potential overfitting. There is an emphatic statement that the model was robust for two years before significant decay in R2 values from linear regression of the training data. However, the training data ends on May of the same year that the first June data set is resolved. One might

35surmise that there is significant hysteresis in the system that would carry over for some time e.g., months before other factors could alter the trajectory of the system. Thus, from the standpoint of training and test data it is unclear why the authors did not include the June 2012 samples in their training data set or at least explain this proximity in time with respect to training and test data. Finally, and with respect to results interpretation, six signal groups were considered accurately predicted in the forecasting part of the work including S1, S2, S4, S5, S10 and S16. The training set for S10 was not fully captured by the model and therefore was excluded from analysis. It would be of interest to relate the remaining signal groups back to the network rendered in Figure 2C. In this case S16 was excluded because it did not demonstrate a temporal relationship with other groups making the total number of groups identified as significant with respect to the network equal to four (S1, S2, S4 and S5). Why were some groups identified from C1 and C2 but not others? What is the interpretation of S4 and S5? Beyond indicating that a model can be constructed and used to forecast BWWTP processes for a year or more how might this framework actually be used in a monitoring or process engineering context?

Answer We acknowledge the reviewer's detailed comments concerning the training and test data. In addition, we recognise the reviewer's comments on relating signal groups back to the overall network and on how the framework might be helpful in an applied sense. Please find below a description of how we have addressed the individual aspects in the revised version of the manuscript:

Fourier terms. We have added a brief explanation (**Line 729**) on the rationale for adding Fourier terms as exogenous regressors to explicitly provide a seasonal component to the model.

The Fourier transform can identify in a series of data the sum of sine and cosine waves underlying the data. In this way, if the period of time is correct (in this case one year), the Fourier terms can explicitly provide the seasonal part of the temporal behaviour. Using multiple terms allows for complex seasonal effects, whilst limiting the maximum number to 4 prevents from overfitting the data.

Minimum number of samples. The minimum number of samples depends on the complexity of the time series and the nature of the process. In general, the minimum requirement to use ARIMA modelling is 50 samples [**Box 2015**] but there are no precedent uses of this technique with omic data to refine the estimate. Considering that we addressed explicitly the seasonality in

the system using the Fourier term components, we believe that with 51 data points in the training set we hit the minimum requirements to use the model.

Separation between training and test data. The separation between the training set - ending in May 2012 - and the test set - starting in June 2012 - is not just temporal, but also technical. Indeed, the training and test sets belong to two different DNA/RNA extraction batches and sequencing campaigns which happened 6 years apart with different protocols and technologies (described in the **Sampling and preprocessing** section for the **Methods**). Although we explicitly addressed this in the “Batch effect correction” section of the Methods, we thought it would be interesting to keep the “technical” split in the data to test when multiple studies are performed targeting the same location in time with different experimental setups.

Network groups. The two groups C1 and C2 were selected because of their particular isolation with respect to the rest of the whole network, which potentially indicate the existence of two almost self-contained phenomena.

S4 and S5. The signals S4 and S5 have only strong taxonomic ties and no functional ones. In particular, S4 is explained only by seasonal components and associated with the predator family Nannocystaceae. S4 might play a role in the cyclical ecology of the system beyond the environmental variables (indeed we did not find any associated with it). On the other hand S5 shows no link to seasonality but is positively influenced by aeration. Among the taxa that contribute to it the most, there are families known to be involved in the bulking process such as Gordoniaceae and Zoogloeaceae. This points to a putative connection between the bulking process and the aeration, which is beyond seasonal effects.

We added the remark above in the manuscript on **Lines 354-359**.

Monitoring and process engineering. The current work forecasts a WWTP during its normal operations and it could be exploited to predict population and gene expression levels in the temporal medium range when knowing the environmental parameters. However, a potentially interesting development would be to test what happens when introducing “critical” values of the environmental parameters in the model to simulate an environmental disturbance. In order to reflect the reviewer’s comment, we have added these considerations to the **Conclusions** section.

Specific comments:

Introduction:

Line 104-105: Why the emphasis on LAO? A clear indication of the specific reason behind choosing to study a specific group of organisms is required in the introduction for readers less familiar with the system under study.

Answer We thank the reviewer for this comment. To account for it, we have now added a precise note on the LAO community in the introduction of the revised manuscript to explain the advantages related to the study of this particular community and the rationale of our choice to focus on it (**Lines 58-82**). In particular, essential considerations include its moderate richness (in addition, the microbial community biodiversity is medialallowing fairly comprehensive data acquisition and sampling depth). The LAO clear seasonal pattern (notably the surface community [**Frigon 2006**]) that has not been faithfully correlated with physico-chemical parameters other than temperature which currently represents an important bottleneck for the recovery of lipids for biodiesel production (In particular, the surface community is recognised to be a potential source of neutral lipids, a family of molecules with high added value for third generation biodiesel production [**Sheik 2014**]).

Line 107-108: Based on this sentence it seems that metaproteomic dataset was also generated by the authors in this study but under the methods section, they state that raw metaproteomic data was retrieved from PRIDE and reanalysed. It is unclear which of the two is correct. This relates back to the general comments above that that data sets used in this study were described in previous manuscripts and more intertextual linkages are needed.

Answer The metaproteomics data were previously generated as stated in line XXX:

The MP data from the PRIDE repository with accession number PXD01365538

And reanalysed with modern techniques as described in **Methods** section, paragraph “**MP quantification and filtering**”.

Line 114-115: It would be useful to also include a line or two on why the three specific models were selected, what other models exist for the same and what are their shortcomings. The latter aspect is really important when introducing time series analysis to new readers and speaks to points raised above about the number of samples used in training.

Answer We chose the models because of their inherent flexibility to customise them. The model are established means to explain complex time series breaking them down to the individual

38components (ARIMA and Prophet) or are powerful for many purposes (Neural Network). However, ARIMA assumes that the parameters behind the process are constant whilst Prophet can model a time-dependent evolution of the ARIMA parameters. On the other hand, ARIMA is the only model (in the current R package fable implementation), for which it is possible to obtain information about the contribution of the individual variables to the forecasting, making it suitable as an explicative model. In accordance with the reviewer's comments, we have now added these considerations to the text.

Line 116: Why was June chosen for future predictions? A line of two explaining this in the introduction would also be good. Have there been any interesting observations within the treatment plant previously during this month?

Answer The month of June was chosen because it is far from disruptive events (such as rain, snow and very cold temperatures) that are occurring in autumn and winter. In order to predict the behaviours of the community in these cases we would have needed a longer training set spanning multiple yearly cycles. This remark has been added at **Lines 407-408**.

Results and discussion:

Line 124-125: 51 metagenomic assemblies instead of genomic assemblies. In the same line they state a "collection of metagenome-assembled genomes" and don't mention exactly how many were obtained. Reword for clarity... "... we obtained 51 weekly samples used in the generation of MG, MT and MP data sets..." Provide a table describing these data with relevant features including sample type, number of reads, number of contigs, number of ORFs, Kos, number of peptides per sample, etc. These data have been previously published so this is a good place to elaborate on the connectivity...

Answer We have generated a new **Supplementary Table 1** with the requested information. Furthermore, we have diligently revised the text to ensure that all methodological aspects are transparently described (see **R3.C2**).

Line 126: Exact numbers of how many contigs were in bins, how many were plasmids, viruses, and were unbinned is required along with read mapping information on a per sample basis.

Answer In accordance with the reviewer's comment, this information has now been added through inclusion of **Supplementary Table 1** (see above).

Line 127 (Sentence formation error): "A week is the In order to..."

Answer Many thanks to the reviewer for picking this up. The sentence has now been fixed.

Line 128-130: Clustering was performed based on sequence identity but what was the identity percent cut off used isn't mentioned. The line points to the methods section but it isn't mentioned there either.

Answer We updated methods including the identity threshold in the following sentence (**Line 643**):

All the plasmidial, viral and the unbinned contigs from the eukaryotic and chromosomal prokaryotic subsets were clustered using CD-HIT and 90% identity threshold on each of those subsets.

Line 130-155: Another place to make connections to prior work.

Answer The current work and the ones previously published on the dataset are not directly comparable under many of the metrics described in the preset work because the previous analysis of the dataset only considered the MAGs, thereby not including a substantial fraction of unbinned data. In contrast to the previous work, the present study explicitly addresses a gene-centric analysis of the data including more sequences and more samples than before (see **R3.C2**).

Figure 1 caption: MAG completion and contamination isn't mentioned under results or the methods section. Based on the caption a completion of 75% and a contamination of maximum 25% is used as a cut-off to filter the MAGs. It is unclear why these specific cutoffs were used. The yellow color is missing from the legend.

Answer The evaluation metrics on the MAGs were computed using CheckM during the dereplication via dRep. To address the reviewer's comment, we have now added to the methods section the following text:

The resulting bins were dereplicated along the entire time series with dRep⁵⁹ to create representative metagenome assembled genomes (rMAGs) based on the results of CheckM⁶⁶ such as contamination and completeness (results for the rMAGs are shown in Figure 1a).

Figure 2: Would be helpful to use same colour coding in this figure as in Figure 1 for Families within a given Phylum level grouping (can shade the colours if more than one representative within a family).

Answer Figure 2 did not include information about taxa but Figure 3 did. To account for the reviewer's comment, we have modified panel b (included below) as requested. We had to extend the colour interval to improve the separation between families linked to the same phylum. Actinobacteria are represented in shades of yellow, Proteobacteria in shades of green while the Spirochaetes only include one family and we therefore used the same colour all of them. The categories of "not classified" and "plasmid" were given the colours grey and red, respectively.

We have also added the following text to the legend:

The families belonging to the same phylum have similar colours to matching phyla in Figure 1a. Therefore, Actinobacteria are in shades of yellow, Proteobacteria in shades of green while Leptospiraceae inherited the blue from the Spirochaetes.

Figure 2B: A clear description of the model panel including the ARIMA components on the x axis would help understand the figure and the results better.

Answer In accordance with the reviewer's comment, we have added a clearer description on the ARIMA components to the **Methods** section (see **R1.C4**)

Line 160-164: Why not adopt established community standards for high and medium quality MAGs described in Bowers and colleagues (Bowers RM, Kyrpides NC, Stepanauskas R, Harmon-Smith M, Doud D, Reddy TBK, Schulz F, Jarett J, Rivers AR, Eloie-Fadrosh EA, Tringe SG, Ivanova NN, Copeland A, Clum A, Becraft ED, Malmstrom RR, Birren B, Podar M, Bork P, Weinstock GM, Garrity GM, Dodsworth JA, Yooseph S, Sutton G, Glöckner FO, Gilbert JA, Nelson WC, Hallam SJ, Jungbluth SP, Etema TJG, Tighe S, Konstantinidis KT, Liu WT, Baker BJ, Rattei T, Eisen JA, Hedlund B, McMahon KD, Fierer N, Knight R, Finn R, Cochrane G, Karsch-Mizrachi I, Tyson GW, Rinke C; Genome Standards Consortium; Lapidus A, Meyer F, Yilmaz P, Parks DH, Eren AM, Schriml L, Banfield JF, Hugenholtz P, Woyke T. Minimum information about a single amplified genome (MISAG) and a metagenome-assembled genome (MIMAG) of bacteria and archaea. *Nat Biotechnol.* 2017 Aug 8;35(8):725-731. doi: 10.1038/nbt.3893. Erratum in: *Nat Biotechnol.* 2018 Feb 6;36(2):196. Erratum in: *Nat Biotechnol.* 2018 Jul 6;36(7):660. PMID: 28787424; PMCID: PMC6436528)?

Answer We thank the reviewer for this comment. We very much appreciate the paper by Bowers et al, however we used the MAGs solely to assign the taxonomy better to individual contigs (when possible). We therefore do not claim the MAGs presented in this work to belong to a "high quality" bracket. We reported the popular scores from CheckM [**Parks 2015**] because they are used in drep [**Olm 2017**] to inform the decision of which genomes were representative.

Line 187-189: Reword for clarity. I think you mean that given their independence the matrices will contain some degree of redundant information...

Answer We have rephrased the sentence to enhance the clarity and it now reads:

Considering that the EGs from the same matrix are linearly independent (i.e. they do not have redundant information), but since we pulled together the EGs from different types and levels of summarising matrices, we expect some level of redundancy in the information.

Line 301-307: What is the figure that summarizes this information?

Answer We referenced **Supplementary Figure 11** in the text.

Data and code availability:

Line 492: Zenodo is misspelled.

Answer Thank you. This has been fixed.

Materials and Methods:

Line 541: Oxygen is written twice.

Answer Thank you. This has been fixed.

Line 558: It would be nice to state the parameters rather than say custom parameters especially if someone is trying to exactly replicate the analysis.

Answer Although we agree in principle with the reviewer, to add all the parameters, including the default ones, for all the tools would very much clutter the manuscript text. We therefore preferred to upload the command lines and functions we used in the repository: https://git-r3lab.uni.lu/ESB/lao/lao_ts and <https://github.com/fdelogu/microforecast>. However, a sentence pointing to the repository has been added to the **Data and code availability** section of the manuscript.

The full list of software and R package versions are listed in the Git pages.

Line 578: What was the reason for using these three bidders specifically?

Answer MaxBin and MetaBAT are commonly used bidders that have proved to be among the best in benchmarking competitions [Yue 2020] while binny has proven to be highly accurate [Hickl 2022]. Moreover, it has been shown that combining the results using a refining tool such as DASTool or MetaWRAP leads to further better results than simply relying on a single tool [Yue 2020]. Therefore IMP uses MaxBin, MetaBAT and binny, while also applying DASTool as a refining tool to the results.

Line 590: It would be better to specify the different databases Mantis uses rather than mentioning that all databases were used.

Answer We agree with the reviewer's assertion and have added the following text to the manuscript:

[...] and annotated using Mantis v1.0263 with the heuristic approach and using kofam⁷⁰, tigrfam⁷¹, EGGNOG⁷², Pfam-A⁷³ and NCBI⁷⁴ all the databases.

Line 660: Performed three tests according to the text above but mentioned four tests here.

Answer The last test was performed two separate times with different hypotheses. We now specify this in the text:

[...] and two the Kwiatkowski–Phillips–Schmidt–Shin (kpss.tests) tests with null hypotheses “trend” and “level” respectively.

Line 692: (Sentence formation error): “test test with the function forecast form the fable package...”

Answer Thank you. We have now substituted “test test” by “test set”

Line 698: change tense “run” should be “ran”

Answer Thank you. This has been fixed.

Line 700: change tense “rebuild” should be “rebuilt or reconstructed”

Answer Thank you. This has now been fixed with “reconstructed”.

General comment: Some software used in this study have version numbers but majority of them are missing this key piece of information. Secondly, at places abbreviations are used without explaining the full form first. Eg. CAT and BAT used for taxonomic assignment.

Answer We uploaded the environments listing all the tools and versions in the repository: https://git-r3lab.uni.lu/ESB/lao/lao_ts and <https://github.com/fdeloqu/microforecast>. A sentence pointing to the repository has been added to the **Data and code availability** section of the manuscript.

The full list of software and R package versions is listed in the Git pages.

CAT and BAT are tools described in von Meijenfeldt et al. **[Meijenfeldt 2019]**, and stand for “Contig Annotation Tool” and “Bin Annotation Tool”. We have added the full names to the sentence introducing them in the revised manuscript.

Supplementary figure 9: caption broken up by the figure.

Answer Thank you. This has now been fixed.

Supplementary figure 2: reduce size of labels in red.

Answer Thank you. This has now been done.

Bibliography

[Alter 2000] Alter, O., Brown, P. O. & Botstein, D. Singular value decomposition for genome-Wide expression data processing and modeling. *Proc. Natl. Acad. Sci. U.S.A.* (2000).

[Box 2015] G. E. P. Box, et al., *Time Series Analysis: Forecasting and Control*, 5th Edition, Wiley (2015).

[Dray 2007] Dray, S., & Dufour, A.-B. (2007). The ade4 Package: Implementing the Duality Diagram for Ecologists. *Journal of Statistical Software*, 22(4), 1–20.

[Frigon 2006] Frigon D, Guthrie RM, Bachman GT, Royer J, Bailey B, Raskin L. Long-term analysis of a full-scale activated sludge wastewater treatment system exhibiting seasonal biological foaming. *Water Res.* 2006 Mar;40(5):990-1008.

[Hickl 2022] Hickl O, Queirós P, Wilmes P, May P, Heintz-Buschart A. binny: an automated binning algorithm to recover high-quality genomes from complex metagenomic datasets. *Brief Bioinform.* 2022 Nov 19;23(6):bbac431.

[Langfelder 2008a] Langfelder, P. & Horvath, S. WGCNA: an R package for weighted correlation network analysis. *BMC Bioinformatics* 9, 559 (2008).

[Langfelder 2008b] Langfelder P, Zhang B, Horvath S. Defining clusters from a hierarchical cluster tree: the Dynamic Tree Cut package for R. *Bioinformatics.* 2008 Mar 1;24(5):719-20. doi: 10.1093/bioinformatics/btm563. Epub 2007 Nov 16. PMID: 18024473.

[Meijenfeldt 2019] Von Meijenfeldt, F. A. B., Arkhipova, K., Cambuy, D. D., Coutinho, F. H. & Dutilh, B. E. Robust taxonomic classification of uncharted microbial sequences and bins with CAT and BAT. *Genome Biol.* (2019).

[O'Hara] O'Hara-Wild, M., 'Handyman, R. & 'Wang Earo'. fable: Forecasting Models for Tidy Time Series.

[Olm 2017] Olm, M. R., Brown, C. T., Brooks, B. & Banfield, J. F. dRep: a tool for fast and accurate genomic comparisons that enables improved genome recovery from metagenomes through de-replication. *ISME J.* 11, 2864–2868 (2017).

[Parks 2015] Parks, D. H., Imelfort, M., Skennerton, C. T., Hugenholtz, P. & Tyson, G. W. CheckM: assessing the quality of microbial genomes recovered from isolates, single cells, and metagenomes. *Genome Res.* 25, 1043–1055 (2015).

[Sheik 2014] Sheik, A. R., Muller, E. E. L. & Wilmes, P. A hundred years of activated sludge: time for a rethink. *Front. Microbiol.* 5, (2014)

[Strang 2006] Strang, G. *Linear Algebra and Its Applications.* SIAM Review (Cengage Learning, 2006).

[Yanai 2011] Yanai, H., Takeuchi, K. & Takane, Y. *Projection Matrices, Generalized Inverse Matrices, and Singular Value Decomposition.* (Springer New York, 2011).

[Ye 2019] Ye, H., 'Clark, A., 'Deyle, E. & 'Sugihara, G. rEDM: An R package for Empirical Dynamic Modeling and Convergent Cross Mapping. (2019).

[Yue 2020] Yue Y, Huang H, Qi Z, Dou HM, Liu XY, Han TF, Chen Y, Song XJ, Zhang YH, Tu J. Evaluating metagenomics tools for genome binning with real metagenomic datasets and CAMI datasets. *BMC Bioinformatics.* 2020 Jul 28;21(1):334.

Decision Letter, first revision:

9th August 2023

Dear Dr. Wilmes,

Thank you for submitting your revised manuscript "Forecasting of a complex microbial community using meta-omics" (NATECOLEVOL-221017808A). I apologize for the length of time to reach this decision, which occurred because we were waiting for one of the reviewers to submit a report. The revision has now been seen again by the three original reviewers and their comments are below. The

46reviewers find that the paper has improved in revision, and therefore we'll be happy in principle to publish it in Nature Ecology & Evolution, pending minor revisions to satisfy the reviewers' final requests and to comply with our editorial and formatting guidelines.

We are now performing detailed checks on your paper and will send you a checklist detailing our editorial and formatting requirements in 1-2 weeks. Please do not upload the final materials and make any revisions until you receive this additional information from us.

[REDACTED]

Reviewer #1 (Remarks to the Author):

I appreciate all the efforts made by the authors when revising the manuscript. They have thoroughly addressed all my concerns. In particular, it was very nice to see how the Convergent Cross Mapping analysis helped confirm the interactions between signals, now shown in panel c of Figure 2.

I have no further comments, and I can now recommend this work for publication in its present form.

Reviewer #2 (Remarks to the Author):

I was very grateful for the opportunity to revisit this very interesting and important manuscript. My requests for references to (excellent) explanatory texts have been met.

However, there are some typographical errors, often words concatenating, and the figures in the text do not align with the figures in supplementary methods.

This is very confusing for an ingenuer such as myself. I was wondering if a draft file had been submitted by accident. Even when I realised that there was an error it was difficult to unpick.

This is a shame and though it is a source of irritation, it did not obscure my impression that this is an interesting and important manuscript that could inspire others.

I have the following detailed comments.

Line 57 and through out "inmeta-omics". There are lots of these compound typos. I will not note them all.

Line 101. Reference 37 and 38 are great introductions to SVD and linear algebra. It might be helpful say so explicitly.

Line 161 “a significant proportion of the gene pool in the .. is not specifically required”. An alternative explanation is that a significant part of the gene expression fell below the detection limit. Bacteria don't tend to carry genetic unwanted material.

Line 212 The manuscript skips from supplementary figure 1 to supplementary figure 4. But figure 4 does not seem to relate to the text. Supplementary figure 5 would make more sense. A subsequent (line 707) reference to supplementary 1 probably refers to supp figure 2.

Line 223 “supplementary figure 5” is this supplementary figure 6.

Line 279 Supplementary figures 5 and 6. Are these the correct figures?

Line 303 Supplementary figure 9 is referred to, but figure 10 makes more sense.

Lines 276 to lines 305. There is very close reasoning here with references to details of figures that I thought I could follow if I managed to infer which figure is which.

Line 431 “We speculate”. This sounds plausible.

Line 455. This is the most exciting part of the work. But the confusion over the figures makes it very difficult to understand. I assume that figures 10 and 11 in the text are really figure 12. In which case I think it makes sense.

Reviewer #3 (Remarks to the Author):

The efforts of the authors to address the issues raised during the first round of review are greatly appreciated and have added more clarity to the work effort. Remaining edits and comments are primarily focused on making the manuscript more accessible to the reader. Line numbers are based on the updated version of the manuscript /18895_1_art_0_rv9vg9.docx. Note that there seem to be multiple grammatical and syntax errors cropping up in this new version that could reflect a problem with track changes (based on the marked-up .pdf version of the text).

Specific comments:

Line 28 Change to “and linked these [signals] over time...”

Line 29-30 Change to “to rebuild the sequence of ecological events, forecasting over five years using additional samples collected at defined time intervals.”

Lines 30-39 Change to “Using this information we were able to predict gene abundance and expression patterns over the same five-year period observing near perfect trend prediction (coefficient of determination ≥ 0.97) for the first two years. Moreover, we were able to correctly forecast five signals accounting for 22.5% of the time-dependent information in the system and generated

48mechanistic predictions on the ecological events in the community (e.g. a predation cycle involving bacteria, viruses and amoebas). Our study demonstrates the power of microbial ecology and statistical modeling to develop time variable predictions of community structure, function and activity spanning multi-year time series and provides an extensible framework when sufficient temporal signal and environmental parameter information exists.”

Lines 54-57 Change to “environmental conditions such as those associated with marine oxygen minimum zones using multi-omic (DNA, RNA, protein) and environmental parameter information to model biogeochemical cycles, a generalized framework for time-variable integration of multi-omic data sets into models of community ecology remains to be established.”

Line 58 Start new paragraph. “The surface community of ...”

Line 59-60 Change to “candidate to become a model system to establish such a modeling framework for the following three reasons.”

Line 67 Change to “BWWTP communities”

Line 69 Do you mean BWWTP here? It would be good to stick with a single acronym throughout the text unless you are differentiating systems for a specific reason.

Line 79-82 The authors write “while at the same time its functioning has to minimise the production of the greenhouse gases such as N₂O” Some WWTPs are specifically designed to include an AD component for bioenergy production in the form of methane biogas that can be upgraded to renewable natural gas. Can you please clarify this statement regarding greenhouse gas abatement or management in light of this comment?

Line 94 Change to “... enable the establishment of multi-omic sample-specific...”

Line 95 Start new paragraph. “When dealing with ...”

Line 95-96 Change to “When dealing with complex microbial communities...”

Line 97 Change to “To achieve this, we ...”

Line 102-103 Change to “the first matrix is associated with the set of temporal patterns underlying the data, and the second as the...”

Lines 111-112 Change to “Here we combine SVD and several time series algorithms into a generalizable framework for modelling the temporal dynamics of multi-layered meta-omics data.”

Lines 114-120 Change to “We demonstrate the power of this framework through analysis of multi-omic and environmental parameter datasets from a Lipid Accumulating Organisms (LAO) surface community from an anaerobic tank of the BWWTP in Schifflange (Luxembourg) using 51 time-resolved samples collected between March 2011 and May 2012 as part of a prior study⁴⁰ for training, with 21

49additional samples collected between 2012-2016 for validation and testing.”

Lines 133-135 “Validation was conducted...” This sentence is redundant with information provided in Lines 114-120.

Line 141 Why is the reference different here e.g., 43 from the one indicated above e.g., 40 to describe the previous work?

Line 148 What does “enhanced bioinformatic strategies” mean? Please explain. Would it not be easier to indicate that the combined dataset was reanalyzed using updated workflows for continuity of effort?

Line 160 Please explain if the 16.8×10^{16} ORFS are expressed. The current wording of this sentence is confusing.

Line 172 Change to “... community and included members...”

Line 185 Figure 1 caption. Earlier 144 rMAGs were indicated but in the legend only 125 are mentioned. Please explain.

Lines 204-212 Change to “The resulting 27 matrices (3 original and 24 summary) were used to compute the system’s eigengenes (EGs)³⁰. In previous work, the first EG in a time-series has been demonstrated to represent “steady state” gene expression, encapsulating the largest explained variance (EV). Therefore, the largest EG (average 50%, s.d. 22% in all the datasets) was removed from this analysis to capture only the time-dependent information resulting in “temporal EVs”³⁰.”

Note you have already explained some of the relevant information in the introduction allowing you to streamline this section of the text.

Lines 216-220 Change to “In order to reduce potential redundancy associated with the time resolved EGs identified across multiple data layers and bring together the same temporal behaviours, we clustered the set of 210 EGs into 17 representative EGs (see Methods).”

Note that the original sentence structure related to linear independence is not clearly rendered.

Line 230 Change to “self-dependence”

Line 266 Figure 2 caption “... hollow one represent an imbalance information transfer” Do you mean an imbalanced information transfer or an imbalance of information transfer?

Lines 278-280 See comment below regarding Lines 289-292.

Lines 286-287 Change to “In doing so we considered the generative processes and causal links of the signals which were applied to the 17 clusters.”

Line 289 Change to “... to microbial community structure and function.”

Lines 289-292 This sentence seems to be redundant with Lines 278-280. Consider removing and replacing with "The power of this representation is the amalgamation from the temporal signals, the loadings contributing to them (Supplementary Figures 5 and 6) and the generative model provided by ARIMA (Figure 2b) to generate ecological hypotheses to be further tested."

Line 380 Did you mean previously? "precedently" does not seem to be the correct word choice here...

Line 395 What you mean by "systemica information" here. Please reword for clarity.

Line 458 Change to "Using this approach we computed..."

Line 527 Do you mean BWWTP?

Lines 528-529 Change to "especially sampling the BWWTP at higher time frequencies..."

Line 531 Change to "...to cover those phenomena poorly constrained by the current model."

Lines 531-534 Please reword for clarity... "Finally, we infer that there are environmental drivers in the macroscopic composition of the LAO community behaviour and that we are able to correctly reconstruct the samples from 2 years in the future but that the foam presents a high islet variation which is beyond the predictability of this method."

There seem to be two separate ideas here that are in conflict, one related to the two-year model prediction and the other to future foaming events.

Line 540 Please consider a concluding statement about the extensibility of this approach and any specific caveats e.g., temporal resolution, validation, etc that should be factored in for others who might be interested in adopting this approach in the context of their own time-series efforts.

Lines 741-742 Change to "Considering we required a clustering approach that is independent of scale we computed the Pearson correlations between pairs of EGs..."

Lines 767-769 Change to "(ARIMA, Prophet and neural network using the functions ARIMA, prophet and NNETAR, respectively, all implemented in the R package fable..."

Line 770 Change to "Each signal was modelled as a separate..."

Line 780 Change to "set..."

Line 793 Change to "This was..."

Our ref: NATECOLEVOL-221017808A

25th August 2023

Dear Dr. Wilmes,

Thank you for your patience as we've prepared the guidelines for final submission of your Nature Ecology & Evolution manuscript, "Forecasting of a complex microbial community using meta-omics" (NATECOLEVOL-221017808A). Please carefully follow the step-by-step instructions provided in the attached file, and add a response in each row of the table to indicate the changes that you have made. Please also check and comment on any additional marked-up edits we have proposed within the text. Ensuring that each point is addressed will help to ensure that your revised manuscript can be swiftly handed over to our production team.

****We would like to start working on your revised paper, with all of the requested files and forms, as soon as possible (preferably within two weeks). Please get in contact with us immediately if you anticipate it taking more than two weeks to submit these revised files.****

In recognition of the time and expertise our reviewers provide to Nature Ecology & Evolution's editorial process, we would like to formally acknowledge their contribution to the external peer review of your manuscript entitled "Forecasting of a complex microbial community using meta-omics". For those reviewers who give their assent, we will be publishing their names alongside the published article.

Nature Ecology & Evolution offers a Transparent Peer Review option for new original research manuscripts submitted after December 1st, 2019. As part of this initiative, we encourage our authors to support increased transparency into the peer review process by agreeing to have the reviewer comments, author rebuttal letters, and editorial decision letters published as a Supplementary item. When you submit your final files please clearly state in your cover letter whether or not you would like to participate in this initiative. Please note that failure to state your preference will result in delays in accepting your manuscript for publication.

52Cover suggestions

We welcome submissions of artwork for consideration for our cover. For more information, please see our [guide for cover artwork](https://www.nature.com/documents/Nature_covers_author_guide.pdf).

Nature Ecology & Evolution has now transitioned to a unified Rights Collection system which will allow our Author Services team to quickly and easily collect the rights and permissions required to publish your work. Approximately 10 days after your paper is formally accepted, you will receive an email in providing you with a link to complete the grant of rights. If your paper is eligible for Open Access, our Author Services team will also be in touch regarding any additional information that may be required to arrange payment for your article.

Please note that *Nature Ecology & Evolution* is a Transformative Journal (TJ). Authors may publish their research with us through the traditional subscription access route or make their paper immediately open access through payment of an article-processing charge (APC). Authors will not be required to make a final decision about access to their article until it has been accepted. [Find out more about Transformative Journals](https://www.springernature.com/gp/open-research/transformative-journals)

Authors may need to take specific actions to achieve [compliance with funder and institutional open access mandates](https://www.springernature.com/gp/open-research/funding/policy-compliance-faqs). If your research is supported by a funder that requires immediate open access (e.g. according to [Plan S principles](https://www.springernature.com/gp/open-research/plan-s-compliance)) then you should select the gold OA route, and we will direct you to the compliant route where possible. For authors selecting the subscription publication route, the journal's standard licensing terms will need to be accepted, including [self-archiving-and-license-to-publish](https://www.nature.com/nature-portfolio/editorial-policies/self-archiving-and-license-to-publish). Those licensing terms will supersede any other terms that the author or any third party may assert apply to any version of the manuscript.

[REDACTED]

[REDACTED]

Reviewer #1:

Remarks to the Author:

I appreciate all the efforts made by the authors when revising the manuscript. They have thoroughly addressed all my concerns. In particular, it was very nice to see how the Convergent Cross Mapping analysis helped confirm the interactions between signals, now shown in panel c of Figure 2.

I have no further comments, and I can now recommend this work for publication in its present form.

Reviewer #2:

Remarks to the Author:

I was very grateful for the opportunity to revisit this very interesting and important manuscript. My requests for references to (excellent) explanatory texts have been met.

However, there are some typographical errors, often words concatenating, and the figures in the text do not align with the figures in supplementary methods.

This is very confusing for an ingenuer such as myself. I was wondering if a draft file had been submitted by accident. Even when I realised that there was an error it was difficult to unpick.

This is a shame and though it is a source of irritation, it did not obscure my impression that this is an interesting and important manuscript that could inspire others.

I have the following detailed comments.

Line 57 and through out "inmeta-omics". There are lots of these compound typos. I will not note them all.

Line 101. Reference 37 and 38 are great introductions to SVD and linear algebra. It might be helpful say so explicitly.

Line 161 "a significant proportion of the gene pool in the .. is not specifically required". An alternative explanation is that a significant part of the gene expression fell below the detection limit. Bacteria don't tend to carry genetic unwanted material.

54Line 212 The manuscript skips from supplementary figure 1 to supplementary figure 4. But figure 4 does not seem to relate to the text. Supplementary figure 5 would make more sense. A subsequent (line 707) reference to supplementary 1 probably refers to supp figure 2.

Line 223 "supplementary figure 5" is this supplementary figure 6.

Line 279 Supplementary figures 5 and 6. Are these the correct figures?

Line 303 Supplementary figure 9 is referred to, but figure 10 makes more sense.

Lines 276 to lines 305. There is very close reasoning here with references to details of figures that I thought I could follow if I managed to infer which figure is which.

Line 431 "We speculate". This sounds plausible.

Line 455. This is the most exciting part of the work. But the confusion over the figures makes it very difficult to understand. I assume that figures 10 and 11 in the text are really figure 12. In which case I think it makes sense.

Reviewer #3:

Remarks to the Author:

The efforts of the authors to address the issues raised during the first round of review are greatly appreciated and have added more clarity to the work effort. Remaining edits and comments are primarily focused on making the manuscript more accessible to the reader. Line numbers are based on the updated version of the manuscript /18895_1_art_0_rv9vg9.docx. Note that there seem to be multiple grammatical and syntax errors cropping up in this new version that could reflect a problem with track changes (based on the marked-up .pdf version of the text).

Specific comments:

Line 28 Change to "and linked these [signals] over time..."

Line 29-30 Change to "to rebuild the sequence of ecological events, forecasting over five years using additional samples collected at defined time intervals."

Lines 30-39 Change to "Using this information we were able to predict gene abundance and expression patterns over the same five-year period observing near perfect trend prediction (coefficient of determination ≥ 0.97) for the first two years. Moreover, we were able to correctly forecast five signals accounting for 22.5% of the time-dependent information in the system and generated mechanistic predictions on the ecological events in the community (e.g. a predation cycle involving bacteria, viruses and amoebas). Our study demonstrates the power of microbial ecology and statistical modeling to develop time variable predictions of community structure, function and activity spanning multi-year time series and provides an extensible framework when sufficient temporal signal and

55environmental parameter information exists.”

Lines 54-57 Change to “environmental conditions such as those associated with marine oxygen minimum zones using multi-omic (DNA, RNA, protein) and environmental parameter information to model biogeochemical cycles, a generalized framework for time-variable integration of multi-omic data sets into models of community ecology remains to be established.”

Line 58 Start new paragraph. “The surface community of ...”

Line 59-60 Change to “candidate to become a model system to establish such a modeling framework for the following three reasons.”

Line 67 Change to “BWWTP communities”

Line 69 Do you mean BWWTP here? It would be good to stick with a single acronym throughout the text unless you are differentiating systems for a specific reason.

Line 79-82 The authors write “while at the same time its functioning has to minimise the production of the greenhouse gases such as N₂O” Some WWTPs are specifically designed to include an AD component for bioenergy production in the form of methane biogas that can be upgraded to renewable natural gas. Can you please clarify this statement regarding greenhouse gas abatement or management in light of this comment?

Line 94 Change to “... enable the establishment of multi-omic sample-specific...”

Line 95 Start new paragraph. “When dealing with ...”

Line 95-96 Change to “When dealing with complex microbial communities...”

Line 97 Change to “To achieve this, we ...”

Line 102-103 Change to “the first matrix is associated with the set of temporal patterns underlying the data, and the second as the...”

Lines 111-112 Change to “Here we combine SVD and several time series algorithms into a generalizable framework for modelling the temporal dynamics of multi-layered meta-omics data.”

Lines 114-120 Change to “We demonstrate the power of this framework through analysis of multi-omic and environmental parameter datasets from a Lipid Accumulating Organisms (LAO) surface community from an anaerobic tank of the BWWTP in Schifflange (Luxembourg) using 51 time-resolved samples collected between March 2011 and May 2012 as part of a prior study⁴⁰ for training, with 21 additional samples collected between 2012-2016 for validation and testing.”

Lines 133-135 “Validation was conducted...” This sentence is redundant with information provided in Lines 114-120.

Line 141 Why is the reference different here e.g., 43 from the one indicated above e.g., 40 to describe the previous work?

Line 148 What does “enhanced bioinformatic strategies” mean? Please explain. Would it not be easier to indicate that the combined dataset was reanalyzed using updated workflows for continuity of effort?

Line 160 Please explain if the 16.8X10¹⁶ ORFS are expressed. The current wording of this sentence is confusing.

Line 172 Change to “... community and included members...”

Line 185 Figure 1 caption. Earlier 144 rMAGs were indicated but in the legend only 125 are mentioned. Please explain.

Lines 204-212 Change to “The resulting 27 matrices (3 original and 24 summary) were used to compute the system’s eigengenes (EGs)³⁰. In previous work, the first EG in a time-series has been demonstrated to represent “steady state” gene expression, encapsulating the largest explained variance (EV). Therefore, the largest EG (average 50%, s.d. 22% in all the datasets) was removed from this analysis to capture only the time-dependent information resulting in “temporal EVs”³⁰.

Note you have already explained some of the relevant information in the introduction allowing you to streamline this section of the text.

Lines 216-220 Change to “In order to reduce potential redundancy associated with the time resolved EGs identified across multiple data layers and bring together the same temporal behaviours, we clustered the set of 210 EGs into 17 representative EGs (see Methods).”

Note that the original sentence structure related to linear independence is not clearly rendered.

Line 230 Change to “self-dependence”

Line 266 Figure 2 caption “... hollow one represent an imbalance information transfer” Do you mean an imbalanced information transfer or an imbalance of information transfer?

Lines 278-280 See comment below regarding Lines 289-292.

Lines 286-287 Change to “In doing so we considered the generative processes and causal links of the signals which were applied to the 17 clusters.”

Line 289 Change to “... to microbial community structure and function.”

Lines 289-292 This sentence seems to be redundant with Lines 278-280. Consider removing and replacing with “The power of this representation is the amalgamation from the temporal signals, the loadings contributing to them (Supplementary Figures 5 and 6) and the generative model provided by

57ARIMA (Figure 2b) to generate ecological hypotheses to be further tested.”

Line 380 Did you mean previously? “precedently” does not seem to be the correct word choice here...

Line 395 What you mean by “systemica information” here. Please reword for clarity.

Line 458 Change to “Using this approach we computed...”

Line 527 Do you mean BWWTP?

Lines 528-529 Change to “especially sampling the BWWTP at higher time frequencies...”

Line 531 Change to “...to cover those phenomena poorly constrained by the current model.”

Lines 531-534 Please reword for clarity... “Finally, we infer that there are environmental drivers in the macroscopic composition of the LAO community behaviour and that we are able to correctly reconstruct the samples from 2 years in the future but that the foam presents a high islet variation which is beyond the predictability of this method.”

There seem to be two separate ideas here that are in conflict, one related to the two-year model prediction and the other to future foaming events.

Line 540 Please consider a concluding statement about the extensibility of this approach and any specific caveats e.g., temporal resolution, validation, etc that should be factored in for others who might be interested in adopting this approach in the context of their own time-series efforts.

Lines 741-742 Change to “Considering we required a clustering approach that is independent of scale we computed the Pearson correlations between pairs of EGs...”

Lines 767-769 Change to “(ARIMA, Prophet and neural network using the functions ARIMA, prophet and NNETAR, respectively, all implemented in the R package fable...”

Line 770 Change to “Each signal was modelled as a separate...”

Line 780 Change to “set...”

Line 793 Change to “This was...”

Author Rebuttal, first revision:

Revision letter for the manuscript NATECOLEVOL-221017808A

We would like to thank the reviewers for their contributions and insightful feedback on our paper. Their comments improved the quality of our work. We integrated the following points in the latest version of the manuscript.

The review has been broken down and the individual comments addressed. The comments have been tagged as **RX.CX**, where R stands for “Reviewer”, C for “Comment” and the Xs their respective number. When multiple comments targeted the same aspect we cross-referenced the comments using their **RX.CX** tags.

Reviewer #1 (Remarks to the Author):

R1.C0 I appreciate all the efforts made by the authors when revising the manuscript. They have thoroughly addressed all my concerns. In particular, it was very nice to see how the Convergent Cross Mapping analysis helped confirm the interactions between signals, now shown in panel c of Figure 2.

I have no further comments, and I can now recommend this work for publication in its present form.

Response We thank the reviewer for their assessment and especially for previously pointing out to the Convergent Cross Mapping analysis to strengthen our work.

Reviewer #2 (Remarks to the Author):

R2.C0 I was very grateful for the opportunity to revisit this very interesting and important manuscript. My requests for references to (excellent) explanatory texts have been met.

However, there are some typographical errors, often words concatenating, and the figures in the text do not align with the figures in supplementary methods.

This is very confusing for an ingenuer such as myself. I was wondering if a draft file had been submitted by accident. Even when I realised that there was an error it was difficult to unpick.

This is a shame and though it is a source of irritation, it did not obscure my impression that this is an interesting and important manuscript that could inspire others.

Response We thank the reviewer for helping us in providing further context and clarity to the methodological aspects of our work. Moreover, we apologise for the typographic mistakes, we have fixed them in the newer version.

I have the following detailed comments.

Line 57 and through out “inmeta-omics”. There are lots of these compound typos. I will not note them all.

Response We corrected as “in the meta-omics”.

Line 101. Reference 37 and 38 are great introductions to SVD and linear algebra. It might be helpful say so explicitly.

Response We modified the sentence to be more explicit:

SVD can decompose a matrix into two separated matrices of eigenvectors and a vector of eigenvalues (the technique can be further explored in the texts by Strang 2006³⁷ and Yanai et al. 2011³⁸).

Line 161 “a significant proportion of the gene pool in the .. is not specifically required”. An alternative explanation is that a significant part of the gene expression fell below the detection limit. Bacteria don't tend to carry genetic unwanted material.

Response We thank the reviewer for pointing this out. We added the following interpretation in the text:

This suggests that a significant fraction of the gene pool in the LAOs is not specifically required for community function or their expression levels are below the detection limit, hinting that their cumulative functional effort may be compartmentalised, supporting the previous results from Roume et al.⁴⁴ showing how a large portion of the community is redundant, and only few functions are keystone.

Line 212 The manuscript skips from supplementary figure 1 to supplementary figure 4. But figure 4 does not seem to relate to the text. Supplementary figure 5 would make more sense. A subsequent (line 707) reference to supplementary 1 probably refers to supp figure 2.

Response We corrected the references to **Supplementary Figure 1** and **2c**, respectively.

Line 223 “supplementary figure 5” is this supplementary figure 6.

Response We corrected the reference to the new **Supplementary Figure 3**.

Line 279 Supplementary figures 5 and 6. Are these the correct figures?

Response We corrected the references to new **Supplementary Figure 7** and **8**, respectively.

Line 303 Supplementary figure 9 is referred to, but figure 10 makes more sense.

Response We corrected the references to **Supplementary Figure 9** to **Supplementary Figure 10**.

Lines 276 to lines 305. There is very close reasoning here with references to details of figures that I thought I could follow if I managed to infer which figure is which.

Response We corrected the numbering of the figures to the two relevant ones for this paragraph, i.e., new **Supplementary Figure 7** and **8**.

Line 431 “We speculate”. This sounds plausible.

Response We agree with the reviewer, however, we believe that the meta-omics alone can be used only to back up the hypothesis of the triangular relationship but not fully prove it. We hope that in the future this relationship will be explored with complementary techniques such as microscopy and co-culturing. We added the following sentence to the manuscript:

The integrated meta-omics data should be supported in the future by complementary techniques such as microscopy and co-culturing to confirm this hypothesis.

Line 455. This is the most exciting part of the work. But the confusion over the figures makes it very difficult to understand. I assume that figures 10 and 11 in the text are really figure 12. In which case I think it makes sense.

Response We corrected the references to the new **Supplementary Figure 10** accordingly.

Reviewer #3 (Remarks to the Author):

62R3.C0 The efforts of the authors to address the issues raised during the first round of review are greatly appreciated and have added more clarity to the work effort. Remaining edits and comments are primarily focused on making the manuscript more accessible to the reader. Line numbers are based on the updated version of the manuscript /18895_1_art_0_rv9vg9.docx. Note that there seem to be multiple grammatical and syntax errors cropping up in this new version that could reflect a problem with track changes (based on the marked-up .pdf version of the text).

Response We very much thank the reviewer for their assessment and detailed comments. Indeed there was an issue with the exporting of the document, we have integrated the highlighted points to address this.

Specific comments:

Line 28 Change to “and linked these [signals] over time...”

Response We rewrote completely the abstract to fit within the 150 words limit. Moreover, we incorporated the indication and the new sentence reads as:

Here we summarise 14 months of longitudinal meta-omics data from a BWWTP anaerobic tank into 17 temporal signals, explaining 91.1% of the temporal variance, and linked those signals to ecological events within the community.

Line 29-30 Change to “to rebuild the sequence of ecological events, forecasting over five years using additional samples collected at defined time intervals.”

Response We included the detail in the new abstract. The sentence now reads as:

We forecast the signals over the five subsequent years and use 21 extra samples collected at defined time intervals for testing and validation.

Lines 30-39 Change to “Using this information we were able to predict gene abundance and expression patterns over the same five-year period observing near perfect trend prediction (coefficient of determination ≥ 0.97) for the first two years. Moreover, we were able to correctly forecast five signals accounting for 22.5% of the time-dependent information in the system and generated mechanistic predictions on the ecological events in the community (e.g. a predation cycle involving bacteria, viruses and amoebas). Our study demonstrates the power of microbial ecology and statistical modeling to develop time variable predictions of community structure, function and activity spanning multi-year time series and provides an extensible framework when sufficient temporal signal and environmental parameter information exists.”

Response We had to cut down on the word count but tried to include the indicated changes. The corresponding sentences of the abstract now read as follow:

Our forecasts are correct for six signals and hint on phenomena such as predation cycles. Using all the 17 forecasts and the environmental variables, we predict gene abundance and expression, with a coefficient of determination ≥ 0.87 for the first subsequent three years. Our study demonstrates the ability to forecast the dynamics of open microbial ecosystems using interactions between community cycles and environmental parameters.

Lines 54-57 Change to “environmental conditions such as those associated with marine oxygen minimum zones using multi-omic (DNA, RNA, protein) and environmental parameter information to model biogeochemical cycles, a generalized framework for time-variable integration of multi-omic data sets into models of community ecology remains to be established.”

Response We have modified the text accordingly using the wording “meta-omics” instead of “multi-omics”.

Line 58 Start new paragraph. “The surface community of ...”

Response We have separated paragraphs as indicated.

Line 59-60 Change to “candidate to become a model system to establish such a modeling framework for the following three reasons.”

Response We have modified the text accordingly.

Line 67 Change to “BWWTP communities”

Response We have edited the manuscript as suggested.

Line 69 Do you mean BWWTP here? It would be good to stick with a single acronym throughout the text unless you are differentiating systems for a specific reason.

Response Yes, we mean BWWTP. We have modified the other instances where it was reported as “WWTP” to “BWWTP”.

Line 79-82 The authors write “while at the same time its functioning has to minimise the production of the greenhouse gases such as N₂O” Some WWTPs are specifically designed to include an AD component for bioenergy production in the form of methane biogas that can be upgraded to renewable natural gas. Can you please clarify this statement regarding greenhouse gas abatement or management in light of this comment?

Response In this context, we specifically meant the unwanted production (and uncontrolled release) of greenhouse gases. We amended the text accordingly and now it reads as follows:

65Finally, forecasting the behaviour of microbial communities in BWWTPs is highly desirable as stable operation allows reclamation of clean water as well as the harnessing of chemical energy²⁰ while at the same time its functioning has to minimise the undesired production (and uncontrolled release) of the greenhouse gases such as N₂O²¹.

Line 94 Change to "... enable the establishment of multi-omic sample-specific..."

Response We have modified the text accordingly using the wording "meta-omics" instead of "multi-omics".

Line 95 Start new paragraph. "When dealing with ..."

Response We have edited the manuscript as required.

Line 95-96 Change to "When dealing with complex microbial communities..."

Response We have changed the text accordingly.

Line 97 Change to "To achieve this, we ..."

Response We have added the comma.

Line 102-103 Change to "the first matrix is associated with the set of temporal patterns underlying the data, and the second as the..."

Response We have changed the sentence accordingly.

Lines 111-112 Change to “Here we combine SVD and several time series algorithms into a generalizable framework for modelling the temporal dynamics of multi-layered meta-omics data.”

Response We have modified the text accordingly.

Lines 114-120 Change to “We demonstrate the power of this framework through analysis of multi-omic and environmental parameter datasets from a Lipid Accumulating Organisms (LAO) surface community from an anaerobic tank of the BWWTP in Schiffflange (Luxembourg) using 51 time-resolved samples collected between March 2011 and May 2012 as part of a prior study⁴⁰ for training, with 21 additional samples collected between 2012-2016 for validation and testing.”

Response We have edited the manuscript to read as follows:

We demonstrate the power of this framework through analysis of integrated meta-omics and environmental parameter datasets from a microbial community enriched in lipid accumulating organisms (LAOs) on the surface of the anaerobic tank of the BWWTP in Schiffflange (Luxembourg). The sample set comprises of 51 time-resolved samples collected between March 2011 and May 2012 for training, with 21 additional samples collected between 2012-2016 for testing and validation. For both sets the biomolecules were co-extracted⁴⁰ and the data for the training set were presented in a previous study⁴³.

Lines 133-135 “Validation was conducted...” This sentence is redundant with information provided in Lines 114-120.

Response We removed the sentence.

Line 141 Why is the reference different here e.g., 43 from the one indicated above e.g., 40 to describe the previous work?

Response Reference 40 (Roume et al., 2013) describes the co-extraction of the macromolecules from the same sample, whilst reference 43 (Herold et al., 2020) is where the data for the period between March 2011 and May 2012 were presented.

Line 148 What does “enhanced bioinformatic strategies” mean? Please explain. Would it not be easier to indicate that the combined dataset was reanalyzed using updated workflows for continuity of effort?

Response We integrated the suggestion and the sentence now reads as follows:

The combined datasets of the previous time series alongside the new samples were analysed together with updated bioinformatic workflows to allow a coherent comparison between samples along the time series.

Line 160 Please explain if the 16.8X10¹⁶ ORFS are expressed. The current wording of this sentence is confusing.

Response We referred to ORFs detected at the DNA level, we realize that this was confusing and we removed the second part of the sentence which now reads:

However, the vast majority of the genes were not found to be expressed over the entire dataset or were only detected in a few samples.

Line 172 Change to “... community and included members...”

Response We removed the comma.

Line 185 Figure 1 caption. Earlier 144 rMAGs were indicated but in the legend only 125 are mentioned. Please explain.

Response We thank the reviewer for spotting this incoherence. We have a total of 144 MAGs, which break down to 126 and 18 bacterial and archaeal ones, respectively, as we reported in the **Supplementary Table 2**. We corrected the sentence in the caption of **Figure 1** and added a note to the 18 archaeal MAGs not being included in the figure. The sentence now reads as follows:

The phylogenetic tree of the rMAGs in LAO (generated using gtdb-tk⁴⁷) contains the 126 bacterial rMAGs in the system (the 18 archaeal MAGs were not included).

Lines 204-212 Change to “The resulting 27 matrices (3 original and 24 summary) were used to compute the system’s eigengenes (EGs)³⁰. In previous work, the first EG in a time-series has been demonstrated to represent “steady state” gene expression, encapsulating the largest explained variance (EV). Therefore, the largest EG (average 50%, s.d. 22% in all the datasets) was removed from this analysis to capture only the time-dependent information resulting in “temporal EVs”³⁰.”

Note you have already explained some of the relevant information in the introduction allowing you to streamline this section of the text.

Response We thank the reviewer for pointing this out. We amended the whole paragraph and it reads now as follows:

The resulting 27 matrices (3 original and 24 summary) were used to compute the system’s eigengenes (EGs)³⁰. In previous work, the first EG in a time-series was shown

69to represent “steady state” gene expression, encapsulating the largest explained variance (EV). Therefore, the first EG (average EV 50%, s.d. 22% in all the datasets) was removed. We screened the subsequent EGs for time-dependency (see Methods) selecting a set of 210 EGs and assessed how much of the EV they explained (Supplementary Figure 2d).

Lines 216-220 Change to “In order to reduce potential redundancy associated with the time resolved EGs identified across multiple data layers and bring together the same temporal behaviours, we clustered the set of 210 EGs into 17 representative EGs (see Methods). The largest EG (average 50%, s.d. 22% in all the datasets) was removed from this analysis to capture only the time-dependent information resulting in “temporal EVs” 30”

Note that the original sentence structure related to linear independence is not clearly rendered.

Response We have replaced the following part of the text:

Considering that the EGs from the same matrix are linearly independent (i.e. they do not have redundant information), but since we pulled together the EGs from different used types and levels of summarising matrices, we expect some level of redundancy in the information. In order to reduce this redundancy and bring together the same temporal behaviours, we clustered the set of 210 EGs into 17 representative EGs (see Methods).

With the suggested sentence:

In order to reduce potential redundancy associated with the time-resolved EGs identified across multiple data layers and bring together the same temporal behaviours, we clustered the set of 210 EGs into 17 representative EGs (see Methods).

Line 230 Change to “self-dependence”

Response We introduced the suggested hyphen.

Line 266 Figure 2 caption "... hollow one represent an imbalance information transfer" Do you mean an imbalanced information transfer or an imbalance of information transfer?

Response We meant an imbalance in information transfer, we specify this now and the sentence now reads as follows:

The dashed lines indicate weak transfer of information whilst a full arrow and a hollow one represent an imbalance in information transfer (in favor of the solid arrow).

Lines 278-280 See comment below regarding Lines 289-292.

Response See response to point below.

Lines 286-287 Change to "In doing so we considered the generative processes and causal links of the signals which were applied to the 17 clusters."

Response We have edited the manuscript as specified.

Line 289 Change to "... to microbial community structure and function."

Response We have changed the text accordingly.

Lines 289-292 This sentence seems to be redundant with Lines 278-280. Consider removing and replacing with "The power of this representation is the amalgamation from the temporal

signals, the loadings contributing to them (Supplementary Figures 5 and 6) and the generative model provided by ARIMA (Figure 2b) to generate ecological hypotheses to be further tested.”

Response We moved the sentence in lines 278-280 in the second block of the paragraph. We believe that the sentences convey how to link the three parts of the analysis (eigengenes/loadings, ARIMA and causality network) and it is better to keep both of them. Consequently, the paragraph now reads as follows:

Incidentally, all signals but S16, demonstrated a temporal relationship with at least another signal, resulting in a single network of causality. We decided to focus on two particular cliques of nodes in the network (Figure 2c) to explore the ecological domino effect: C1 (including S1, S10 and S17) and C2 (S9, S4, S7, S8). In order to explore the ecological and environmental aspects of the system we recalled the two-way relationship between the signals and the other eigengenes they clustered with. In doing so, we considered the generative processes and causal links of the signals which were applied to the 17 clusters. In this way it was possible to use the top/bottom loadings of the EGs to link the high-level depiction of the system (the signals) to microbial community structure and function. The power of this representation is the amalgamation from the temporal signals, the loadings contributing to them (Supplementary Figures 7 and 8) and the generative model provided by ARIMA (Figure 2b) to generate ecological hypotheses to be further tested. The analysis of the causal network has to be considered as a tool to generate hypotheses on how the ecological events in the community have unfolded, utilising a data-driven approach allowed by the multi-layered meta-omic angle of the experiment.

Line 380 Did you mean previously? “precedently” does not seem to be the correct word choice here...

Response Yes, we changed the word to “previously”.

Line 395 What you mean by “systemica information” here. Please reword for clarity.

Response We corrected the typo and precised what information should be required. The sentence now reads:

To forecast such events, experimental information (such as the one derived from co-culturing) on microbial interactions would be required, which is beyond the scope of this study.

Line 458 Change to “Using this approach we computed...”

Response We have modified the text accordingly.

Line 527 Do you mean BWWTP?

Response Yes, we have corrected the acronym to be always “BWWTP”.

Lines 528-529 Change to “especially sampling the BWWTP at higher time frequencies...”

Response We have edited the text accordingly.

Line 531 Change to “...to cover those phenomena poorly constrained by the current model.”

Response We have modified the draft as indicated.

Lines 531-534 Please reword for clarity... “Finally, we infer that there are environmental drivers in the macroscopic composition of the LAO community behavior and that we are able to correctly reconstruct the samples from 2 years in the future but that the foam presents a high islet variation which is beyond the predictability of this method.”

There seem to be two separate ideas here that are in conflict, one related to the two-year model prediction and the other to future foaming events.

Response We separated the sentence into two and reworded it to enhance clarity. The text now reads as follows:

Finally, we infer that there are environmental drivers in the macroscopic composition of the LAO community behaviour and that we are able to correctly reconstruct the samples from three years into the future when averaged over a month period. However, we infer also that the community exhibits a high degree of variation, making the prediction of a specific sample inaccurate with this method.

Line 540 Please consider a concluding statement about the extensibility of this approach and any specific caveats e.g., temporal resolution, validation, etc that should be factored in for others who might be interested in adopting this approach in the context of their own time-series efforts.

Response We have added the following text as concluding statement on the paper:

In order to use this approach one should consider some details about the experimental design. The chosen (micro)biological system should be sampled at time intervals that are relevant for the research question (e.g., cell doubling times if one wants to study microbial community composition dynamics) and spanning multiple time cycles.

Lines 741-742 Change to “Considering we required a clustering approach that is independent of scale we computed the Pearson correlations between pairs of EGs...”

Response We have changed the text as suggested.

Lines 767-769 Change to “(ARIMA, Prophet and neural network using the functions ARIMA, prophet and NNETAR, respectively, all implemented in the R package fable...”

Response We have changed the text as suggested.

Line 770 Change to “Each signal was modelled as a separate...”

Response We have amended the manuscript as suggested.

Line 780 Change to “set...”

Response We have changed the text accordingly.

Line 793 Change to “This was...”

Response We have changed the text accordingly.

Additional changes

We made the following further changes to the text to improve clarity, readability and comply with the journal guidelines.

Title We changed the title from: “**Forecasting of a complex microbial community using meta-omics**” to “**Forecasting the dynamics of a complex microbial community using integrated meta-omics**”.

Abstract We have rewritten the **Abstract** section to fit within 150 words and went from reading:

Microbial communities are complex assemblages whose dynamics are shaped by abiotic and biotic factors. A major challenge concerns correctly forecasting community behaviour. In this context, communities in biological wastewater treatment plants (BWWTPs) represent excellent model systems, because forecasting them is required to ultimately control and operate the plants in a sustainable manner. Here, we forecast the microbial community from the water-air interface of the anaerobic tank of a BWWTP via longitudinal meta-omics (metagenomics, metatranscriptomics and metaproteomics) data covering 14 months at weekly intervals. We extracted all the available time-dependent information, summarised it in 17 temporal signals (explaining 91.1% of the temporal variance) and linked these over time to rebuild the sequence of ecological phenomena underpinning the community dynamics. We forecast the signals over the following five years and tested the predictions with 21 extra samples. We were able to correctly forecast five signals accounting for 22.5% of the time-dependent information in the system and generated mechanistic predictions on the ecological events in the community (e.g. a predation cycle involving bacteria, viruses and amoebas). Through the forecasting of the 17 signals and the environmental variables readings, we reconstructed the gene abundance and expression for the following 5 years, showing a nearly perfect trend prediction (coefficient of determination ≥ 0.97) for the subsequent 2 years. Our study demonstrates the chance of microbial ecology to forecast composition and gene expression of open microbial ecosystems using year-spanning interactions between community cycles and environmental parameters.

76to being as follows:

Predicting the behavior of complex microbial communities is challenging. This is however essential for complex biotechnological processes such as biological wastewater treatment plants (BWWTPs) which require sustainable operation. Here we summarise 14 months of longitudinal meta-omics data from a BWWTP anaerobic tank into 17 temporal signals, explaining 91.1% of the temporal variance, and linked those signals to ecological events within the community. We forecast the signals over the five subsequent years and use 21 extra samples collected at defined time intervals for testing and validation. Our forecasts are correct for six signals and hint on phenomena such as predation cycles. Using all the 17 forecasts and the environmental variables, we predict gene abundance and expression, with a coefficient of determination ≥ 0.87 for the first subsequent three years. Our study demonstrates the ability to forecast the dynamics of open microbial ecosystems using interactions between community cycles and environmental parameters.

Line 104 We moved the following section to the Supplementary Information section under the name “**Time series analysis of biological data**”:

There exist several categories of time-series analysis. These are based on: i) previous knowledge (such as curve fitting²² and classification^{23,24}), ii) subsetting (e.g. segmentation²⁵), iii) clustering (e.g. based on various metrics such as Euclidean Distance²⁶ or Dynamic Time Warping²⁷) iv) prediction (such as forecasting²⁸ and intervention analysis²⁹), and v) decomposition (e.g. Singular Value Decomposition - SVD³⁰). The prediction of future states of ecological communities and their interplay with the environment have been successfully tackled in the case of available interaction models and/or limited number of species^{31,32}. However, predictions of microbial metabolic behaviour are rendered challenging for naturally occurring microbial ecosystems as well as industrially-relevant ones, such as in BWWTPs. In this context, metagenomics (MG)^{33,34}, metatranscriptomics (MT)³⁵ and metaproteomics (MP)³⁶ enable the establishment of meta-omic sample-specific reference databases that simultaneously resolve both compositional and functional aspects of the system.

Line 118 We have added a pointer to the Supplementary Information section after removing the paragraph above. We amended the sentence:

When dealing with complex and uncharacterised microbial systems, far from lab-scale experiments, empirical modelling can enable efficient representation and forecasting.

to read as:

*When dealing with complex microbial communities, far from lab-scale experiments, empirical modelling can enable efficient representation and forecasting (see **Supplementary Information** for a short summary of techniques).*

Line 156 We have corrected “*following*” to “*subsequent*”.

Line 160 We have corrected the text from reading:

[...] were trained to forecast the following five years' signals.

To read as follows:

[...] were trained to forecast the signals for the subsequent five years.

Line 174 We modified the header from “Functional and genetic characterization of LAO” to “**Characterization of the microbial community**”.

Line 175 We have corrected “*multi-omics*” to “*meta-omics*”.

Line 374 We have corrected “**Supplementary Figure 8**” to “**Supplementary Figure 7**”.

Line 206 We have added “ as depicted in **Supplementary Figure 2c**” at the end of the sentence.

Line 322 We have changed the header of the section from “***The temporal domino of ecological events in LAO***” to “***The ecological events in the microbial community***”.

Line 330 We corrected “ten” to “sixteen”.

Line 385 We have corrected “**Supplementary Figure 6d**” to “**Supplementary Figure 4d**”.

Line 385 We have moved the following section in the Supplementary Information with the title “**Time-independent Fatty Acid and Triacylglycerol accumulation**”:

The second clique, C2, includes S9, S4 and S7 leading to S8. Both S4 and S8 represent oscillatory “perturbations” (Figure 2b, Supplementary Figure 4d). Whilst S4 is increasing in amplitude, S8 is decreasing. Interestingly, out of the four only S8 has an autoregressive component and S7 is missing any seasonal signal (Figure 2b). The nitrogen-associated S9 has a simple dependency on NH4 (Figure 2b) and indeed influences positively the family Nitrosomonadaceae (Supplementary Figure 7). S7 is weakly influenced by seasonality and has a relatively strong intercept (Figure 2b) but is affected by both pH and NH4. The bacterial taxonomic contributions to S7 show a mixed response of the transcriptome whereby the only positive MG association is with the viral family Mimiviridae. It is possible that S7 encodes fluctuations in the parameters and the immediate response of the microbiome (through RNA), without a defined overarching pattern. The pair S4 and S8 are however more intriguing, because of the counterintuitive idea that an escalating perturbation could contribute to the resolution of another

79perturbation. S4 is explained solely by seasonal components, whilst S8 also includes pH effects from both the sampling site and the inflow, even if with opposite effects (Figure 2b). The signal S4 is -in general- negatively associated with gene expression and protein levels, however it is positively impacted by the level of the putative predator Nannocystaceae⁴⁹. The functional associations of S8 include a negative one for porphyrin and chlorophyll and positive ones for glycerophospholipids and simple sugars, hinting at a switch between autotrophic and medium-dependent metabolisms in the foam community (Supplementary Figure 8). This seems to suggest that an interplay between the predation by the family Nannocystaceae, supported by parameter fluctuations in pH and NH4 might lead to further general instability in the RNA expression of the microbiome. Even more curious is how the exacerbation of the amplitude of S4 might drive the stabilisation of S8, according to the idea that higher predation levels have been linked to the stability of ecosystems⁵⁰. Moreover, S4 might play a role in the cyclical ecology of the system beyond the environmental variables (indeed we did not find any associated with it). On the other hand S5 shows no link to seasonality but is positively influenced by aeration. Among the taxa that contribute to it the most, there are families known to be involved in the bulking process such as Gordoniaceae and Zoogloeaceae. This points to a putative connection between the bulking process and the aeration, which is beyond seasonal effects.

Line 387 We have added the following sentence to the end of the paragraph to point to **Supplementary Figure 9** and the **Supplementary Information**:

*We generated an ecological hypothesis also for clique C2, and addressed specifically the temporal independence regarding presence and expression of pathways for Fatty Acid and Triacylglycerol in the community (**Supplementary Figure 9**). Both topics are discussed in the **Supplementary Information**.*

Line 388 We have corrected "**Supplementary Figure 10**" to "**Supplementary Figure 8**".

Line 426 We changed the section name from "*Fatty Acid and Triacylglycerol accumulation are mostly time-independent*" to "*Time-independent Fatty Acid and Triacylglycerol accumulation*" to fit within the 60 character limit. Then we moved the whole following section to the Supplementary Information to reduce the word count of the manuscript.

80Time-independent Fatty Acid and Triacylglycerol accumulation

For a LAO community, the biosynthesis of Triacylglycerol (TG) and Fatty Acids (FA) are crucial steps⁵¹ involving multiple enzyme classes and with several entry points (Figure 3a). The abundant and expressed classes cover the circuit going from Acyl-Phosphate (Acyl-P) to fatty acid (FA) as shown in Figure 3a, however none of the enzymes' quantities are in the top/bottom 5% of the loadings for the time-dependent EGs. It looks, in general, that the accumulation of TG and FA is time-independent. This is consistent with the observation that functions are mostly conserved in a BWWT¹⁸. Interestingly K22848 is mostly encoded and expressed by the family Moraxella which is one of the two dominant families in the system (Figure 3b). Together with Moraxella, plasmid-encoded enzymes are also present, which was previously unknown to our knowledge⁵², and indicates that the ability to convert DAG to TG can likely be shared between bacteria and across different taxonomic families.

Line 433 We have corrected "**Supplementary Figure 6**" to "**Supplementary Figure 4**".

Line 456 We have corrected "**Supplementary Figure 4**" to "**Supplementary Figure 3**".

Line 464 We have corrected corrected the text:

The 51 weeks spanning 2011-2012 data were used as a training set and the model with the smallest Root Mean Square Error (RMSE) was selected for forecasting.

To read as follows:

*The 51 weeks spanning 2011-2012 data were used as a training set as well as to select the three best scoring models to build a combined one (see **Methods**). In the end, the model with the smallest Root Mean Square Error (RMSE) was selected for forecasting.*

The five correctly forecast signals account for 22.5% of EV and 24.7% of the EV by the complete S1-17 model.

Line 468 We have corrected “*following*” to “*subsequent*”.

Line 475 We updated the text from:

The cases in which the modelling was fully successful were six: S1, S2, S4, S5, S10 and S16. However, it is worth noting that the training set for S10 was not fully captured by the model, and therefore we excluded it from further considerations. The six correctly forecast signals account for 34.4% of EV and 37.7% of the EV by the complete S1-17 model. However, the most common outcome of the validation was a good fit to the training set and an insufficient one in the testing (9 out of 17 cases), including signals from all the groups.

To read as follows:

The cases in which the modelling was fully successful were six: S1, S2, S4, S7, S9, S14 and S17. The six correctly forecast signals account for 22.5% of EV and 24.7% of the EV by the complete S1-17 model. However, the most common outcome of the validation was a good fit to the training set and an insufficient one in the testing (10 out of 17 cases), including signals from all the groups.

Line 476 We have corrected “**Supplementary Figure 6f**” to “**Supplementary Figure 4f**”.

Line 488 We have corrected “*following*” to “*subsequent*”.

Line 475 We updated the text from:

However, the forecasting and the testing hinted at a cyclical occurrence, hence what appeared like a crash is predicted to be constitutive and repeated behaviour.

To read as follows:

However, the forecasting hinted at a cyclical occurrence, hence what appeared like a crash is predicted to be constitutive and repeated behaviour.

Line 506 We updated the text from:

Well known bacteria involved in bulking such as Moraxellaceae and Gordoniaceae have loadings contributing toward S5, hinting to a quick jolt in thickening of the foam in Summer and an overall cyclical effect that can be forecast over time.

To read as follows:

Well known bacteria involved in bulking such as Moraxellaceae and Gordoniaceae have loadings contributing toward S5, hinting to a quick jolt in thickening of the foam in Summer and an overall cyclical effect that can be forecast over time.

Line 554 We have corrected “*multi-omic*” to “*meta-omic*”.

Line 554 We have corrected “The R^2 is strikingly high (≥ 0.97) [...]” to “The R^2 is strikingly high (≥ 0.89) [...]”.

Line 560 We have corrected “**Figure 4**” to “**Figure 3**”.

Line 561 We have updated the text to read from:

The R^2 is strikingly high (≥ 0.97) in all the six matrices for the first two years following the training set but the predictability starts decreasing from the third year after the last sample.

to:

The R^2 is strikingly high (≥ 0.87) in all the six matrices for the subsequent three years after the training set but the predictability starts decreasing from the fourth year after the training samples.

Line 564 We have updated the text to read from:

[...] at any given point within the two years following the sampling.

to:

[...] at any given point within the subsequent three years after the training set.

Line 577 We have updated the text from:

We demonstrate that five of the forecast signals (S1, S2, S4, S5 and S16) are indeed validated by the future samples (Figure 3) and cover some interesting aspects of the BWWTP surface community like Nitrogen metabolism (S4) and viral interplay (S1 and S16), as well as well-known foam-related dynamics (S5).

to read as follows:

We demonstrate that six of the forecast signals (S1, S4, S7, S9, S14 and S17) are indeed validated by the future samples (Figure 3) and cover some interesting aspects of the BWWTP surface community like Nitrogen metabolism (S4 and S9) and viral interplay (S1 and possibly S7), as well as changes in the foam-related metabolism (S17).

Line 584 We have added the pointer to the paragraph moved to the Supplementary Information. The original text:

The signals were tied in a “temporal domino” (Figure 2b), from which we selected two cliques to successfully describe: the “autumn crash” (C1) and an oscillatory perturbation (C2).

has been amended as:

*The signals were tied in a “temporal domino” (Figure 2b), from which we selected two cliques to successfully describe: the “autumn crash” (C1) and an oscillatory perturbation (C2, in the **Supplementary Information**).*

Line 589 We have updated the text from:

We demonstrate that five of the forecast signals (S1, S2, S4, S5 and S16) are indeed validated by the future samples (Figure 3) and cover some interesting aspects of the BWWTP surface community like Nitrogen metabolism (S4) and viral interplay (S1 and S16), as well as well-known foam-related dynamics (S5).

to read as follows:

Importantly, when rebuilding the gene abundance and expression data at the levels of taxonomic families, reactions and pathways and extrapolating to the future samples (June 2012-2016) the results over the averaged month of June showed a very high degree of predictability for the subsequent three years after the training set ($R^2 \geq 0.87$).

Line 598 We have corrected “[...] for the first two years after sampling ($R^2 \geq 0.97$).” to “[...] for the first two years after sampling ($R^2 \geq 0.89$).”.

Line 669 We have corrected the text from:

[...] using a Pearson Correlation Coefficient threshold of 0.7. For each cluster of correlated variables a single one was selected, resulting in 15 variables used from the 59 initial ones.

to read as follows:

[...] using the Pearson Correlation Coefficients to allow a rational selection, resulting in 15 variables used from the 59 initial ones.

Lines 717-on We added the version of the following tools: EukRep v0.6.7, Plasflow v1.1.0, cbar v1.2, virstorter v1.0.6, deepvirfinder v1.0, dRep v0.5.4, CheckM v1.0.7, CD-HIT v4.6.8, Contig Annotation Tool (CAT) and Bin Annotation Tool (BAT) v5.1.2, samtools v1.11, bam2hits v1.0.9, mmseq v1.0.9, fable v0.3.1, Imtest v0.9-38, rEDM v1.14.0.

Line 803 We have corrected “*transformtranform*” to “*transform*”.

Line 812 We have corrected “*complex effectsones*” to “*complex ones*”.

Line 813 We have corrected “**Supplementary Figure 5**” to “**Supplementary Figure 3**”.

Line 825 We have corrected “Supplementary Figure 8” to “**Supplementary Figure 5**”.

Line 854 We have corrected the text from:

The RMSE was calculated for the fourteen models as well. For each signal, the model with the lowest RMSE was selected for the putative generative process. and used to forecast the test set with the function forecast from the fable package and supplying environmental parameter readings.

and now it reads as follows:

*The RMSE was calculated for the fourteen models as well (**Supplementary Figure 6**). For each signal, the model with the lowest RMSE was selected for the putative generative process and used to forecast the test set with the function forecast from the fable package and supplying environmental parameter readings.*

Line 874 We have corrected “**Supplementary Figure 12**” to “**Supplementary Figure 10**”.

Figure 2 We updated panel **b**, even if there are no appreciable differences. We updated **Figure 2** from:

to:

nature portfolioFigure 3 We updated **Figure 3** from:

to:

nature portfolioFigure 3 We updated **Figure 4** from:

to:Supplementary Figure 6 We updated the figure from:

to:

Supplementary Figure 10 We updated the figure from:

to:Data and code availability We moved the section after Methods and split it into two distinct ones as requested by the editors. The sections now read as:

Data availability

The generated MG and MT reads (FASTQ) files, as well as the previously produced data, are available as NCBI BioProject PRJNA230567. The MP data from the PRIDE repository with accession number PXD013655⁴³.

Code availability

The meta-omics pipeline IMP v3.0⁵³ is maintained and developed at the GitLab page: <https://git-r3lab.uni.lu/IMP/imp3>. The code used in the analysis is available at https://git-r3lab.uni.lu/ESB/lao/lao_ts and <https://github.com/fdelogu/microforecast>, whilst the data required to start the analysis is available on Zenodo with the doi 10.5281/zenodo.7225349. The full list of software and R package versions are listed in the Git pages.

Author contributions statement We moved the statement after the **Acknowledgments** section.

Competing interests statement We added the statement which reads as follows:

Competing interests

The authors declare no competing interests.

Captions We moved all the legends to the end of the manuscript and complied with the editors' requests to add further information for the caption of **Figure 2**, which now reads as follows:

Figure 2. Eigengene modelling using ARIMA augmented with environmental parameters and Fourier terms. **a.** The signals S1-17 encapsulate the time-dependent dynamics underlying the microbial community. The scale of the y-axis is dimensionless as the eigenvectors. **b.** The S1-17 are explained as ARIMA processes under the influence of the environmental variables. The five blocks of explanatory variables are: Model (ARIMA components), Manual (manually collected environmental variables, directly on the sampling location), Inflow (inflow stream of wastewater in the plant), V1 (first anaerobic tank in the plant), and V2 (second anaerobic tank in the plant). Every circle represents a significant variable according to the ANOVA test (Benjamini-Hochberg adjusted $p < 0.05$) for the corresponding signals among S1-17, the size represents the value of the coefficient, the ring colour its sign and the fill colour the $\log_{10}(p\text{-value})$. **c.** The signals are connected by a temporal transfer of information, suggesting a succession of ecological events. The signals with a purple edge are putatively nonlinear and their relationships have been confirmed with Convergent Cross Mapping analysis. The dashed lines indicate weak transfer of information whilst a full arrow and a hollow one represent an imbalance in information transfer (in favor of the solid arrow).

We added the requested information in **Figure 3**, which now reads as follows:

Figure 3. Forecasting of the signals. The 17 signals are predicted for the years 2011-2016 and compared with the data from June for those years. The green and blue dots represent the training and test data respectively; the solid line depicts the median of the prediction whilst the shaded area represents the 95% confidence interval. The green and blue boxplot on the right of every box depict the distributions of the model residuals from the training and the test sets, respectively. Corresponding scales are provided on the right y-axis. The residue displacement from the null distribution was assessed by a Wilcoxon two-sided test ($n=21$). The star on top of the boxplot indicates a statistical difference (BY corrected p value < 0.01) between the mean of the residual distribution and 0, indicating an incorrect/incomplete modelling (exact p values in **Supplementary Table 7**). In the boxplots, the central line indicates the second quartile, the lower and upper hinges correspond to the first and third quartiles and the whiskers extend from the hinge to the smallest/largest value no further than ± 1.5 x the distance between the first and third quartiles. The samples beyond the range are plotted as individual outlier dots.

Final Decision Letter:

2nd October 2023

Dear Professor Wilmes,

We are pleased to inform you that your Article entitled "Forecasting the dynamics of a complex microbial community using integrated meta-omics", has now been accepted for publication in Nature Ecology & Evolution.

Over the next few weeks, your paper will be copyedited to ensure that it conforms to Nature Ecology and Evolution style. Once your paper is typeset, you will receive an email with a link to choose the appropriate publishing options for your paper and our Author Services team will be in touch regarding any additional information that may be required

Due to the importance of these deadlines, we ask you please us know now whether you will be difficult to contact over the next month. If this is the case, we ask you provide us with the contact information (email, phone and fax) of someone who will be able to check the proofs on your behalf, and who will be available to address any last-minute problems . Once your paper has been scheduled for online publication, the Nature press office will be in touch to confirm the details.

Acceptance of your manuscript is conditional on all authors' agreement with our publication policies (see www.nature.com/authors/policies/index.html). In particular your manuscript must not be published elsewhere and there must be no announcement of the work to any media outlet until the publication date (the day on which it is uploaded onto our web site).

Please note that *Nature Ecology & Evolution* is a Transformative Journal (TJ). Authors may publish their research with us through the traditional subscription access route or make their paper immediately open access through payment of an article-processing charge (APC). Authors will not be required to make a final decision about access to their article until it has been accepted. [Find out more about Transformative Journals](https://www.springernature.com/gp/open-research/transformative-journals)

Authors may need to take specific actions to achieve [compliance with funder and institutional open access mandates](https://www.springernature.com/gp/open-research/funding/policy-compliance-faqs). If your research is supported by a funder that requires immediate open access (e.g. according to [Plan S principles](https://www.springernature.com/gp/open-research/plan-s-compliance)) then you should select the gold OA route, and we will direct you to the compliant route where possible. For authors selecting the subscription publication route, the journal's standard licensing

100terms will need to be accepted, including <https://www.nature.com/nature-portfolio/editorial-policies/self-archiving-and-license-to-publish>. Those licensing terms will supersede any other terms that the author or any third party may assert apply to any version of the manuscript.

We welcome the submission of potential cover material (including a short caption of around 40 words) related to your manuscript; suggestions should be sent to Nature Ecology & Evolution as electronic files (the image should be 300 dpi at 210 x 297 mm in either TIFF or JPEG format). Please note that such pictures should be selected more for their aesthetic appeal than for their scientific content, and that colour images work better than black and white or grayscale images. Please do not try to design a cover with the Nature Ecology & Evolution logo etc., and please do not submit composites of images related to your work. I am sure you will understand that we cannot make any promise as to whether any of your suggestions might be selected for the cover of the journal.

You can generate the link yourself when you receive your article DOI by entering it here: <http://authors.springernature.com/share>.

[REDACTED]

P.S. Click on the following link if you would like to recommend Nature Ecology & Evolution to your librarian <http://www.nature.com/subscriptions/recommend.html#forms>

** Visit the Springer Nature Editorial and Publishing website at http://editorial-jobs.springernature.com?utm_source=ejp_NEcoE_email&utm_medium=ejp_NEcoE_email&utm_campaign=ejp_NEcoE for more information about our career opportunities. If you have any questions please click [here](mailto:editorial.publishing.jobs@springernature.com). **